# Do Neural Networks Need Gradient Descent to Generalize? A Theoretical Study

**Yotam Alexander**
Tel Aviv University
yotam.alexander@gmail.com

**Yonatan Slutzky**
Tel Aviv University
slutzky1@mail.tau.ac.il

**Yuval Ran-Milo**
Tel Aviv University
yuv.milo@gmail.com

**Nadav Cohen**
Tel Aviv University
cohennadav@tauex.tau.ac.il

## Abstract

Conventional wisdom attributes the mysterious generalization abilities of over-parameterized neural networks to gradient descent (and its variants). The recent volume hypothesis challenges this view: it posits that these generalization abilities persist even when gradient descent is replaced by Guess & Check (G&C), *i.e.*, by randomly drawing weight settings until one that fits the training data is found. The validity of the volume hypothesis for wide and deep neural networks remains an open question. In this paper, we theoretically investigate this question for matrix factorization (with linear and non-linear activation): a canonical testbed in neural network theory. We first prove that generalization under G&C deteriorates with increasing width, establishing what is, to our knowledge, the first canonical case where G&C is provably inferior to gradient descent. Conversely, we prove that generalization under G&C improves with increasing depth, revealing a stark contrast between wide and deep networks, which we further validate empirically. These findings suggest that even in simple settings, there may not be a simple answer to the question of whether neural networks need gradient descent to generalize well.

## 1 Introduction

*Overparameterized neural networks* trained by (variants of) *gradient descent* are a cornerstone of modern artificial intelligence (AI) [117, 51, 3, 17, 64, 3]. Typically, an overparameterized neural network can fit its training data with any of multiple weight settings, some of which *generalize* well (*i.e.*, perform well on unseen test data), while others do not. The fact that weight settings found by gradient descent often generalize well is a mystery attracting vast attention [134, 57, 90, 87, 85]. Conventional wisdom states that this phenomenon stems from a special implicit bias induced by gradient descent when applied to overparameterized neural networks [107, 44, 63, 70].

Recently, it has been argued that gradient descent is not necessary for overparameterized neural networks to generalize well, and in fact, any reasonable (non-adversarial) optimizer that fits the training data can suffice [22, 18, 116, 13]. Notable empirical support for this argument was provided by Chiang et al. [22], who demonstrated that generalization comparable to that of gradient descent can be attained by *Guess and Check* (*G&C*), *i.e.*, by repeatedly drawing weight settings from a specified *prior distribution*, until a weight setting that happens to fit the training data is drawn. For a particular prior distribution over weight settings, hypothesizing that G&C attains good generalization is equivalent to the so-called *volume hypothesis* [22, 89], which states the following. Define the *volume* of a collection of weight settings to be the probability assigned to it by a *posterior distribution* obtained from conditioning the prior distribution on the training data being fit. Then, the volume of weight settings that generalize well is much greater than the volume of weight settings that do not.

39th Conference on Neural Information Processing Systems (NeurIPS 2025).

Aside from Chiang et al. [22], several works have supported the volume hypothesis in certain cases involving wide and deep overparameterized neural networks [18, 47, 48]. However, the literature also includes contrasting evidence. In particular, Peleg and Hein [89] systematically experimented with overparameterized neural networks of varying width and depth, and found that the generalization attainable by G&C is inferior to that of gradient descent, most prominently with larger network widths. Overall, the current literature on overparameterized neural networks presents a conflicting view of how the generalization attainable by G&C compares to that of gradient descent, and how this relationship depends on network width and depth.

In this paper, we present a theoretical study that takes a step toward clarifying the foregoing view, *i.e.*, toward delineating the extent to which wide or deep overparameterized neural networks need gradient descent in order to generalize well. Our theoretical study centers on *matrix factorization*: a canonical testbed in the theory of neural networks, used for studying generalization [43, 6, 73, 25, 127, 77] as well as other phenomena [40, 14, 108, 39]. Past analyses of matrix factorization have contributed to real-world neural networks—yielding theoretical insights [4, 5], concrete mathematical tools [96, 97], and practical methods that improve empirical performance [61, 109]. In its basic form, matrix factorization is akin to using overparameterized neural networks with linear (no) activation for tackling the low rank matrix sensing problem. We consider a more general form that allows for alternative (non-linear) activations as well [93].

Our first contribution is a theorem proving that, with an anti-symmetric activation (*e.g.*, linear, tanh or sine), if the width of a network increases, then the generalization attained by G&C deteriorates, to the point of being no better than chance—or more precisely, no better than the generalization attained by drawing a single weight setting from the prior distribution while disregarding the training data. The theorem applies to any prior distribution satisfying mild conditions, and in particular to the canonical Gaussian and uniform prior distributions considered in previous works [18, 47]. In light of known results proving that gradient descent attains good generalization [23, 75, 76, 63, 107], we conclude that there are canonical cases where the generalization attainable by G&C (with a conventional prior distribution) is provably inferior to that of gradient descent—that is, canonical cases where overparameterized neural networks provably need gradient descent in order to generalize well. To our knowledge, this is the first establishment of the existence of such cases.[1]

As a second contribution, we provide a theorem proving—for linear activation and a normalized Gaussian prior distribution—that if the depth of a network increases, then the generalization attained by G&C improves, to the point of being perfect. This theorem, which essentially implies that increasing network depth renders gradient descent not necessary for good generalization, stands in stark contrast to our analysis of increasing network width. We empirically showcase this contrast, demonstrating that in matrix factorization, the generalization attained by G&C improves with network depth but deteriorates with network width, whereas gradient descent attains good generalization throughout.

The findings in this paper suggest that even in simple settings, there may not be a simple answer to the question of whether neural networks need gradient descent to generalize well: the answer may hinge on subtle dependencies between network width and depth. We hope that our study of matrix factorization will serve as a stepping stone toward deriving a complete answer for real-world settings, thereby illuminating the role of gradient descent in modern AI.

## 1.1 Paper Organization

The remainder of the paper is organized as follows. Section 2 reviews related work. Section 3 introduces notation and the setting we study. Section 4 delivers our theoretical analysis, followed by Section 5 which presents an empirical demonstration. Section 6 discusses the limitations of our theory. Finally, Section 7 concludes.

---

[1]It is relatively straightforward to contrive *some* cases where the generalization attainable by G&C (with a conventional prior distribution) is provably inferior to that of gradient descent: indeed, one can retroactively define the ground truth distribution to be such that gradient descent attains perfect generalization. The novelty of our conclusion is that the foregoing provable inferiority arises in cases that are canonical, *i.e.*, that are fundamental and extensively studied.

## 2 Related Work

Numerous works have been devoted to understanding why overparameterized neural networks trained by gradient descent (or variants thereof) often generalize well [134, 57, 12, 59, 1, 11, 135, 86, 21, 105, 101, 42]. While this generalization is most commonly attributed to an implicit bias induced by gradient descent [104, 66, 88, 107, 44, 75, 58, 126, 23, 133, 7, 82, 76, 30, 128, 125, 80, 63, 121, 70, 95, 138, 94], an emerging view is that much of it stems from the architectures of neural networks. Results supporting this emerging view include: *(i)* results that establish a certain notion of simplicity when a weight setting is drawn from a prior distribution [13, 116, 54, 83, 32]; *(ii)* results that establish good generalization in a Bayesian framework, *i.e.*, when predictions are defined through an expectation over weight settings, where the probability of weight settings is higher the better they fit the training data [123, 56]; and *(iii)* results that establish good generalization with G&C, *i.e.*, when a weight setting is drawn from a posterior distribution obtained from conditioning a prior distribution on the training data being fit [47, 18, 48, 112]. Among these, the third category—*i.e.*, results concerning G&C—is arguably the most aligned with the standard learning paradigm, as it involves selecting a single weight setting that fits the training data.

Chiang et al. [22] and Peleg and Hein [89] compared the generalization attainable by G&C (with a conventional prior distribution) to that of gradient descent, by experimenting with overparameterized neural networks of varying width and depth. Chiang et al. [22] provided evidence suggesting that: *(i)* the generalization attainable by G&C is on par with that of gradient descent; and *(ii)* increasing the width of an overparameterized neural network improves the generalization of G&C. Peleg and Hein [89] pointed to confounding factors in the experimental protocol of Chiang et al. [22], and made different observations, namely: *(i)* the generalization attainable by G&C is inferior to that of gradient descent; *(ii)* increasing the width of an overparameterized neural network improves the generalization of gradient descent but not that of G&C; and *(iii)* increasing the depth of an overparameterized neural network deteriorates the generalization of both gradient descent and G&C. The latter observation does not align with the conventional wisdom, *i.e.*, with the extensive empirical and theoretical evidence that deep neural networks generalize better than shallow ones [6, 110, 79, 50]. Peleg and Hein [89] accordingly hedge this observation, effectively implying that it may result from confounding factors.

Our work is similar to those of Chiang et al. [22] and Peleg and Hein [89] in that it compares the generalization attainable by G&C (with a conventional prior distribution) to that of gradient descent for overparameterized neural networks of varying width and depth. It markedly differs from these past works in that it provides a rigorous theoretical analysis (the works of Chiang et al. [22] and Peleg and Hein [89] are purely empirical), and focuses on a canonical testbed model (matrix factorization). This allows for a controlled study free from confounding factors. Moreover, it allows us to conclude—for the first time, to our knowledge—that there exist canonical cases where the generalization attainable by G&C (with a conventional prior distribution) is provably inferior to that of gradient descent.[1]

## 3 Preliminaries

### 3.1 Notation

We use non-boldface lowercase letters for denoting scalars (*e.g.*, $\alpha \in \mathbb{R}$, $d \in \mathbb{N}$), boldface lowercase letters for denoting vectors (*e.g.*, $\mathbf{v} \in \mathbb{R}^d$), and non-boldface uppercase letters for denoting matrices (*e.g.*, $A \in \mathbb{R}^{d,d}$). For $d \in \mathbb{N}$, we define $[d] := \{1, \ldots, d\}$. We let $\|\cdot\|_2$ and $\|\cdot\|_F$ stand for the Euclidean norm of a vector and the Frobenius norm of a matrix, respectively.

### 3.2 Low Rank Matrix Sensing

*Low rank matrix sensing* is a fundamental and extensively studied problem in science and engineering [38, 113, 92, 19, 106, 127, 9]. In its basic form, the goal in low rank matrix sensing is to reconstruct a low rank matrix based on linear measurements. Namely, for $m, m', r, n \in \mathbb{N}$, where $r < \min\{m, m'\}$ and $n < m \cdot m'$, the goal is to reconstruct a *ground truth* matrix $W^* \in \mathbb{R}^{m,m'}$ of rank $r$ based on $(A_i \in \mathbb{R}^{m,m'}, y_i \in \mathbb{R})_{i=1}^n$, where:

$$y_i = \langle A_i, W^* \rangle := \mathrm{Tr}(A_i^\top W^*), \quad i \in [n]. \tag{1}$$

We refer to $(A_i)_{i=1}^n$ as *measurement matrices*, and to $(y_i)_{i=1}^n$ as the corresponding *measurements*.

The above can be cast as a supervised learning problem. Indeed, we may identify a matrix $W \in \mathbb{R}^{m,m'}$ with the linear functional that maps $A \in \mathbb{R}^{m,m'}$ to $\langle A, W \rangle \in \mathbb{R}$. Our goal is then to learn the linear functional $W^*$ based on the *training data* $(A_i, y_i)_{i=1}^n$, *i.e.*, based on the training *instances* $(A_i)_{i=1}^n$ and their corresponding *labels* $(y_i)_{i=1}^n$. The training data induces a *training loss* defined over linear functionals (or equivalently, over matrices):

$$\mathcal{L}_{\text{train}} : \mathbb{R}^{m,m'} \to \mathbb{R}_{\geq 0} \ , \ \ \mathcal{L}_{\text{train}}(W) := \frac{1}{n} \sum_{i=1}^n \left( \langle A_i, W \rangle - y_i \right)^2 . \tag{2}$$

Any $W \in \mathbb{R}^{m,m'}$ that minimizes the training loss, *i.e.*, that fits the training data, necessarily coincides with $W^*$ on instances in $\text{span}\{A_i\}_{i=1}^n$ (meaning $\langle A, W \rangle$ coincides with $\langle A, W^* \rangle$ for all $A \in \text{span}\{A_i\}_{i=1}^n$). Accordingly, we quantify generalization (performance on unseen test data) through instances orthogonal to $\text{span}\{A_i\}_{i=1}^n$, or more precisely, via the following *generalization loss*:

$$\mathcal{L}_{\text{gen}} : \mathbb{R}^{m,m'} \to \mathbb{R}_{\geq 0} \ , \ \ \mathcal{L}_{\text{gen}}(W) := \frac{1}{|\mathcal{B}|} \sum_{A \in \mathcal{B}} \left( \langle A, W \rangle - \langle A, W^* \rangle \right)^2 , \tag{3}$$

where $\mathcal{B} \subset \mathbb{R}^{m,m'}$ is some orthonormal basis for the orthogonal complement of $\text{span}\{A_i\}_{i=1}^n$ (it is straightforward to show that $\mathcal{L}_{\text{gen}}(\cdot)$ is independent of the particular choice of $\mathcal{B}$).

Much of the literature on low rank matrix sensing concerns a canonical special case where the measurement matrices $(A_i)_{i=1}^n$ satisfy a *restricted isometry property* (*RIP*) as defined below [38]. Such a property holds with high probability when $(A_i)_{i=1}^n$ are drawn from common distributions, for example Gaussian or Bernoulli [10].

**Definition 1.** We say that the measurement matrices $(A_i)_{i=1}^n$ satisfy a *restricted isometry property* (*RIP*) *of order* $\rho \in \mathbb{N}$ *with a constant* $\delta \in (0, 1)$, if for every matrix $W \in \mathbb{R}^{m,m'}$ whose rank is at most $\rho$, it holds that:

$$(1 - \delta) \|W\|_F^2 \leq \|\mathcal{A}(W)\|_2^2 \leq (1 + \delta) \|W\|_F^2 ,$$

where $\mathcal{A}(W) := (\langle A_1, W \rangle, \ldots, \langle A_n, W \rangle)^\top \in \mathbb{R}^n$.

### 3.3 Matrix Factorization

*Matrix factorization* is a canonical testbed in the theory of neural networks, used for studying generalization [43, 6, 73, 25, 127, 77] as well as other phenomena [40, 14, 108, 39]. Past analyses of matrix factorization have contributed to real-world neural networks—yielding theoretical insights [4, 5], concrete mathematical tools [96, 97], and practical methods that improve empirical performance [61, 109].

In its basic form, matrix factorization is akin to using overparameterized neural networks with linear (no) activation for tackling the low rank matrix sensing problem described in Section 3.2. We consider a more general form that allows for alternative (non-linear) activations as well [93]. Concretely, in our context, a matrix factorization with *activation* $\sigma : \mathbb{R} \to \mathbb{R}$, *width* $k \in \mathbb{N}$ and *depth* $d \in \mathbb{N}_{\geq 2}$, refers to learning a matrix $W \in \mathbb{R}^{m,m'}$ aimed at approximating the ground truth rank $r$ matrix $W^*$, through the following parameterization:

$$W = W_d \, \sigma(W_{d-1} \, \sigma(W_{d-2} \cdots \sigma(W_1)) \cdots) , \tag{4}$$

where $W_1 \in \mathbb{R}^{k,m'}$, $W_j \in \mathbb{R}^{k,k}$ for all $j \in \{2, \ldots, d-1\}$, $W_d \in \mathbb{R}^{m,k}$, and the application of $\sigma(\cdot)$ to a matrix signifies an application of $\sigma(\cdot)$ to each of the matrix's entries. We refer to $W_1, \ldots, W_d$ as the *weight matrices* of the factorization, and to a value assumed by $(W_1, \ldots, W_d)$ as a *weight setting*. Our interest lies in the overparameterized regime, where the width $k$ does not restrict the rank of the learned matrix $W$. Accordingly, we assume throughout that $k \geq \min\{m, m'\}$.

The low rank matrix sensing losses $\mathcal{L}_{\text{train}}(\cdot)$ and $\mathcal{L}_{\text{gen}}(\cdot)$ in Equations (2) and (3), respectively, induce training and generalization losses for the matrix factorization. With a slight overloading of notation, these are:

$$\mathcal{L}_{\text{train}}(W_1, \ldots, W_d) := \frac{1}{n} \sum_{i=1}^n \left( \langle A_i, W_d \, \sigma(W_{d-1} \cdots \sigma(W_1) \cdots) \rangle - y_i \right)^2 , \tag{5}$$

and:

$$\mathcal{L}_{\text{gen}}(W_1, \ldots, W_d) := \frac{1}{|\mathcal{B}|} \sum_{A \in \mathcal{B}} \left( \langle A, W_d \, \sigma(W_{d-1} \cdots \sigma(W_1) \cdots) \rangle - \langle A, W^* \rangle \right)^2 . \tag{6}$$

## 3.4 Gradient Descent

Similar to real-world neural networks, matrix factorization admits a non-convex training loss, for which a baseline optimizer is *gradient descent* emanating from small random initialization [26, 114]. Various studies—theoretical [26, 14, 43, 39, 137] and empirical [6, 35, 16]—were devoted to training matrix factorization with this baseline optimizer. In our context, this amounts to implementing the following iterations:

$$W_j^{(t+1)} \leftarrow W_j^{(t)} - \eta \frac{\partial}{\partial W_j} \mathcal{L}_{\text{train}}(W_1^{(t)}, \ldots, W_d^{(t)}) \ , \ j \in [d] \ , \ t \in \mathbb{N} \cup \{0\}, \qquad (7)$$

where $\mathcal{L}_{\text{train}}(\cdot)$ is the training loss defined in Equation (5), $\eta \in \mathbb{R}_{>0}$ is a predetermined *step size* (learning rate), and $(W_1^{(0)}, \ldots, W_d^{(0)})$ holds a randomly chosen initial weight setting of small magnitude.

## 3.5 Guess & Check

A conceptual alternative to gradient descent is *Guess and Check* (*G&C*) [22, 18, 48]. In the context of the matrix factorization described in Section 3.3, let $\mathcal{P}(\cdot)$ be a probability distribution over weight settings, *i.e.*, over values that may be assumed by the tuple of weight matrices $(W_1, \ldots, W_d)$. Regard $\mathcal{P}(\cdot)$ as a *prior distribution*, and let $\epsilon_{\text{train}} > 0$ be some threshold on the training loss $\mathcal{L}_{\text{train}}(\cdot)$ (Equation (5)). Applying G&C to the matrix factorization then consists of repeatedly drawing $(W_1, \ldots, W_d)$ from $\mathcal{P}(\cdot)$, until the condition $\mathcal{L}_{\text{train}}(W_1, \ldots, W_d) < \epsilon_{\text{train}}$ is met. From a statistical perspective, this is equivalent[2] to a single draw of $(W_1, \ldots, W_d)$ from $\mathcal{P}(\cdot | \mathcal{L}_{\text{train}}(W_1, \ldots, W_d) < \epsilon_{\text{train}})$, where the latter is the *posterior distribution* obtained from conditioning $\mathcal{P}(\cdot)$ on the event $\mathcal{L}_{\text{train}}(W_1, \ldots, W_d) < \epsilon_{\text{train}}$.

# 4 Theoretical Analysis

Consider a matrix factorization (Section 3.3) optimized by gradient descent (Section 3.4) or G&C (Section 3.5). A large body of theoretical work [43, 72, 6, 77, 34, 127, 136, 73, 106, 60, 131] has been devoted to establishing that gradient descent attains good generalization under various choices of width and depth for the factorization. In this section we tackle the question of whether gradient descent is needed for good generalization. Specifically, we theoretically analyze the generalization attainable by G&C as the width and depth of the factorization vary.

## 4.1 Distributions Over Weight Settings

Both G&C and gradient descent are defined with respect to a probability distribution over weight settings: for G&C it is the prior distribution (see Section 3.5), and for gradient descent it is the distribution from which initialization is drawn (see Section 3.4). We consider a broad class of distributions over weight settings specified by Definitions 2 and 3 below.

Definition 2 defines a *regular* distribution over $\mathbb{R}$ as one that has zero mean, is symmetric, and assigns positive probability to every neighborhood of the origin. This definition of regularity covers canonical distributions over $\mathbb{R}$, for example zero-centered Gaussian distributions and uniform distributions over symmetric intervals. Definition 3 builds on Definition 2 to specify the class of distributions over weight settings we consider. Namely, given a regular distribution (over $\mathbb{R}$) $\mathcal{Q}(\cdot)$, Definition 3 defines a distribution over weight settings *generated by* $\mathcal{Q}(\cdot)$ to be one in which entries are independently drawn from $\mathcal{Q}(\cdot)$, and then subject to Kaiming scaling [52], *i.e.*, to scaling that preserves magnitudes when the width of the factorization increases. This definition of a generated distribution covers Kaiming Gaussian and Kaiming Uniform distributions: common choices for the initialization of gradient descent [53, 130, 111] and the prior of G&C [18, 47, 48]. Definition 3 also defines a distribution over weight settings *generated by* $\mathcal{Q}(\cdot)$ *with normalization*, as one that is generated by $\mathcal{Q}(\cdot)$, with an additional normalization (scaling) that ensures the product of weight matrices has unit norm. The role of this normalization is to preserve magnitudes when the depth of the factorization increases, analogously to the role of normalization techniques applied when training real-world neural networks with large depth [115, 129, 103, 8, 55].

---

[2]See [37, 100] for folklore arguments justifying the equivalence.

**Definition 2.** Let $\mathcal{Q}(\cdot)$ be a probability distribution over $\mathbb{R}$. We say that $\mathcal{Q}(\cdot)$ is *regular* if the following conditions hold: *(i)* $\mathcal{Q}(\cdot)$ has zero mean and all of its moments exist, *i.e.*, $\mathbb{E}_{\alpha \sim \mathcal{Q}(\cdot)}[\alpha] = 0$ and $\mathbb{E}_{\alpha \sim \mathcal{Q}(\cdot)}[|\alpha^p|] < \infty$ for all $p \in \mathbb{N}$; *(ii)* $\mathcal{Q}(\cdot)$ is symmetric, meaning $\alpha \sim \mathcal{Q}(\cdot)$ implies $-\alpha \sim \mathcal{Q}(\cdot)$; and *(iii)* $\mathcal{Q}(\cdot)$ assigns positive probability to every neighborhood of the origin (*i.e.*, for any neighborhood $\mathcal{I}$ of $0 \in \mathbb{R}$, if $\alpha \sim \mathcal{Q}(\cdot)$ then the probability of the event $\alpha \in \mathcal{I}$ is positive).

**Definition 3.** Let $\mathcal{Q}(\cdot)$ be a regular probability distribution over $\mathbb{R}$ (Definition 2), and let $\mathcal{P}(\cdot)$ be a probability distribution over weight settings, *i.e.*, over values that may be assumed by the tuple of weight matrices $(W_1, \ldots, W_d)$. For every $j \in [d]$, denote by $m_j$ the number of columns in the weight matrix $W_j$, and by $\mathcal{Q}_j(\cdot)$ the probability distribution over $\mathbb{R}$ obtained from scaling $\mathcal{Q}(\cdot)$ by $1/\sqrt{m_j}$ (meaning $\alpha \sim \mathcal{Q}_j(\cdot)$ implies $\sqrt{m_j}\alpha \sim \mathcal{Q}(\cdot)$). We say that $\mathcal{P}(\cdot)$ is *generated by* $\mathcal{Q}(\cdot)$ if $(W_1, \ldots, W_d) \sim \mathcal{P}(\cdot)$ implies that $W_1, \ldots, W_d$ are statistically independent, and for every $j \in [d]$ the entries of $W_j$ are independently distributed per $\mathcal{Q}_j(\cdot)$. We say that $\mathcal{P}(\cdot)$ is *generated by* $\mathcal{Q}(\cdot)$ *with normalization* if $(W_1, \ldots, W_d) \sim \mathcal{P}(\cdot)$ can be implemented by drawing $(W_1, \ldots, W_d)$ from a distribution generated by $\mathcal{Q}(\cdot)$, and then, for all $j \in [d]$, dividing each entry of $W_j$ by $\|W_d \cdots W_1\|_F^{1/d}$.

## 4.2   Increasing Width: Need for Gradient Descent

In this subsection, we consider a regime where the width of the matrix factorization increases, and prove that gradient descent is needed for good generalization. In particular, we present an analysis leading to the conclusion that there exist canonical cases where the generalization attainable by G&C (with a prior distribution generated by a regular distribution) is provably inferior to that of gradient descent. To our knowledge, this is the first establishment of the existence of such cases.[1]

Definition 4 below defines an *admissible* activation as one that is non-constant, piece-wise continuously differentiable, has a polynomially bounded derivative, and does not vanish on both sides of the origin. This definition of admissibility covers most activations used in practice (*e.g.*, tanh, sigmoid, ReLU and Leaky ReLU [102, 68, 84, 78]).

**Definition 4.** We say that the activation $\sigma : \mathbb{R} \to \mathbb{R}$ is *admissible* if the following conditions hold: *(i)* $\sigma(\cdot)$ is non-constant; *(ii)* $\sigma(\cdot)$ is (continuous and) piece-wise continuously differentiable; *(iii)* the derivative of $\sigma(\cdot)$ is polynomially bounded, *i.e.*, there exist $p \in \mathbb{N}$ and $c \in \mathbb{R}_{>0}$ such that $|\sigma'(\alpha)| \leq c(1 + |\alpha|^p)$ for every $\alpha \in \mathbb{R}$ at which $\sigma'(\cdot)$ is defined; and *(iv)* $\sigma(\cdot)$ does not vanish on both sides of the origin, *i.e.*, any neighborhood of $0 \in \mathbb{R}$ includes some $\alpha \in \mathbb{R}$ for which $\sigma(\alpha) \neq 0$.

Theorem 1 below proves—for cases where the activation is admissible and anti-symmetric (*e.g.*, it is linear, tanh or sine)—that as the width of the factorization increases, the generalization attained by G&C deteriorates, to the point of being no better than chance, *i.e.*, no better than the generalization attained by drawing a single weight setting from the prior distribution while disregarding the training data. In the limit of width tending to infinity, the theorem applies to any prior distribution generated by some regular distribution over $\mathbb{R}$ (Definitions 2 and 3). In the canonical case where the regular distribution over $\mathbb{R}$ is a zero-centered Gaussian, the theorem also accounts for finite widths.

**Theorem 1.** *Suppose the activation $\sigma(\cdot)$ is admissible (Definition 4), and that it is anti-symmetric, meaning $\sigma(-\alpha) = -\sigma(\alpha)$ for all $\alpha \in \mathbb{R}$. Let $\mathcal{Q}(\cdot)$ be a regular probability distribution over $\mathbb{R}$ (Definition 2), and let $\mathcal{P}(\cdot)$ be the probability distribution over weight settings that is generated by $\mathcal{Q}(\cdot)$ (Definition 3). Let $\epsilon_{\text{train}}, \epsilon_{\text{gen}} \in \mathbb{R}_{>0}$. Regard $\mathcal{P}(\cdot)$ as a prior distribution, and consider the posterior distribution $\mathcal{P}(\cdot | \mathcal{L}_{\text{train}}(W_1, \ldots, W_d) < \epsilon_{\text{train}})$, i.e., the distribution obtained from conditioning $\mathcal{P}(\cdot)$ on the event that the training loss $\mathcal{L}_{\text{train}}(\cdot)$ is smaller than $\epsilon_{\text{train}}$. Then, as the width $k$ of the matrix factorization tends to infinity, the posterior probability of the event that the generalization loss $\mathcal{L}_{\text{gen}}(\cdot)$ is smaller than $\epsilon_{\text{gen}}$, converges to its prior probability, i.e.:*

$$\mathcal{P}\big(\mathcal{L}_{\text{gen}}(W_1, \ldots, W_d) < \epsilon_{\text{gen}} \mid \mathcal{L}_{\text{train}}(W_1, \ldots, W_d) < \epsilon_{\text{train}}\big) - \mathcal{P}\big(\mathcal{L}_{\text{gen}}(W_1, \ldots, W_d) < \epsilon_{\text{gen}}\big) \xrightarrow[k \to \infty]{} 0.$$

*Moreover, in the case where $\mathcal{Q}(\cdot)$ is a zero-centered Gaussian distribution, i.e., $\mathcal{Q}(\cdot) = \mathcal{N}(\cdot; 0, \nu)$ for some $\nu \in \mathbb{R}_{>0}$, it holds that for any $k$:[3]*

$$\mathcal{P}\big(\mathcal{L}_{\text{gen}}(W_1, \ldots, W_d) < \epsilon_{\text{gen}} \mid \mathcal{L}_{\text{train}}(W_1, \ldots, W_d) < \epsilon_{\text{train}}\big) - \mathcal{P}\big(\mathcal{L}_{\text{gen}}(W_1, \ldots, W_d) < \epsilon_{\text{gen}}\big) = O\big(\tfrac{1}{\sqrt{k}}\big).$$

---

[3]The $O$-notation below hides constants that depend on $\sigma(\cdot)$, $\epsilon_{\text{train}}$, $\epsilon_{\text{gen}}$ and $\nu$, as well as the ground truth matrix $W^*$, the measurement matrices $(A_i)_{i=1}^n$, the depth $d$ and the dimensions $m$ and $m'$ of the matrix factorization. See Appendix A.3 for details.

*Proof sketch (full proof in Appendix A).* The proof begins by establishing an equivalence between a matrix factorization and a feedforward fully connected neural network: each column of a factorized matrix $W$ (Equation (4)) can be seen as the output of a feedforward fully connected neural network when its input is a standard basis vector. This equivalence facilitates utilization of the following results from Hanin [45] and Favaro et al. [36]:

*(i)* as the width $k$ of a feedforward fully connected neural network tends to infinity, drawing its weight settings from any distribution $\mathcal{P}(\cdot)$ generated by some regular probability distribution $\mathcal{Q}(\cdot)$ (Definitions 2 and 3), leads the output of each of the network's layers to converge in distribution to a Gaussian process, whose covariance structure can be computed recursively using the covariance structures of previous layers; and

*(ii)* in the case where $\mathcal{Q}(\cdot)$ is a zero-centered Gaussian distribution, the convex distance (a standard distance metric for multivariate distributions) between the output of a network's layer and the Gaussian process to which it converges, is $O(1/\sqrt{k})$.

For treating the case where $\mathcal{Q}(\cdot)$ is an arbitrary regular probability distribution and the width $k$ tends to infinity, the proof utilizes result *(i)* above. Namely, it utilizes the recursive computation of covariance structures to show that the columns of the factorized matrix $W$ (*i.e.*, the vectors obtained by applying the feedforward fully connected neural network to standard basis vectors) converge in distribution to Gaussian vectors with statistically independent entries, and furthermore, since the activation $\sigma(\cdot)$ is anti-symmetric, these Gaussian vectors are statistically independent of one another. Overall, it holds that $W$ converges in distribution to a random matrix $W_{\text{iid}}$ whose entries are independently drawn from a zero-centered Gaussian distribution. Since the measurement matrices $(A_i)_{i=1}^n$ are orthogonal to the basis $\mathcal{B}$ that defines the generalization loss (see Equations (2) and (3)), the events $\mathcal{L}_{\text{train}}(W_{\text{iid}}) < \epsilon_{\text{train}}$ and $\mathcal{L}_{\text{gen}}(W_{\text{iid}}) < \epsilon_{\text{gen}}$ are statistically independent. Therefore, as the width $k$ of the matrix factorization tends to infinity, conditioning on the event that the training loss is lower than $\epsilon_{\text{train}}$ does not change the probability of the event that the generalization loss is lower than $\epsilon_{\text{gen}}$.

The proof concludes by treating the case where $\mathcal{Q}(\cdot)$ is a zero-centered Gaussian distribution and the width $k$ is finite. There, result *(ii)* above is utilized to show that the probabilities of the events $\mathcal{L}_{\text{train}}(W) < \epsilon_{\text{train}}$ and $\mathcal{L}_{\text{gen}}(W) < \epsilon_{\text{gen}}$ converge to those of the events $\mathcal{L}_{\text{train}}(W_{\text{iid}}) < \epsilon_{\text{train}}$ and $\mathcal{L}_{\text{gen}}(W_{\text{iid}}) < \epsilon_{\text{gen}}$, respectively, at a sufficiently fast rate. $\qquad\square$

Theorem 3.3 from Soltanolkotabi et al. [106]—restated as Proposition 1 below—is a representative result from the large body of work establishing that, in matrix factorization, gradient descent attains good generalization [43, 72, 6, 77, 34, 127, 136, 73, 60, 131]. The result proves—for cases where the activation is linear, the depth is two, and the measurement matrices satisfy an RIP (Definition 1)— that gradient descent (with small step size and small Kaiming Gaussian initialization) attains good generalization, with probability (over the initialization) tending to one as the width of the factorization increases. In light of this result, Theorem 1 leads to the conclusion that there exist canonical cases where the generalization attainable by G&C (with a prior distribution generated by a regular distribution) is provably inferior to that of gradient descent. To our knowledge, this is the first establishment of the existence of such cases.[1]

**Proposition 1** (restatement of Theorem 3.3 from [106]). *There exists a universal constant $c_1 \in \mathbb{R}_{>0}$ with which the following holds. Suppose the activation $\sigma(\cdot)$ is linear (i.e., $\sigma(\alpha) = \alpha$ for all $\alpha \in \mathbb{R}$), and the depth $d$ equals two. Let $\mathcal{Q}(\cdot)$ be a zero-centered Gaussian probability distribution, i.e., $\mathcal{Q}(\cdot) = \mathcal{N}(\cdot\,; 0, \nu)$, with variance $\nu \in \big(0, O(k^{-27/2})\big)$. Let $\mathcal{P}(\cdot)$ be the probability distribution over weight settings that is generated by $\mathcal{Q}(\cdot)$ (Definition 3). Assume the measurement matrices $(A_i)_{i=1}^n$ satisfy an RIP (Definition 1) of order $2r + 1$ (recall that $r$ is the rank of the ground truth matrix $W^*$) with a constant $\delta \in \big(0, \tilde{O}(1)\big)$. Consider minimization of the training loss $\mathcal{L}_{\text{train}}(\cdot)$ via gradient descent (Equation (7)) with initialization drawn from $\mathcal{P}(\cdot)$ and step size $\eta \in \big(0, \tilde{O}(1)\big)$. Then, there exists some $\tau \in \mathbb{N}, \tau = \tilde{O}(\eta^{-1})$, such that for any width $k$ of the matrix factorization, after $\tau$ iterations of gradient descent, with probability at least $1 - O(e^{-c_1 k})$ over its initialization, the generalization loss $\mathcal{L}_{\text{gen}}(\cdot)$ is $O(\nu^{3/10} k^{-3/20})$.[4]*

---

[4]Throughout the statement of Proposition 1, the $O$- and $\tilde{O}$-notations hide constants that depend on the dimensions $m$ and $m'$ of the matrix factorization, and on the ground truth matrix $W^*$. The $\tilde{O}$-notation also hides factors logarithmic in $k$ and $\nu$. See Appendix F for details.

## 4.3 Increasing Depth: No Need for Gradient Descent

In this subsection, we consider a regime where the depth of the matrix factorization increases, and prove that gradient descent is not necessary for good generalization. In particular, Theorem 2 below establishes cases where, as the depth of the factorization increases, the generalization attained by G&C improves, to the point of being perfect (in the sense that, with probability tending to one, the generalization loss is no greater than a constant times the threshold set for the training loss). This stands in stark contrast to our analysis of increasing width (Section 4.2), which established canonical cases where the generalization attainable by G&C (with a prior distribution generated by a regular distribution) is provably inferior to that of gradient descent. A large body of theoretical work has argued that neural networks benefit from depth more than from width in terms of expressiveness [81, 29, 28] and the implicit bias induced by gradient descent [41, 49, 97]. Our analyses suggest another potential advantage: depth may render neural network architectures more amenable to generalization than width, in the sense that the volume hypothesis holds to a greater extent.

The cases to which Theorem 2 applies are those where the activation is linear, the measurement matrices satisfy an RIP (Definition 1), the prior distribution is generated with normalization (Definition 3) from a zero-centered Gaussian distribution (over $\mathbb{R}$), and the ground truth matrix has norm and rank equal to one (an extension to ground truth matrices of higher rank is discussed in Appendix D). The theorem is non-asymptotic, meaning it applies to finite depths, not only to the limit of depth tending to infinity.

**Theorem 2.** *Suppose the ground truth matrix $W^*$ satisfies $\|W^*\|_F = 1$ and its rank $r$ equals one.[5] Suppose the activation $\sigma(\cdot)$ is linear (i.e., $\sigma(\alpha) = \alpha$ for all $\alpha \in \mathbb{R}$). Assume the measurement matrices $(A_i)_{i=1}^n$ satisfy an RIP (Definition 1) of order two with some constant $\delta \in (0, 1)$. Let $\mathcal{Q}(\cdot)$ be a zero-centered Gaussian probability distribution, i.e., $\mathcal{Q}(\cdot) = \mathcal{N}(\cdot\,; 0, \nu)$ for some $\nu \in \mathbb{R}_{>0}$. Let $\mathcal{P}(\cdot)$ be the probability distribution over weight settings that is generated by $\mathcal{Q}(\cdot)$ with normalization (Definition 3). Then, there exists $c \in \mathbb{R}_{>0}$ (dependent only on $\delta$) such that, for any $\epsilon_{\text{train}} \in \mathbb{R}_{>0}$ and any depth $d$ of the matrix factorization:[6]*

$$1 - \mathcal{P}\big(\mathcal{L}_{\text{gen}}(W_1, \ldots, W_d) < \epsilon_{\text{train}} c \,\big|\, \mathcal{L}_{\text{train}}(W_1, \ldots, W_d) < \epsilon_{\text{train}}\big) = O\big(\tfrac{1}{d}\big).$$

*Proof sketch (full proof in Appendix B).* The proof begins by decomposing the factorized matrix $W$ (Equation (4)) into a product of three matrices: $W = W_d W_{d-1:2} W_1$, where $W_{d-1:2} := W_{d-1} \cdots W_2$. Then, temporarily disregarding the normalization in $\mathcal{P}(\cdot)$, *i.e.*, temporarily treating $\mathcal{P}(\cdot)$ as if it were generated by $\mathcal{Q}(\cdot)$ without normalization (Definition 3), the proof applies concentration bounds established by Hanin and Paouris [46] to the *Lyapunov exponents* of $W_{d-1:2}$ (when normalization is disregarded, $W_{d-1:2}$ is a product of $d-2$ zero-centered Gaussian $k$-by-$k$ matrices). These Lyapunov exponents, denoted $\lambda_1, \ldots, \lambda_k$, are defined by $\lambda_i := \log(\sigma_i)/(d-2)$ for $i \in [k]$, where $\sigma_1 \geq \cdots \geq \sigma_k$ stand for the singular values of $W_{d-1:2}$. Hanin and Paouris [46] give non-asymptotic bounds on the deviation of $\lambda_1, \ldots, \lambda_k$ from their respective infinite depth ($d \to \infty$) limits $\mu_1, \ldots, \mu_k$, which satisfy $0 > \mu_1 > \ldots > \mu_k$. Since $\sigma_i = \exp((d-2)\lambda_i)$, and since $\mu_i \neq \mu_j$ for all $i \neq j$, the non-asymptotic bounds imply that when the depth $d$ is large, with high probability, $W_{d-1:2}$ has one singular value much larger than the rest, *i.e.*, $W_{d-1:2}$ has an approximate rank one structure.

Next, while still (temporarily) disregarding the normalization in $\mathcal{P}(\cdot)$, the proof uses properties of standard multivariate Gaussian distributions (namely, their rotational invariance and concentration bounds on their norms) to show that, with high probability, multiplication of $W_{d-1:2}$ by $W_d$ from the left and by $W_1$ from the right preserves its approximate rank one structure. In other words, with high probability, the factorized matrix $W = W_d W_{d-1:2} W_1$ has an approximate rank one structure. Now (and hereinafter) accounting for the normalization in $\mathcal{P}(\cdot)$, the latter finding is formalized as follows: for any $\gamma \in \mathbb{R}_{>0}$, the probability that $W$ is within $\gamma$ (in Frobenius norm) of a rank one matrix is $1 - O(1/d)$.

Choosing $\gamma$ appropriately, while employing compressed sensing arguments that rely on the RIP and the ground truth matrix having rank one (see, *e.g.*, [65]), it is then proven that the probability of the events $\mathcal{L}_{\text{train}}(W) < \epsilon_{\text{train}}$ and $\mathcal{L}_{\text{gen}}(W) \geq \epsilon_{\text{train}} c$ occurring simultaneously is $O(1/d)$. Finally, using the facts

---

[5]An extension of Theorem 2 to ground truth matrices of higher rank is discussed in Appendix D.

[6]The $O$-notation below hides constants that depend on the measurement matrices $(A_i)_{i=1}^n$, the ground truth matrix $W^*$, the dimensions $m$ and $m'$ of the matrix factorization, and the width $k$ of the matrix factorization. See Appendix B for details.

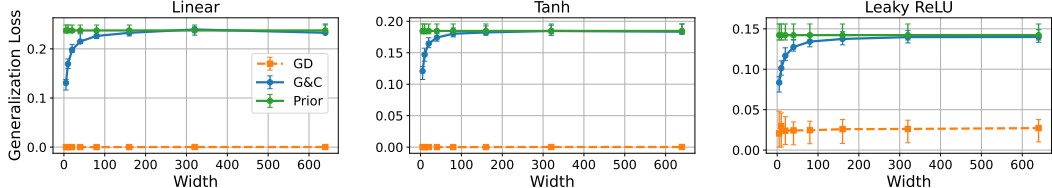

Figure 1: In line with our theory (Section 4.2), as the width of a matrix factorization increases, the generalization attained by G&C deteriorates, to the point of being no better than chance, *i.e.*, no better than the generalization attained by drawing a single weight setting from the prior distribution while disregarding the training data. In contrast, gradient descent attains good generalization across all widths. Each of the above plots corresponds to a matrix factorization as described in Section 3.3, with a different activation $\sigma(\cdot)$: linear activation ($\sigma(\alpha) = \alpha$) for the left plot; tanh activation ($\sigma(\alpha) = \tanh(\alpha)$) for the middle plot; and Leaky ReLU activation ($\sigma(\alpha) = \max\{c \cdot \alpha, \alpha\}$, with $c = 0.2$) for the right plot.[7] In each plot, the generalization loss (Equation (6)) is shown against the width of the matrix factorization, for three optimizers: gradient descent with small step size and small initialization (Section 3.4); G&C with a Kaiming Gaussian prior distribution (Section 3.5); and simply drawing a single weight setting from the prior distribution while disregarding the training data. For each combination of width and optimizer, we report the median (marker) and interquartile range (error bar) of generalization losses attained over eight trials (differing only in random seed). Across all experiments reported in this figure: the matrix factorization had depth two and dimensions $m = m' = 5$; the ground truth matrix had (Frobenius) norm and rank equal to one; and the training data size was $n = 15$. We note that with Leaky ReLU activation, which lies beyond the scope of our theory, the generalization attained by gradient descent is not as good as it is with linear and tanh activations. For further experiments and implementation details see Appendices G and H, respectively.

that the distribution of $W$ is rotationally invariant, that (due to the normalization in $\mathcal{P}(\cdot)$) the norm of $W$ equals one with probability one, and that $W$ is with high probability close to a rank one matrix, the proof establishes that the probability of $\mathcal{L}_{\text{train}}(W) < \epsilon_{\text{train}}$ is independent of the depth $d$, *i.e.*, it is $\Omega(1)$. By the definition of conditional probability, the latter two findings imply that the probability of $\mathcal{L}_{\text{gen}}(W) \geq \epsilon_{\text{train}}c$ conditioned on $\mathcal{L}_{\text{train}}(W) < \epsilon_{\text{train}}$ is $O(1/d)$. This is the desired result. □

## 5 Empirical Demonstration

In this section, we corroborate our theory (Section 4) by empirically demonstrating that in matrix factorization (Section 3.3), the generalization attained by G&C (Section 3.5) improves as depth increases but deteriorates as width increases, whereas gradient descent (Section 3.4) attains good generalization throughout. Figures 1 and 2 present such demonstrations, plotting generalization as a function of width and depth, respectively, for both G&C and gradient descent. The demonstrations in Figures 1 and 2 cover the theoretically analyzed linear and tanh activations, as well as the Leaky ReLU activation [78] which lies beyond the scope of our theory.[7] Additional demonstrations covering further cases (including gradient descent with momentum [91]) are provided in Appendix G. Code for reproducing all demonstrations can be found in `https://github.com/YoniSlutzky98/nn-gd-gen-mf`.

## 6 Limitations

It is important to acknowledge several limitations of our theory. First, while a large body of theoretical work [43, 72, 6, 77, 34, 127, 136, 73, 106, 60, 131] has been devoted to establishing that gradient descent attains good generalization in matrix factorization, Theorem 3.3 from [106] (restated as Proposition 1 herein)—which applies only when the activation is linear, the depth is two, and the measurement matrices satisfy an RIP (Definition 1)—is the only result at our disposal that formally guarantees low generalization loss with high probability for gradient descent with a positive (non-infinitesimal) step size and conventional (data-independent) initialization. Second, the guarantees we prove for G&C—namely, Theorems 1 and 2—include unspecified constant factors, and in particular, are non-vacuous only when the width or depth of the matrix factorization is sufficiently large. Third, Theorem 1 assumes that the activation is anti-symmetric. Fourth, Theorem 2 imposes even stronger assumptions: the activation is linear, the measurement matrices satisfy an RIP, and the ground truth matrix has norm and rank equal to one (an extension to ground truth matrices of higher rank is discussed in Appendix D, but is not formally established). Fifth, Theorem 1 requires the G&C

---

[7]We attempted to include a demonstration with the more popular ReLU activation [84], but its tendency to zero out matrix entries made G&C computationally infeasible, requiring an excessive number of draws to obtain a weight setting that fits the training data.

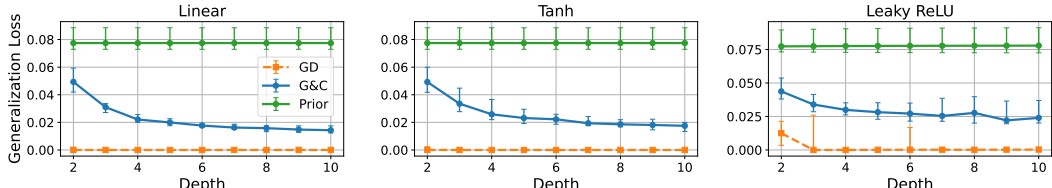

Figure 2: In line with our theory (Section 4.3), as the depth of a matrix factorization increases, the generalization attained by G&C improves, drawing closer to that of gradient descent. This figure adheres to the caption of Figure 1, except for the following differences: *(i)* the matrix factorization had variable (rather than fixed) depth and fixed (rather than variable) width, with the latter set to five; *(ii)* generalization losses are shown against the depth (rather than the width) of the factorization; and *(iii)* the prior distribution of G&C included normalization (Definition 3). We did not include depths greater than ten in our experiments, as they led to excessively long run times for gradient descent (due to vanishing gradients). Note that such greater depths would not necessarily lead the generalization attained by G&C to match that of gradient descent. Indeed, our theory for increasing depth (Theorem 2) guarantees that the generalization loss attained by G&C tends to zero only if the threshold set for its training loss (see Section 3.5) tends to zero, which makes G&C computationally infeasible (as it requires an infeasible number of draws). For further experiments and implementation details see Appendices G and H, respectively.

training loss threshold $\epsilon_{\text{train}}$ to be specified (the theorem does not rule out the possibility that for any width, a sufficiently small $\epsilon_{\text{train}}$ will lead G&C to attain good generalization), and although Appendix C proves a result that allows unspecified $\epsilon_{\text{train}}$, it does so under strong assumptions not imposed by Theorem 1. Finally, Theorems 1 and 2 consider different types of prior distributions: Theorem 1 excludes normalization (Definition 3), whereas Theorem 2 includes it.

While we empirically demonstrate that the conclusions of our theory hold beyond its formal scope, the above limitations remain. We hope that this paper will serve as a stepping stone towards addressing these limitations, and more broadly, towards extending our theory from matrix factorization to real-world neural networks.

# 7 Conclusion

Conventional wisdom attributes the miraculous generalization abilities of neural networks to gradient descent. A recent bold argument claims that gradient descent is not necessary for neural networks to generalize well, and in fact, any reasonable optimizer can suffice. This is justified by the so-called volume hypothesis, which posits that among the weight settings that fit the training data, the volume of the weight settings that generalize well is much greater than the volume of the weight settings that do not. While several works have supported the volume hypothesis in certain cases involving wide and deep neural networks, the literature also includes contrasting evidence.

In this paper, we presented a theoretical study for matrix factorization (with linear and non-linear activation)—a canonical and important testbed in the theory of neural networks—to rigorously examine the validity of the volume hypothesis. Our first contribution is a proof that the volume hypothesis fails when the width of a network is large (compared to its depth), thereby establishing—for the first time, to our knowledge—a canonical case where gradient descent is provably necessary for a neural network to generalize well. As a second contribution, we proved that the volume hypothesis holds when the depth of a network is large (compared to its width). These contributions reveal a stark contrast between wide and deep networks, which we further validated through empirical demonstrations.

Overall, our findings suggest that even in simple settings, whether the volume hypothesis holds may hinge on subtle dependencies between network width and depth. Delineating when it holds is of interest not only from a theoretical perspective, but also in practice. Indeed, when the volume hypothesis is known to hold, one may confidently employ fast-converging optimizers (*e.g.*, Adam [62] or AdamW [74]) without worrying about their potential to deteriorate generalization [124, 139].

Our study of matrix factorization may serve as a stepping stone toward analogous studies of more realistic models. A natural next step is the study of *tensor factorization*: a model obtained by lifting matrices (two-dimensional arrays) to *tensors* (multi-dimensional arrays). Tensor factorization has been studied extensively in the theory of neural networks [29, 28, 96, 98, 2], partly due to its ability to capture convolutional [27], recurrent [132] and self-attention [122, 69] architectures. Delineating when the volume hypothesis holds in tensor factorization—*i.e.*, when tensor factorization needs gradient descent to generalize well—would comprise an important milestone toward elucidating the role of gradient descent in modern AI.

## Acknowledgments and Disclosure of Funding

This work was supported by the European Research Council (ERC) grant NN4C 101164614, a Google Research Scholar Award, a Google Research Gift, Meta, the Yandex Initiative in Machine Learning, the Israel Science Foundation (ISF) grant 1780/21, the Tel Aviv University Center for AI and Data Science, the Adelis Research Fund for Artificial Intelligence, Len Blavatnik and the Blavatnik Family Foundation, and Amnon and Anat Shashua.

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

# A Proof of Theorem 1

This appendix proves Theorem 1. Appendix A.1 establishes an equivalence between a matrix factorization and a feedforward fully connected neural network. This equivalence allows us to utilize the theoretical results of Hanin [45] and Favaro et al. [36], developed for feedforward fully connected neural networks of large widths. Relying on these results: Appendix A.2 treats the case where $\mathcal{Q}(\cdot)$ is an arbitrary regular probability distribution and the width $k$ tends to infinity; and Appendix A.3 treats the case where $\mathcal{Q}(\cdot)$ is a zero-centered Gaussian distribution and the width $k$ is finite.

## A.1 An Equivalence Between Matrix Factorizations and Fully Connected Neural Networks

We begin by defining the concept of a fully connected neural network.

**Definition 5.** A *fully connected neural network* of depth $d \in \mathbb{N}$ with input dimension $m' \in \mathbb{N}$, output dimension $m \in \mathbb{N}$, hidden dimension $k \in \mathbb{N}$ and activation function $\sigma(\cdot)$ is a function $\mathbf{x}_\alpha \in \mathbb{R}^{m'} \mapsto \mathbf{z}_\alpha^{(d)} \in \mathbb{R}^m$ of the following recursive form:

$$
\mathbf{z}_\alpha^{(j)} = \begin{cases} W_1 \mathbf{x}_\alpha, & j = 1 \\ W_j \sigma(\mathbf{z}_\alpha^{(j-1)}), & i = 2, \ldots, d \end{cases},
$$

where $W_1 \in \mathbb{R}^{k,m'}$, $W_d \in \mathbb{R}^{m,k}$ and $W_2, \ldots, W_d \in \mathbb{R}^{k,k}$ are the networks weights, and $\sigma$ applied to a vector is shorthand for $\sigma$ applied to each entry.

Next, we prove a useful equivalence which shows that when a matrix factorization and a fully connected neural network share their weights and activation function, each of the columns of the former are equal to the outputs of the latter when input the appropriate standard basis vectors.

**Lemma 1.** *Let $\alpha \in [m']$. For any weight matrices $W_1, \ldots, W_d$ and activation function $\sigma(\cdot)$, the $\alpha$ column of the matrix factorization $W$ (Equation (4)) produced by the weight settings $(W_1, \ldots, W_d)$ and the activation function $\sigma(\cdot)$, is equal to the the output of the fully connected neural network (Definition 5) produced by the weights $(W_1, \ldots, W_d)$ and the activation function $\sigma(\cdot)$, when the input is $\mathbf{e}_\alpha \in \mathbb{R}^{m'}$, the standard basis vector holding $1$ in its $\alpha$ coordinate and zeros in the rest. Formally, we denote this as*

$$
[W]_{.\alpha} = \mathbf{z}_\alpha^{(d)},
$$

*where $[W]_{.\alpha}$ is the $\alpha$ column of $W$ and $\mathbf{z}_\alpha^{(d)}$ is the output of the fully connected neural network when the input is $\mathbf{e}_\alpha \in \mathbb{R}^{m'}$.*

*Proof.* We prove the claim via induction on $d$. First, for the base case, it trivially holds that

$$
[W_{1:1}]_{.\alpha} = W_{1:1} \mathbf{e}_\alpha = \mathbf{z}_\alpha^{(1)}.
$$

Next, fix $j \in [d]$ and assume that $[W_{1:j}]_{.\alpha} = \mathbf{z}_\alpha^{(j)}$. We thus have that

$$
\begin{aligned}
[W_{1:j+1}]_{.\alpha} &= W_{1:j+1} \mathbf{e}_\alpha \\
&= W_{j+1} \sigma(W_{1:j}) \mathbf{e}_\alpha \\
&= W_{j+1} \sigma(W_{1:j} \mathbf{e}_\alpha) \\
&= W_{j+1} \sigma([W_{1:j}]_{.\alpha}) \\
&= W_{j+1} \sigma(\mathbf{z}_\alpha^{(j)}) \\
&= \mathbf{z}_\alpha^{j+1},
\end{aligned}
$$

where the third equality is due to Lemma 21, and the fourth equality is due to the inductive assumption. With this we complete the proof. $\qquad\square$

## A.2 Proof for Arbitrary Regular Distribution and Infinite Width

The outline of the proof for the arbitrary prior case is as follows; Appendix A.2.1 presents Theorem 3, of which the arbitrary prior case of Theorem 1 is a special case. Appendix A.2.2 provides a useful Lemma used in the proof of Theorem 3. Appendix A.2.3 adapts a result from Hanin [45] showing that an infinitely wide matrix factorization converges in distribution to a centered Gaussian matrix (Definition 8). Finally, Appendix A.2.4 applies tools from probability theory to show that the latter convergence implies the conditions required for Lemma 2 in Appendix A.2.2.

### A.2.1 Restatement of the Arbitrary Prior Case of Theorem 1

The arbitrary prior case of Theorem 1 follows from Theorem 3, which allows for the distribution $\mathcal{Q}(\cdot)$ and the activation $\sigma(\cdot)$ to be slightly more general. Theorem 3 is presented below; afterwards, we demonstrate how it implies the arbitrary prior case of Theorem 1.

**Theorem 3.** *Let $d \in \mathbb{N}$ be a fixed depth. Let $\mathcal{Q}(\cdot)$ be some probability distribution on $\mathbb{R}$ which satisfies*

$$\mathbb{E}_{x \sim \mathcal{Q}(\cdot)}[x] = 0, \qquad \mathbb{E}_{x \sim \mathcal{Q}(\cdot)}[x^2] = 1,$$

*has finite higher moments and is symmetric, i.e., if $x \sim \mathcal{Q}(\cdot)$ then $-x \sim \mathcal{Q}(\cdot)$. Let $\sigma(\cdot)$ be an activation function that is not constant and anti-symmetric, i.e.,*

$$\forall x \in \mathbb{R}. \ \ \sigma(x) = -\sigma(-x),$$

*furthermore suppose that $\sigma$ is absolutely continuous, and that its almost-everywhere defined derivative is polynomially bounded, i.e:*

$$\exists p > 0 \text{ s.t. } \forall x \in \mathbb{R} \ \ \left\| \frac{\sigma'(x)}{1 + |x|^p} \right\|_{L^\infty(\mathbb{R})} < \infty.$$

*Suppose also that*

$$\mathbb{E}_{x \sim \mathcal{Q}(\cdot)}[\sigma^2(x)] > 0.$$

*Let $\epsilon_{\text{gen}}, \epsilon_{\text{train}}, c_W > 0$. Suppose that for any $j \in [d]$, the entries of $W_j \in \mathbb{R}^{m_{j+1}, m_j}$ are drawn independently by first sampling $x \sim \mathcal{Q}(\cdot)$ and then setting $[W_j]_{rs} = \sqrt{\frac{c_W}{m_j}} x$. Then the matrix factorization $W$ satisfies*

$$\lim_{k \to \infty} \mathcal{P}\left( \mathcal{L}_{gen}(W) < \epsilon_{\text{gen}} \Big| \mathcal{L}_{train}(W) < \epsilon_{\text{train}} \right) = \lim_{k \to \infty} \mathcal{P}\left( \mathcal{L}_{gen}(W) < \epsilon_{\text{gen}} \right).$$

Let $\mathcal{Q}(\cdot)$ be a regular distribution over $\mathbb{R}$ (Definition 2), and let $\sigma(\cdot)$ be an admissible activation function (Definition 4) that is anti-symmetric. First, observe that since Theorem 3 allows for arbitrary $c_W > 0$, the condition for $\mathbb{E}_{x \sim \mathcal{Q}(\cdot)}[x^2] = 1$ is satisfied with $c_W = \mathbb{E}_{x \sim \mathcal{Q}(\cdot)}[x^2]$. Next, note that since $\mathcal{Q}(\cdot)$ assigns a positive probability to every neighborhood of the origin and has finite higher moments, and since $\sigma(\cdot)$ does not vanish on both sides of the origin, it must hold that

$$\mathbb{E}_{x \sim \mathcal{Q}(\cdot)}[\sigma^2(x)] > 0.$$

The rest of the conditions in Theorem 3 are directly fulfilled by the properties of regular distributions (Definition 2) and the properties of admissible activation functions (Definition 4) that are anti-symmetric. Overall we showed that Theorem 3 applies for $\mathcal{Q}(\cdot)$ and $\sigma(\cdot)$, and so the arbitrary prior case of Theorem 1 will follow from Theorem 1.

### A.2.2 Sufficient Condition for Theorem 3

A useful Lemma used in the proof of Theorem 3 is provided below. The Lemma shows that for an infinitely wide matrix factorization (Equation (4)) with probabilities for low training loss and low generalization loss equal to that of a centered Gaussian matrix (Definition 8), the probability for having low generalization loss (Equation (3)) conditioned on having low training loss (Equation (2)) is equal to the probability of having low generalization loss.

**Lemma 2.** *Let $\epsilon_{\text{gen}}, \epsilon_{\text{train}} > 0$. Let $W_{\text{iid}} \in \mathbb{R}^{m,m'}$ be a centered Gaussian matrix (Definition 8). Suppose that as $k \to 0$, the quantities*

$$\left| \mathcal{P}\left( \mathcal{L}_{\text{train}}(W) < \epsilon_{\text{train}}, \mathcal{L}_{\text{gen}}(W) < \epsilon_{\text{gen}} \right) - \mathcal{P}\left( \mathcal{L}_{\text{train}}(W_{\text{iid}}) < \epsilon_{\text{train}}, \mathcal{L}_{\text{gen}}(W_{\text{iid}}) < \epsilon_{\text{gen}} \right) \right|,$$

$$\left| \mathcal{P}\left( \mathcal{L}_{\text{train}}(W) < \epsilon_{\text{train}} \right) - \mathcal{P}\left( \mathcal{L}_{\text{train}}(W_{\text{iid}}) < \epsilon_{\text{train}} \right) \right|,$$

*and*

$$\left| \mathcal{P}\left( \mathcal{L}_{\text{gen}}(W) < \epsilon_{\text{gen}} \right) - \mathcal{P}\left( \mathcal{L}_{\text{gen}}(W_{\text{iid}}) < \epsilon_{\text{gen}} \right) \right|$$

*all tend to 0. Then*

$$\lim_{k \to \infty} \mathcal{P}\left( \mathcal{L}_{\text{gen}}(W) < \epsilon_{\text{gen}} \Big| \mathcal{L}_{\text{train}}(W) < \epsilon_{\text{train}} \right) = \lim_{k \to \infty} \mathcal{P}\left( \mathcal{L}_{\text{gen}}(W) < \epsilon_{\text{gen}} \right).$$

*Proof.* By the definition of the conditional probability

$$
\mathcal{P}\left(\mathcal{L}_{\text{gen}}(W) < \epsilon_{\text{gen}}\Big|\mathcal{L}_{\text{train}}(W) < \epsilon_{\text{train}}\right) = \frac{\mathcal{P}\left(\mathcal{L}_{\text{train}}(W) < \epsilon_{\text{train}}\,,\mathcal{L}_{\text{gen}}(W) < \epsilon_{\text{gen}}\right)}{\mathcal{P}\left(\mathcal{L}_{\text{train}}(W) < \epsilon_{\text{train}}\right)}.
$$

Observe that $\mathcal{P}\left(\mathcal{L}_{\text{train}}(W_{\text{iid}}) < \epsilon_{\text{train}}\right) > 0$ does not depend on $k$. Therefore, we have that

$$
\begin{aligned}
\lim_{k\to\infty} \mathcal{P}\left(\mathcal{L}_{\text{gen}}(W) < \epsilon_{\text{gen}}\Big|\mathcal{L}_{\text{train}}(W) < \epsilon_{\text{train}}\right) &= \frac{\lim_{k\to\infty}\mathcal{P}\left(\mathcal{L}_{\text{train}}(W) < \epsilon_{\text{train}}\,,\mathcal{L}_{\text{gen}}(W) < \epsilon_{\text{gen}}\right)}{\lim_{k\to\infty}\mathcal{P}\left(\mathcal{L}_{\text{train}}(W) < \epsilon_{\text{train}}\right)} \\
&= \frac{\mathcal{P}\left(\mathcal{L}_{\text{train}}(W_{\text{iid}}) < \epsilon_{\text{train}}\,,\mathcal{L}_{\text{gen}}(W_{\text{iid}}) < \epsilon_{\text{gen}}\right)}{\mathcal{P}\left(\mathcal{L}_{\text{train}}(W_{\text{iid}}) < \epsilon_{\text{train}}\right)} \\
&= \mathcal{P}\left(\mathcal{L}_{\text{gen}}(W_{\text{iid}}) < \epsilon_{\text{gen}}\Big|\mathcal{L}_{\text{train}}(W_{\text{iid}}) < \epsilon_{\text{train}}\right) \\
&= \mathcal{P}\left(\mathcal{L}_{\text{gen}}(W_{\text{iid}}) < \epsilon_{\text{gen}}\right) \\
&= \lim_{k\to\infty}\mathcal{P}\left(\mathcal{L}_{\text{gen}}(W) < \epsilon_{\text{gen}}\right).
\end{aligned}
$$

In the penultimate transition we have used the fact that the measurement matrices $A$ in $\mathcal{B}$ are orthogonal to $A_1, \ldots, A_n$ and thus

$$
\begin{aligned}
&\mathcal{P}\left(\mathcal{L}_{\text{gen}}(W_{\text{iid}}) < \epsilon_{\text{gen}}\Big|\mathcal{L}_{\text{train}}(W_{\text{iid}}) < \epsilon_{\text{train}}\right) \\
&= \mathcal{P}\left(\frac{1}{\mathcal{B}}\sum_{A\in\mathcal{B}}\left(\langle A, W_{\text{iid}}\rangle - \langle A, W^*\rangle\right)^2 < \epsilon_{\text{gen}}\Big|\frac{1}{n}\sum_{i=1}^{n}\left(\langle A_i, W_{\text{iid}}\rangle - y_i\right)^2 < \epsilon_{\text{train}}\right) \\
&= \mathcal{P}\left(\frac{1}{\mathcal{B}}\sum_{A\in\mathcal{B}}\left(\langle A, W_{\text{iid}}\rangle - \langle A, W^*\rangle\right)^2 < \epsilon_{\text{gen}}\right) \\
&= \mathcal{P}\left(\mathcal{L}_{\text{gen}}(W_{\text{iid}}) < \epsilon_{\text{gen}}\right),
\end{aligned}
$$

where the second equality stems from the fact that for any fixed vectors $v_1, \ldots, v_r$ which are orthogonal (each of the flattened matrices $A_1, \ldots, A_n$ and the flattened $A$), and a vector of independent identically distributed zero-centered Gaussian variables $X$ (the flattened $W_{\text{iid}}$), the variables $\{\langle X, v_i\rangle\}_{1\le i\le r}$ are independent. $\qquad\square$

### A.2.3 Convergence in Distribution to a Centered Gaussian Matrix

In this section we prove that in the limit of infinite width, the matrix factorization converges in distribution to a centered Gaussian matrix (Definition 8). Key to the proof is the main result of Hanin [45] which characterizes the convergence of infinitely wide fully connected neural networks to Gaussian processes. We present here a slightly adapted version which is sufficient for our needs.

**Theorem 4** (Theorem 1.2 of [45] (adapted)). *Let $T \subseteq \mathbb{R}^{m'}$ be some compact set. Let $\mathcal{Q}(\cdot)$ be some probability distribution on $\mathbb{R}$ which satisfies*

$$
\mathbb{E}_{x\sim\mathcal{Q}(\cdot)}[x] = 0, \qquad \mathbb{E}_{x\sim\mathcal{Q}(\cdot)}[x^2] = 1
$$

*and has finite higher moments. Suppose that for any $j \in [d]$, the entries of $W_i \in \mathbb{R}^{m_{j+1}, m_j}$ are drawn independently by first sampling $x \sim \mathcal{Q}(\cdot)$ and then setting $[W_j]_{rs} = \sqrt{\frac{c_W}{m_j}}x$. Additionally, suppose that $\sigma$ is absolutely continuous and that its almost-everywhere defined derivative is polynomially bounded:*

$$
\exists p > 0 \text{ s.t. } \forall x \in \mathbb{R} \quad \left\|\frac{\sigma'(x)}{1+|x|^p}\right\|_{L^\infty(\mathbb{R})} < \infty.
$$

*Then as $k \to \infty$, the sequence of stochastic processes $\mathbf{x}_\alpha \in \mathbb{R}^{m'} \mapsto \mathbf{z}_\alpha^{(d)} \in \mathbb{R}^m$ given by a fully connected neural network (Definition 5) set with weights $W_1, \ldots, W_d$ converges weakly in $C^0(T, \mathbb{R}^m)$ to $\Gamma_\alpha^d$, a zero-centered Gaussian process taking values in $\mathbb{R}^m$ with independent identically distributed coordinates. For any $r \in [m]$ and inputs $\mathbf{x}_\alpha, \mathbf{x}_\beta \in T$, the coordinate-wise covariance function*

$$
K_{\alpha\beta}^{(d)} := \text{Cov}\left(\left[\Gamma_\alpha^{(d)}\right]_r, \left[\Gamma_\beta^{(d)}\right]_r\right) = \lim_{k\to\infty}\text{Cov}\left(\left[\mathbf{z}_\alpha^{(d)}\right]_r, \left[\mathbf{z}_\beta^{(d)}\right]_r\right)
$$

*for this limiting process satisfies the following recursive relation:*

$$K_{\alpha\beta}^{(j)} = c_W \, \mathbb{E}[\sigma(z_\alpha)\sigma(z_\beta)], \quad \begin{pmatrix} z_\alpha \\ z_\beta \end{pmatrix} \sim \mathcal{N} \left( 0, \begin{pmatrix} K_{\alpha\alpha}^{(j-1)} & K_{\alpha\beta}^{(i-1)} \\ K_{\alpha\beta}^{(j-1)} & K_{\beta\beta}^{(j-1)} \end{pmatrix} \right)$$

*for $j = 2, \ldots, d$, with the initial condition*

$$K_{\alpha\beta}^{(1)} = c_W \, \mathbb{E} \left[ \sigma \left( \left[ \mathbf{z}_\alpha^{(1)} \right]_1 \right) \sigma \left( \left[ \mathbf{z}_\beta^{(1)} \right]_1 \right) \right],$$

*where the distribution of $\left( \left[ \mathbf{z}_\alpha^{(1)} \right]_1, \left[ \mathbf{z}_\beta^{(1)} \right]_1 \right) = (W_1 \mathbf{x}_\alpha, W_1 \mathbf{x}_\beta)$ is determined by the distribution of the weights $W_1$.*

*Proof.* The above is an adaption of Theorem 1.2 in [45], where the fully connected neural network has no biases. For a full proof see Hanin [45]. □

We now move to the following Proposition arising from Theorem 4, showing that for a symmetric distribution $\mathcal{Q}(\cdot)$ and an anti-symmetric activation function $\sigma$, the random variables corresponding to the network's outputs when the inputs $\mathbf{x}_\alpha, \mathbf{x}_\beta$ are two distinct standard basis vectors, converge in distribution to independent identically distributed zero-centered Gaussian vectors.

**Proposition 2.** *Let $T \subseteq \mathbb{R}^{m'}$ be the unit sphere. Suppose the assumptions of Theorem 4 hold. Suppose also that:*

- *The distribution $\mathcal{Q}(\cdot)$ is symmetric, i.e., if $x \sim \mathcal{Q}(\cdot)$ then $-x \sim \mathcal{Q}(\cdot)$.*

- *The activation function $\sigma$ is not constant and anti-symmetric, i.e.,*
$$\forall x \in \mathbb{R}. \ \ \sigma(x) = -\sigma(-x).$$

- *It holds that*
$$\mathbb{E}_{x \sim \mathcal{Q}(\cdot)} \left[ \sigma^2(x) \right] > 0.$$

*Let $\alpha, \beta \in [m']$ be two distinct indices. Denote $\mathbf{e}_\alpha \in \mathbb{R}^{m'}$ the standard basis vector holding $1$ in its $\alpha$ coordinate and zeros in the rest. Denote $\mathbf{e}_\beta$ similarly. Then as $k \to \infty$ the random output vectors $\mathbf{z}_\alpha^{(d)}$ and $\mathbf{z}_\beta^{(d)}$ corresponding to $\mathbf{e}_\alpha$ and $\mathbf{e}_\beta$ respectively converge in distribution to two independent random vectors each with independent entries drawn from the same zero-centered Gaussian distribution.*

*Proof.* Per Theorem 4, as $k \to \infty$ the variables $\mathbf{z}_\alpha^{(d)}$ and $\mathbf{z}_\beta^{(d)}$ converge in distribution to zero-centered Gaussian vectors where for any distinct indices $r, r' \in [m]$:

- The entries $\left[ \mathbf{z}_\alpha^{(d)} \right]_r, \left[ \mathbf{z}_\alpha^{(d)} \right]_{r'}$ are independent.

- The entries $\left[ \mathbf{z}_\beta^{(d)} \right]_r, \left[ \mathbf{z}_\beta^{(d)} \right]_{r'}$ are independent.

- The entries $\left[ \mathbf{z}_\alpha^{(d)} \right]_r, \left[ \mathbf{z}_\beta^{(d)} \right]_{r'}$ are independent.

Next, using the notation of Theorem 4, we prove via induction on $d$ that $K_{\alpha\beta}^{(d)} = 0$, $K_{\alpha\alpha}^{(d)} = K_{\beta\beta}^{(d)}$ and that $K_{\alpha\alpha}^{(d)}$ is finite and positive. First, for the base case, note that we have

$$\begin{aligned} K_{\alpha\beta}^{(1)} &= c_W \, \mathbb{E} \left[ \sigma \left( \left[ \mathbf{z}_\alpha^{(1)} \right]_1 \right) \sigma \left( \left[ \mathbf{z}_\beta^{(1)} \right]_1 \right) \right] \\ &= c_W \, \mathbb{E} \left[ \sigma \left( [W_1 \mathbf{e}_\alpha]_1 \right) \sigma \left( [W_1 \mathbf{e}_\beta]_1 \right) \right] \\ &= c_W \, \mathbb{E} \left[ \sigma \left( [W_1]_{1,\alpha} \right) \sigma \left( [W_1]_{1,\beta} \right) \right] \\ &= c_W \, \mathbb{E} \left[ \sigma \left( [W_1]_{1,\alpha} \right) \right] \mathbb{E} \left[ \sigma \left( [W_1]_{1,\beta} \right) \right], \end{aligned}$$

where the ultimate transition is due to the independence of $[W_1]_{1,\alpha}$ and $[W_1]_{1,\beta}$. Next, since $\mathcal{Q}(\cdot)$ is symmetric and $\sigma$ is anti-symmetric, we obtain by Lemma 20 that

$$\mathbb{E}\left[\sigma\left([W_1]_{1,\alpha}\right)\right] = \mathbb{E}\left[\sigma\left([W_1]_{1,\beta}\right)\right] = 0\,.$$

Overall, we obtain that

$$K_{\alpha\beta}^{(1)} = c_W \cdot 0 \cdot 0 = 0\,.$$

Additionally, since $[W_1]_{1,\alpha}$ and $[W_1]_{1,\beta}$ are both drawn from $\mathcal{Q}(\cdot)$, we obtain that

$$\begin{aligned}
K_{\alpha\alpha}^{(1)} &= c_W\, \mathbb{E}\left[\sigma\left(\left[\mathbf{z}_\alpha^{(1)}\right]_1\right)\sigma\left(\left[\mathbf{z}_\alpha^{(1)}\right]_1\right)\right]\\
&= c_W\, \mathbb{E}\left[\sigma\left([W_1]_{1,\alpha}\right)\sigma\left([W_1]_{1,\alpha}\right)\right]\\
&= c_W\, \mathbb{E}\left[\sigma\left([W_1]_{1,\beta}\right)\sigma\left([W_1]_{1,\beta}\right)\right]\\
&= K_{\beta\beta}^{(1)}\,.
\end{aligned}$$

Finally, by our assumption we have that

$$K_{\alpha\alpha}^{(1)} = c_W\, \mathbb{E}\left[\sigma\left([W_1]_{1,\alpha}\right)\sigma\left([W_1]_{1,\alpha}\right)\right] = c_W \mathop{\mathbb{E}}_{x\sim\mathcal{Q}(\cdot)}[\sigma^2(x)] > 0\,.$$

as required. Next, fix $j \in [d]$ and assume that $K_{\alpha\beta}^{(j)} = 0$, $K_{\alpha\alpha}^{(j)} = K_{\beta\beta}^{(j)}$ and that $K_{\alpha\alpha}^{(j)}$ is finite and positive. Hence, plugging the inductive assumption into Theorem 4, we obtain that

$$K_{\alpha\beta}^{(j+1)} = c_W\, \mathbb{E}\left[\sigma(z_\alpha)\sigma(z_\beta)\right]$$

where

$$\begin{pmatrix} z_\alpha \\ z_\beta \end{pmatrix} \sim \mathcal{N}\left(0, \begin{pmatrix} K_{\alpha\alpha}^{(j)} & K_{\alpha\beta}^{(j)} \\ K_{\alpha\beta}^{(j)} & K_{\beta\beta}^{(j)} \end{pmatrix}\right) = \mathcal{N}\left(0, \begin{pmatrix} K_{\alpha\alpha}^{(j)} & 0 \\ 0 & K_{\alpha\alpha}^{(j)} \end{pmatrix}\right)\,.$$

Therefore, $z_\alpha$ and $z_\beta$ are independent identically distributed zero-centered Gaussian variables. Hence, we obtain that

$$K_{\alpha\beta}^{(j+1)} = c_W\, \mathbb{E}\left[\sigma(z_\alpha)\right]\mathbb{E}\left[\sigma(z_\beta)\right] = c_W \cdot 0 \cdot 0 = 0\,,$$

where the penultimate transition is due to Lemma 20. Additionally,

$$K_{\alpha\alpha}^{(j+1)} = c_W\, \mathbb{E}\left[\sigma(z_\alpha)\sigma(z_\alpha)\right] = c_W\, \mathbb{E}\left[\sigma(z_\beta)\sigma(z_\beta)\right] = K_{\beta\beta}^{(j+1)}\,.$$

Finally, we have by our inductive assumption that $K_{\alpha\alpha}^{(j)}$ is finite and positive, thus the non-constant random variable $z_\alpha \sim \mathcal{N}(0, K_{\alpha\alpha}^{(j)})$ has finite moments. Therefore, since $\sigma$ has a polynomially bounded derivative almost-everywhere and it is not constant, it holds that

$$K_{\alpha\alpha}^{(j+1)} = c_W\, \mathbb{E}\left[\sigma(z_\alpha)\sigma(z_\alpha)\right] > 0$$

as required. Thus by Theorem 4, for any $j \in [m]$, the entries $\left[\mathbf{z}_\alpha^{(d)}\right]_j$ and $\left[\mathbf{z}_\beta^{(d)}\right]_j$ converge in distribution to two independent identically distributed zero-centered Gaussian variables as $k \to \infty$. Overall we have shown that as $k \to \infty$, the random vectors $\mathbf{z}_\alpha^{(d)}$ and $\mathbf{z}_\beta^{(d)}$ converge to two independent random vectors each with independent entries drawn from the same zero-centered Gaussian distribution, completing the proof. $\qquad\square$

The last two arguments imply the following important Corollary, which states that as $k \to \infty$, the matrix factorization $W$ converges in distribution to a centered Gaussian matrix (Definition 8).

**Corollary 1.** *As $k \to \infty$, the matrix factorization $W$ converges in distribution to the random matrix $W_{\text{iid}} \in \mathbb{R}^{m,m'}$ whose entries are drawn independently from the same zero-centered Gaussian distribution.*

*Proof.* Per Proposition 2, as $k \to \infty$, the random output vectors $\mathbf{z}_1^{(d)}, \ldots, \mathbf{z}_{m'}^{(d)}$ corresponding to the inputs $\mathbf{e}_1, \ldots, \mathbf{e}_{m'}$ converge in distribution to independent random vectors each with independent entries drawn from the same zero-centered Gaussian distribution. Therefore, as $k \to \infty$, the random matrix

$$\begin{pmatrix} \mathbf{z}_1^{(d)} & \cdots & \mathbf{z}_{m'}^{(d)} \end{pmatrix} \in \mathbb{R}^{m,m'}$$

converges in distribution to the random matrix $W_{\text{iid}} \in \mathbb{R}^{m,m'}$ whose entries are drawn independently from the same zero-centered Gaussian distribution. The claim follows by Lemma 1 which states that the above matrix is equal to $W$. $\qquad\square$

### A.2.4 Convergence in Distribution Implies Sufficient Condition

In the previous section, Corollary 1 showed that $W$ converges in distribution to a random matrix with independent entries drawn from the same zero-centered Gaussian distribution. In this section, we use basic tools from probability theory in order to show that this convergence in fact implies the quantities in Lemma 2 converge, completing the proof of Theorem 1. We begin by introducing the concept of continuity sets:

**Definition 6.** Let $X$ be some random variable on the space $\Omega$. A set $A \subseteq \Omega$ is a *continuity set* of $X$ when

$$\mathcal{P}(X \in \partial A) = 0$$

where $\partial A$ is the boundary of $A$.

The main tool we employ in this part of the proof is Portmanteau's Theorem, which states that convergence in distribution implies convergence in the probability of any continuty set:

**Theorem 5.** *Let $\{X_k\}_{k=1}^{\infty}$ be a series of random variables on the same space $\Omega$. Let $X$ be a random variable on the space $\Omega$. If*

$$X_k \xrightarrow[k\to\infty]{dist.} X$$

*then for any continuity set $A$ of $X$ (Definition 6) it holds that*

$$\lim_{k\to\infty} \mathcal{P}(X_k \in A) = \mathcal{P}(X \in A)$$

*Proof.* See Duchi [33]. $\qquad\square$

In order to invoke Theorem 5, we continue to showing that the sets in question are all continuity sets of $W_{\text{iid}}$. We begin by showing that the set with low training error and the set with low generalization error are both continuity sets of $W_{\text{iid}}$.

**Proposition 3.** *The sets*

$$S_{gen} := \{W \in \mathbb{R}^{m,m'} : \mathcal{L}_{gen}(W) < \epsilon_{\text{gen}}\}, \quad S_{train} := \{W \in \mathbb{R}^{m,m'} : \mathcal{L}_{train}(W) < \epsilon_{\text{train}}\}$$

*are continuity sets of $W_{\text{iid}}$ (Definition 6).*

*Proof.* Consider the first set (the proof is identical for the second). The boundary of the set is of the form

$$\{W \in \mathbb{R}^{m,m'} : \mathcal{L}_{\text{gen}}(W) - \epsilon_{\text{gen}} = 0\}$$

Since $\mathcal{L}_{\text{gen}}(W)$ is a polynomial in the entries of $W$ and $\mathcal{L}_{\text{gen}}(W^*) - \epsilon_{\text{gen}} \neq 0$, the polynomial $\mathcal{L}_{\text{gen}}(W) - \epsilon_{\text{gen}}$ is not the zero polynomial. Therefore by Lemma 23 the boundary has Lebesgue measure zero. Per Corollary 1, $W_{\text{iid}}$ has a continuous distribution over $\mathbb{R}^{m,m'}$ and thus it must hold that

$$\mathcal{P}\left(W_{\text{iid}} \in \{W \in \mathbb{R}^{m,m'} : \mathcal{L}_{\text{gen}}(W) - \epsilon_{\text{gen}} = 0\}\right) = 0,$$

*i.e.*, the set is a continuity set. $\qquad\square$

The next Lemma shows that the intersection of two continuity sets is also a continuity set, hence Proposition 3 implies that $S_{gen} \cap S_{train}$ is also a continuity set.

**Lemma 3.** *Let $X$ be a random variable over the space $\Omega$. Let $A, B \subseteq \Omega$ be continuity sets of $X$ (Definition 6). Then the set $A \cap B$ is a continuity set of $X$.*

*Proof.* Per Definition 6 it holds that

$$\mathcal{P}(X \in \partial A) = 0, \quad \mathcal{P}(X \in \partial B) = 0$$

and so

$$\mathcal{P}(X \in \partial A \cup \partial B) = 0\,.$$

Hence, the proof follows if

$$\partial(A \cap B) \subseteq \partial A \cup \partial B\,.$$

First, recall that for any $X \subseteq \Omega$

$$\partial X = \overline{X} \cap \overline{C_\Omega(X)}\,,$$

where $\overline{X}$ is the closure of $X$ and $C_\Omega(\cdot)$ is the complement operator. Next, we have that

$$\overline{(A \cap B)} \subseteq \overline{A}, \quad \overline{(A \cap B)} \subseteq \overline{B}\,.$$

Finally, it holds that

$$\overline{C_\Omega(A \cap B)} = \overline{C_\Omega(A) \cup C_\Omega(B)} = \overline{C_\Omega(A)} \cup \overline{C_\Omega(B)}\,,$$

therefore,

$$
\begin{aligned}
\partial(A \cap B) &= \overline{(A \cap B)} \cap \overline{C_\Omega(A \cap B)} \\
&= \overline{(A \cap B)} \cap \left( \overline{C_\Omega(A)} \cup \overline{C_\Omega(B)} \right) \\
&= \left( \overline{(A \cap B)} \cap \overline{C_\Omega(A)} \right) \cup \left( \overline{(A \cap B)} \cap \overline{C_\Omega(B)} \right) \\
&\subseteq \left( \overline{A} \cap \overline{C_\Omega(A)} \right) \cup \left( \overline{B} \cap \overline{C_\Omega(B)} \right) \\
&= \partial A \cup \partial B
\end{aligned}
$$

as required. $\qquad \square$

Overall, we have shown that Corollary 1 implies together with Theorem 5 and Proposition 3 that

$$\lim_{k \to \infty} |\mathcal{P}\left(\mathcal{L}_{\text{gen}}(W) < \epsilon_{\text{gen}}\right\} \cap \{\mathcal{L}_{\text{train}}(W) < \epsilon_{\text{train}}\}) - \mathcal{P}\left(\mathcal{L}_{\text{gen}}(W_{\text{iid}}) < \epsilon_{\text{gen}}\right\} \cap \{\mathcal{L}_{\text{train}}(W_{\text{iid}}) < \epsilon_{\text{train}}\})| = 0\,,$$

$$\lim_{k \to \infty} |\mathcal{P}\left(\{\mathcal{L}_{\text{train}}(W) < \epsilon_{\text{train}}\}\right) - \mathcal{P}\left(\{\mathcal{L}_{\text{train}}(W_{\text{iid}}) < \epsilon_{\text{train}}\}\right)| = 0\,,$$

and

$$\lim_{k \to \infty} |\mathcal{P}\left(\mathcal{L}_{\text{gen}}(W) < \epsilon_{\text{gen}}\right\}) - \mathcal{P}\left(\mathcal{L}_{\text{gen}}(W_{\text{iid}}) < \epsilon_{\text{gen}}\right\})| = 0\,.$$

Hence, the proof follows by invoking Lemma 2 which implies Theorem 1.

### A.3 Proof for Gaussian Distribution and Finite Width

The outline of the proof is as follows; Appendix A.3.1 presents Theorem 6, of which the canonical case of Theorem 1 is a special case. Appendix A.3.2 provides a useful Lemma used in the proof. Finally, Appendix A.3.3 adapts a result from Favaro et al. [36] showing that a matrix factorization with Gaussian weights has a bounded convex distance from a centered Gaussian matrix (Definition 8) and arguing that the latter bound implies the conditions required for the Lemma in Appendix A.3.2.

### A.3.1 Restatement of the Canonical Case of Theorem 1

The canonical case of Theorem 1 follows from Theorem 6, which allows for the activation $\sigma(\cdot)$ to be slightly more general. Theorem 6 is presented below; afterwards, we demonstrate how it implies the canonical case of Theorem 1.

**Theorem 6.** *Let $d \in \mathbb{N}$ be a fixed depth. Let $\mathcal{N}(\cdot)$ be the standard Gaussian distribution, i.e., $\mathcal{N}(\cdot) := \mathcal{N}(\cdot; 0, 1)$. Let $\sigma(\cdot)$ be an activation function that is not constant and anti-symmetric, i.e.,*

$$\forall x \in \mathbb{R}. \ \sigma(x) = -\sigma(-x),$$

*furthermore suppose that $\sigma$ is absolutely continuous, and that its almost-everywhere defined derivative is polynomially bounded, i.e:*

$$\exists p > 0 \ s.t. \ \forall x \in \mathbb{R} \ \left\| \frac{\sigma'(x)}{1 + |x|^p} \right\|_{L^\infty(\mathbb{R})} < \infty.$$

*Suppose also that*

$$\underset{x \sim \mathcal{N}(\cdot)}{\mathbb{E}} \left[ \sigma^2(x) \right] > 0.$$

*Let $\epsilon_{\text{gen}}, \epsilon_{\text{train}}, c_W > 0$. Suppose that for any $j \in [d]$, the entries of $W_j \in \mathbb{R}^{m_{j+1}, m_j}$ are drawn independently by first sampling $x \sim \mathcal{N}(\cdot)$ and then setting $[W_j]_{rs} = \sqrt{\frac{c_W}{m_j}} x$. Then there exists a constant $c > 0$ dependent on $m, m', d, \sigma, c_W, n, \epsilon_{\text{train}}$ and $\epsilon_{\text{gen}}$, and a constant $k_0 \in \mathbb{N}$ dependent on $c$ and $\mathcal{P}(\mathcal{L}_{\text{train}}(W_{\text{iid}}) < \epsilon_{\text{train}})$, such that for any $k \geq k_0$ the matrix factorization $W$ satisfies*

$$\mathcal{P}\left( \mathcal{L}_{gen}(W) < \epsilon_{\text{gen}} \Big| \mathcal{L}_{train}(W) < \epsilon_{\text{train}} \right) - \mathcal{P}\left( \mathcal{L}_{gen}(W) < \epsilon_{\text{gen}} \right) \leq \frac{\frac{2c}{\sqrt{k}}}{\mathcal{P}(\mathcal{L}_{\text{train}}(W_{\text{iid}}) < \epsilon_{\text{train}}) - \frac{c}{\sqrt{k}}}.$$

Note that the above bound is of order $\frac{1}{\sqrt{k}}$.

**Remark 1.** *For any $k \geq k_0$ it holds that*

$$\frac{\frac{2c}{\sqrt{k}}}{\mathcal{P}(\mathcal{L}_{\text{train}}(W_{\text{iid}}) < \epsilon_{\text{train}}) - \frac{c}{\sqrt{k}}} \cdot \sqrt{k} = \frac{2c}{\mathcal{P}(\mathcal{L}_{\text{train}}(W_{\text{iid}}) < \epsilon_{\text{train}}) - \frac{c}{\sqrt{k}}} = \Omega(1),$$

*hence*

$$\frac{\frac{2c}{\sqrt{k}}}{\mathcal{P}(\mathcal{L}_{\text{train}}(W_{\text{iid}}) < \epsilon_{\text{train}}) - \frac{c}{\sqrt{k}}} = O\left( \frac{1}{\sqrt{k}} \right).$$

Let $\mathcal{N}(\cdot; 0, \nu)$ be a zero-centered Gaussian distribution, and let $\sigma(\cdot)$ be an admissible activation function (Definition 4) that is anti-symmetric. First, observe that since Theorem 6 allows for arbitrary $c_W > 0$, one may view $\mathcal{N}(\cdot; 0, \nu)$ as the standard Gaussian distribution $\mathcal{N}(\cdot)$ scaled by $\sqrt{\nu}$. Next, note that since $\mathcal{N}(\cdot)$ assigns a positive probability to every neighborhood of the origin and has finite higher moments, and since $\sigma(\cdot)$ does not vanish on both sides of the origin, it must hold that

$$\underset{x \sim \mathcal{N}(\cdot)}{\mathbb{E}} \left[ \sigma^2(x) \right] > 0.$$

The rest of the conditions in Theorem 6 are directly fulfilled by the properties of admissible activation functions (Definition 4) that are anti-symmetric. Overall we showed that Theorem 6 applies for $\mathcal{N}(\cdot; 0, \nu)$ and $\sigma(\cdot)$, and so it suffices to prove Theorem 6.

### A.3.2 Sufficient Condition for Theorem 6

A useful Lemma used in the proof of Theorem 6 is provided below. Before presenting the Lemma, we define the convex distance between random variables, and prove that the sets of matrices with either low training error or low generalization error are convex.

**Definition 7.** *Let $m \in \mathbb{N}$ and let $X$ and $Y$ be two $m$-dimensional random variables. The convex distance between $X$ and $Y$ is defined as*

$$d_c(X, Y) := \sup_B |\mathcal{P}(X \in B) - \mathcal{P}(Y \in B)|,$$

where the supremum runs over all convex $B \subset \mathbb{R}^m$.

**Remark 2.** *The convex distance between two random matrices is naturally defined as the convex distance between their corresponding flattened vector representations.*

**Lemma 4.** *Let $\epsilon_{\text{gen}}, \epsilon_{\text{train}} > 0$. The sets*

$$S_{train} := \left\{ W \in \mathbb{R}^{m,m'} : \mathcal{L}_{\text{train}}(W) < \epsilon_{\text{train}} \right\}, \quad S_{gen} := \left\{ W \in \mathbb{R}^{m,m'} : \mathcal{L}_{\text{gen}}(W) < \epsilon_{\text{gen}} \right\}$$

*are convex.*

*Proof.* We prove that $S_{gen}$ is convex (one can prove the same claim about $S_{train}$ using identical arguments). To do this, it suffices to show that $\mathcal{L}_{\text{train}}(W)$ is a convex function. Because sums of convex functions are convex, it suffices to show that the function corresponding to a single test matrix, namely

$$(\langle A, W \rangle - \langle A, W^* \rangle)^2$$

for some $A \in \mathcal{B}$ is convex, and this is the case because it is the composition of an affine function with the convex function $x \to x^2$. $\qquad\square$

**Remark 3.** *The intersection of two convex sets is convex, thus the following set is also convex*

$$\left\{ W \in \mathbb{R}^{m,m'} : \mathcal{L}_{\text{train}}(W) < \epsilon_{\text{train}}, \mathcal{L}_{\text{gen}}(W) < \epsilon_{\text{gen}} \right\}.$$

We are now ready to present the Lemma. The Lemma show that if the convex distance between the matrix factorization (Equation (4)) and a centered Gaussian matrix (Definition 8) is $O\left(\frac{1}{\sqrt{k}}\right)$, then for any large enough $k$ the probability for having low generalization loss (Equation (3)) conditioned on having low training loss (Equation (2)) is no more than order $O\left(\frac{1}{\sqrt{k}}\right)$ larger than the prior probability of having low generalization loss.

**Lemma 5.** *Let $\epsilon_{\text{gen}}, \epsilon_{\text{train}} > 0$. Let $W_{\text{iid}} \in \mathbb{R}^{m,m'}$ be a centered Gaussian matrix (Definition 8). Suppose that there exists some $c > 0$ such that the convex distance between $W$ and $W_{\text{iid}}$ (Definition 7) satisfies*

$$d_c(W, W_{\text{iid}}) \leq \frac{c}{\sqrt{k}}.$$

*Then there exists some $k_0 \in \mathbb{N}$ dependent on $c$ and $\mathcal{P}(\mathcal{L}_{\text{train}}(W_{\text{iid}}) < \epsilon_{\text{train}})$ such that for any $k \geq k_0$ it holds that*

$$\mathcal{P}\left(\mathcal{L}_{\text{gen}}(W) < \epsilon_{\text{gen}} \Big| \mathcal{L}_{\text{train}}(W) < \epsilon_{\text{train}}\right) \leq \mathcal{P}(\mathcal{L}_{\text{gen}}(W) < \epsilon_{\text{gen}}) + \frac{\frac{2c}{\sqrt{k}}}{\mathcal{P}(\mathcal{L}_{\text{train}}(W_{\text{iid}}) < \epsilon_{\text{train}}) - \frac{c}{\sqrt{k}}}.$$

*Proof.* Per Definition 7, Lemma 4, , and Remark 3, the fact that $d_c(W, W_{\text{iid}}) \leq \frac{c}{\sqrt{k}}$ implies that

$$|\mathcal{P}(\mathcal{L}_{\text{train}}(W) < \epsilon_{\text{train}}) - \mathcal{P}(\mathcal{L}_{\text{train}}(W_{\text{iid}}) < \epsilon_{\text{train}})| \leq \frac{c}{\sqrt{k}},$$

$$|\mathcal{P}(\mathcal{L}_{\text{gen}}(W) < \epsilon_{\text{gen}}) - \mathcal{P}(\mathcal{L}_{\text{gen}}(W_{\text{iid}}) < \epsilon_{\text{gen}})| \leq \frac{c}{\sqrt{k}},$$

and

$$|\mathcal{P}(\mathcal{L}_{\text{train}}(W) < \epsilon_{\text{train}}, \mathcal{L}_{\text{gen}}(W) < \epsilon_{\text{gen}}) - \mathcal{P}(\mathcal{L}_{\text{train}}(W_{\text{iid}}) < \epsilon_{\text{train}}, \mathcal{L}_{\text{gen}}(W_{\text{iid}}) < \epsilon_{\text{gen}})| \leq \frac{c}{\sqrt{k}}.$$

By the definition of the conditional probability we have that

$$\mathcal{P}(\mathcal{L}_{\text{gen}}(W) < \epsilon_{\text{gen}} \Big| \mathcal{L}_{\text{train}}(W) < \epsilon_{\text{train}}) = \frac{\mathcal{P}(\mathcal{L}_{\text{train}}(W) < \epsilon_{\text{train}}, \mathcal{L}_{\text{gen}}(W) < \epsilon_{\text{gen}})}{\mathcal{P}(\mathcal{L}_{\text{train}}(W) < \epsilon_{\text{train}})}.$$

Since $W_{\text{iid}}$ is a centered Gaussian matrix (Definition 8), it holds that $\mathcal{P}(\mathcal{L}_{\text{train}}(W_{\text{iid}}) < \epsilon_{\text{train}}) > 0$ and so for any

$$k \geq \left(\frac{c}{\mathcal{P}(\mathcal{L}_{\text{train}}(W_{\text{iid}}) < \epsilon_{\text{train}})}\right)^2 =: k_0$$

it holds that $\mathcal{P}(\mathcal{L}_{\text{train}}(W_{\text{iid}}) < \epsilon_{\text{train}}) - \frac{c}{\sqrt{k}} > 0$. Therefore, for any such $k$ the above is bound by

$$\mathcal{P}(\mathcal{L}_{\text{gen}}(W) < \epsilon_{\text{gen}}\big|\mathcal{L}_{\text{train}}(W) < \epsilon_{\text{train}})$$

$$\leq \frac{\mathcal{P}(\mathcal{L}_{\text{train}}(W_{\text{iid}}) < \epsilon_{\text{train}}\,,\mathcal{L}_{\text{gen}}(W_{\text{iid}}) < \epsilon_{\text{gen}}) + \frac{c}{\sqrt{k}}}{\mathcal{P}(\mathcal{L}_{\text{train}}(W_{\text{iid}}) < \epsilon_{\text{train}}) - \frac{c}{\sqrt{k}}}$$

$$= \frac{\mathcal{P}(\mathcal{L}_{\text{train}}(W_{\text{iid}}) < \epsilon_{\text{train}}) \cdot \mathcal{P}(\mathcal{L}_{\text{gen}}(W_{\text{iid}}) < \epsilon_{\text{gen}}) + \frac{c}{\sqrt{k}}}{\mathcal{P}(\mathcal{L}_{\text{train}}(W_{\text{iid}}) < \epsilon_{\text{train}}) - \frac{c}{\sqrt{k}}}$$

$$\leq \frac{\mathcal{P}(\mathcal{L}_{\text{train}}(W_{\text{iid}}) < \epsilon_{\text{train}}) \cdot \left(\mathcal{P}(\mathcal{L}_{\text{gen}}(W) < \epsilon_{\text{gen}}) + \frac{c}{\sqrt{k}}\right) + \frac{c}{\sqrt{k}}}{\mathcal{P}(\mathcal{L}_{\text{train}}(W_{\text{iid}}) < \epsilon_{\text{train}}) - \frac{c}{\sqrt{k}}}$$

$$= \mathcal{P}(\mathcal{L}_{\text{gen}}(W) < \epsilon_{\text{gen}}) \cdot \frac{\mathcal{P}(\mathcal{L}_{\text{train}}(W_{\text{iid}}) < \epsilon_{\text{train}})}{\mathcal{P}(\mathcal{L}_{\text{train}}(W_{\text{iid}}) < \epsilon_{\text{train}}) - \frac{c}{\sqrt{k}}} + \frac{\mathcal{P}(\mathcal{L}_{\text{train}}(W_{\text{iid}}) < \epsilon_{\text{train}}) \cdot \frac{c}{\sqrt{k}} + \frac{c}{\sqrt{k}}}{\mathcal{P}(\mathcal{L}_{\text{train}}(W_{\text{iid}}) < \epsilon_{\text{train}}) - \frac{c}{\sqrt{k}}}$$

$$\leq \mathcal{P}(\mathcal{L}_{\text{gen}}(W) < \epsilon_{\text{gen}}) + \frac{\frac{2c}{\sqrt{k}}}{\mathcal{P}(\mathcal{L}_{\text{train}}(W_{\text{iid}}) < \epsilon_{\text{train}}) - \frac{c}{\sqrt{k}}}\,.$$

In the third transition we have used the fact that the measurement matrices $A$ in $\mathcal{B}$ are orthogonal to $A_1, \ldots, A_n$ and thus

$$\mathcal{P}\left(\mathcal{L}_{\text{gen}}(W_{\text{iid}}) < \epsilon_{\text{gen}}\,,\mathcal{L}_{\text{train}}(W_{\text{iid}}) < \epsilon_{\text{train}}\right)$$

$$= \mathcal{P}\left(\frac{1}{\mathcal{B}}\sum_{A\in\mathcal{B}}(\langle A, W_{\text{iid}}\rangle - \langle A, W^*\rangle)^2 < \epsilon_{\text{gen}}\,,\frac{1}{n}\sum_{i=1}^{n}(\langle A_i, W_{\text{iid}}\rangle - y_i)^2 < \epsilon_{\text{train}}\right)$$

$$= \mathcal{P}\left(\frac{1}{\mathcal{B}}\sum_{A\in\mathcal{B}}(\langle A, W_{\text{iid}}\rangle - \langle A, W^*\rangle)^2 < \epsilon_{\text{gen}}\right) \cdot \mathcal{P}\left(\frac{1}{n}\sum_{i=1}^{n}(\langle A_i, W_{\text{iid}}\rangle - y_i)^2 < \epsilon_{\text{train}}\right)$$

$$= \mathcal{P}\left(\mathcal{L}_{\text{gen}}(W_{\text{iid}}) < \epsilon_{\text{gen}}\right) \cdot \mathcal{P}\left(\mathcal{L}_{\text{train}}(W_{\text{iid}} < \epsilon_{\text{train}})\right),$$

where the second equality stems from the fact that for any fixed vectors $v_1, \ldots, v_r$ which are orthogonal (each of the flattened matrices $A_1, \ldots, A_n$ and the flattened $A$), and a vector of independent identically distributed zero-centered Gaussians $X$ (the flattened $W_{\text{iid}}$), the variables $\{\langle X, v_i\rangle\}_{1\leq i\leq r}$ are independent. $\qquad\square$

### A.3.3 Bound on Convex Distance from a Centered Gaussian Matrix

In this section we prove that for any width $k$, the matrix factorization has a bounded convex distance from a centered Gaussian matrix (Definition 8). Key to the proof is a result of Favaro et al. [36] which provides a bound on the convex distance a fully connected neural network has from a Gaussian process. We present a softer adaption of it sufficient for our needs.

**Theorem 7** (Theorem 3.6 of [36] (adapted)). *Let $\mathcal{N}(\cdot)$ be the standard Gaussian distribution. Suppose that for any $j \in [d]$, the entries of $W_j \in \mathbb{R}^{m_{j+1}, m_j}$ are drawn independently by first sampling $x \sim \mathcal{N}(\cdot)$ and then setting $[W_j]_{rs} = \sqrt{\frac{c_W}{m_j}}x$. Additionally, suppose that $\sigma$ is absolutely continuous and that its almost-everywhere defined derivative is polynomially bounded:*

$$\exists p > 0 \text{ s.t. } \forall x \in \mathbb{R} \quad \left\|\frac{\sigma'(x)}{1 + |x|^p}\right\|_{L^\infty(\mathbb{R})} < \infty\,.$$

*For any $j = 2, \ldots, d$ denote the matrix $K^{(j)} \in \mathbb{R}^{m', m'}$ by*

$$\forall \alpha, \beta \in [m']. \ K_{\alpha\beta}^{(j)} = c_W \, \mathbb{E}[\sigma(z_\alpha)\sigma(z_\beta)], \quad \begin{pmatrix}z_\alpha\\z_\beta\end{pmatrix} \sim \mathcal{N}\left(0, \begin{pmatrix}K_{\alpha\alpha}^{(j-1)} & K_{\alpha\beta}^{(j-1)}\\K_{\alpha\beta}^{(j-1)} & K_{\beta\beta}^{(j-1)}\end{pmatrix}\right)$$

*with the initial condition*

$$\forall \alpha, \beta \in [m']. \ K_{\alpha\beta}^{(1)} = c_W \, \mathbb{E}\left[\sigma\left([W_1\mathbf{e}_\alpha]_1\right)\sigma\left([W_1\mathbf{e}_\beta]_1\right)\right]$$

where for $\alpha \in [m']$, the vector $\mathbf{e}_\alpha \in \mathbb{R}^{m'}$ is the standard basis vector holding 1 in its $\alpha$ coordinate and zeros in the rest. Additionally, for $\alpha \in [m']$ denote by $\mathbf{z}_\alpha^{(d)}$ the output given by a fully connected neural network (Definition 5) set with weights $W_1, \ldots, W_d$ for the input $\mathbf{e}_\alpha$. Lastly, for $\alpha \in [m']$ denote by $\Gamma_\alpha^{(d)}$ a $m$-dimensional zero-centered Gaussian vector with

$$\forall \alpha, \beta \in [m'], r, r' \in [m]. \ \mathrm{Cov}\left(\left[\Gamma_\alpha^{(d)}\right]_r, \left[\Gamma_\beta^{(d)}\right]_{r'}\right) = \mathbf{1}_{r=r'} K_{\alpha\beta}^{(d)}.$$

If for any $j \in [d]$ the matrix $K^{(j)}$ is invertible, then there exists $c > 0$ dependent on $m, m', d, \sigma, c_W, n, \epsilon_{\text{train}}$ and $\epsilon_{\text{gen}}$ such that for any $k \in \mathbb{N}$ it holds that

$$d_c\left(\left(\mathbf{z}_\alpha^{(d)}\right)_{\alpha \in [m']}, \left(\Gamma_\alpha^{(d)}\right)_{\alpha \in [m']}\right) \leq \frac{c}{\sqrt{k}},$$

where we have implicitly regarded $\left(\mathbf{z}_\alpha^{(d)}\right)_{\alpha \in [m']}$ and $\left(\Gamma_\alpha^{(d)}\right)_{\alpha \in [m']}$ as $m' \cdot m$-dimensional random vectors.

*Proof.* The above is an adaption of case (1) of Theorem 3.6 in [36], where the fully connected neural network has no biases, the partial derivatives in question are all of order zero and the finite collection of distinct non-zero network inputs is $\{\mathbf{e}_\alpha\}_{\alpha=1}^{m'}$. For a full proof see Favaro et al. [36]. $\square$

We move forward to the following Lemma, showing that for an anti-symmetric activation function $\sigma$ the covariance matrices $K^{(j)}$ are not only invertible but also a positive multiple of the identity.

**Lemma 6.** *Suppose the assumptions of Theorem 7 hold. Suppose also that*

- *The activation function $\sigma$ is not constant and anti-symmetric, i.e.,*

$$\forall x \in \mathbb{R}. \ \sigma(x) = -\sigma(-x).$$

- *It holds that*

$$\mathbb{E}_{x \sim \mathcal{N}(\cdot)}\left[\sigma^2(x)\right] > 0.$$

*Then for any $j \in [d]$ there exists a positive constant $b^{(j)} > 0$ such that $K^{(j)} = b^{(j)} I_{m'}$*

*Proof.* The proof is extremely similar to that of Proposition 2. We prove via induction on $d$ that $K_{\alpha\beta}^{(d)} = 0$, $K_{\alpha\alpha}^{(d)} = K_{\beta\beta}^{(d)}$ and that $K_{\alpha\alpha}^{(d)}$ is finite and positive. First, for the base case, note that we have

$$
\begin{aligned}
K_{\alpha\beta}^{(1)} &= c_W \, \mathbb{E}\left[\sigma\left([W_1 \mathbf{e}_\alpha]_1\right) \sigma\left([W_1 \mathbf{e}_\beta]_1\right)\right] \\
&= c_W \, \mathbb{E}\left[\sigma\left([W_1]_{1,\alpha}\right) \sigma\left([W_1]_{1,\beta}\right)\right] \\
&= c_W \, \mathbb{E}\left[\sigma\left([W_1]_{1,\alpha}\right)\right] \mathbb{E}\left[\sigma\left([W_1]_{1,\beta}\right)\right],
\end{aligned}
$$

where the ultimate transition is due to the independence of $[W_1]_{1,\alpha}$ and $[W_1]_{1,\beta}$. Next, since $\mathcal{N}(\cdot)$ is symmetric and $\sigma$ is anti-symmetric, we obtain by Lemma 20 that

$$\mathbb{E}\left[\sigma\left([W_1]_{1,\alpha}\right)\right] = \mathbb{E}\left[\sigma\left([W_1]_{1,\beta}\right)\right] = 0.$$

Overall, we obtain that

$$K_{\alpha\beta}^{(1)} = c_W \cdot 0 \cdot 0 = 0.$$

Additionally, since $[W_1]_{1,\alpha}$ and $[W_1]_{1,\beta}$ are both drawn from $\mathcal{N}(\cdot)$, we obtain that

$$
\begin{aligned}
K_{\alpha\alpha}^{(1)} &= c_W \, \mathbb{E}\left[\sigma\left([W_1]_{1,\alpha}\right) \sigma\left([W_1]_{1,\alpha}\right)\right] \\
&= c_W \, \mathbb{E}\left[\sigma\left([W_1]_{1,\beta}\right) \sigma\left([W_1]_{1,\beta}\right)\right] \\
&= K_{\beta\beta}^{(1)}.
\end{aligned}
$$

Finally, by our assumption we have that

$$b^{(1)} := K_{\alpha\alpha}^{(1)} = c_W \, \mathbb{E}\left[ \sigma\left([W_1]_{1,\alpha}\right) \sigma\left([W_1]_{1,\alpha}\right)\right] = c_W \, \underset{x \sim \mathcal{Q}(\cdot)}{\mathbb{E}}[\sigma^2(x)] > 0\,.$$

as required. Next, fix $j \in [d]$ and assume that $K_{\alpha\beta}^{(j)} = 0$, $K_{\alpha\alpha}^{(j)} = K_{\beta\beta}^{(j)}$ and that $K_{\alpha\alpha}^{(j)}$ is finite and positive. Hence, plugging the inductive assumption into Theorem 4, we obtain that

$$K_{\alpha\beta}^{(j+1)} = c_W \, \mathbb{E}\left[\sigma(z_\alpha)\sigma(z_\beta)\right]$$

where

$$\begin{pmatrix} z_\alpha \\ z_\beta \end{pmatrix} \sim \mathcal{N}\left(0, \begin{pmatrix} K_{\alpha\alpha}^{(j)} & K_{\alpha\beta}^{(j)} \\ K_{\alpha\beta}^{(j)} & K_{\beta\beta}^{(j)} \end{pmatrix}\right) = \mathcal{N}\left(0, \begin{pmatrix} K_{\alpha\alpha}^{(j)} & 0 \\ 0 & K_{\alpha\alpha}^{(j)} \end{pmatrix}\right)\,.$$

Therefore, $z_\alpha$ and $z_\beta$ are independent identically distributed zero-centered Gaussian variables. Hence, we obtain that

$$K_{\alpha\beta}^{(j+1)} = c_W \, \mathbb{E}\left[\sigma(z_\alpha)\right] \mathbb{E}\left[\sigma(z_\beta)\right] = c_W \cdot 0 \cdot 0 = 0\,,$$

where the penultimate transition is due to Lemma 20. Additionally,

$$K_{\alpha\alpha}^{(j+1)} = c_W \, \mathbb{E}\left[\sigma(z_\alpha)\sigma(z_\alpha)\right] = c_W \, \mathbb{E}\left[\sigma(z_\beta)\sigma(z_\beta)\right] = K_{\beta\beta}^{(j+1)}\,.$$

Finally, we have by our inductive assumption that $K_{\alpha\alpha}^{(j)}$ is finite and positive, thus the non-constant random variable $z_\alpha \sim \mathcal{N}(0, K_{\alpha\alpha}^{(j)})$ has finite moments. Therefore, since $\sigma$ has a polynomially bounded derivative almost-everywhere and it is not constant, it holds that

$$b^{(j+1)} := K_{\alpha\alpha}^{(i+1)} = c_W \, \mathbb{E}\left[\sigma(z_\alpha)\sigma(z_\alpha)\right] > 0$$

completing the proof. $\qquad\square$

Theorem 7 and Lemma 6 together imply the following Corollary, which states that $\left(\mathbf{z}_\alpha^{(d)}\right)_{\alpha \in [m']}$ is bounded away from a zero-centered Gaussian vector with independent entries.

**Corollary 2.** *There exists $c > 0$ dependent on $m, m', d, \sigma, c_W, n, \epsilon_{\text{train}}$ and $\epsilon_{\text{gen}}$ such that for any $k \in \mathbb{N}$, the $m' \cdot m$-dimensional random vector $\left(\mathbf{z}_\alpha^{(d)}\right)_{\alpha \in [m']}$ corresponding to the concatenated outputs of the fully connected neural network (Definition 5) for the inputs $(\mathbf{e}_\alpha)_{\alpha \in [m']}$ satisfies*

$$d_c\left(\left(\mathbf{z}_\alpha^{(d)}\right)_{\alpha \in [m']}, \left(\Gamma_\alpha^{(d)}\right)_{\alpha \in [m']}\right) \le \frac{c}{\sqrt{k}}\,,$$

*where the random variables $\left(\left[\Gamma_\alpha^{(d)}\right]_j\right)_{\alpha \in [m'], j \in [m]}$ are independently drawn from $\mathcal{N}\left(0, b^{(d)}\right)$.*

Lemma 1 and Corollary 2 together imply that the matrix factorization $W$ has the required bound on its convex distance from a centered Gaussian matrix $W_{\text{iid}}$ (Definition 8). Hence, the proof follows by invoking Lemma 5 which implies Theorem 6.

## B  Proof of Theorem 2

This appendix proves Theorem 2. Appendix B.1 begins by establishing that for any $\gamma \in \mathbb{R}_{>0}$, the factorized matrix $W$ (Equation (4)) is within $\gamma$ (in Frobenius norm) of a rank one matrix with probability $1 - O(1/d)$. This finding is utilized by Appendix B.2, which establishes that the probability of the events $\mathcal{L}_{\text{train}}(W) < \epsilon_{\text{train}}$ and $\mathcal{L}_{\text{gen}}(W) \ge \epsilon_{\text{train}}c$ occurring simultaneously is $O(1/d)$. Appendix B.3 shows that the probability of $\mathcal{L}_{\text{train}}(W) < \epsilon_{\text{train}}$ is $\Omega(1)$. Finally, Appendix B.4 combines the findings of Appendices B.2 and B.3 to prove the sought-after result.

### B.1 $W$ is Close to a Rank One with High Probability

According to Definition 3, the matrices $W_j$ are obtained by normalizing matrices $W_j'$ where each entry of $W_j'$ is drawn independently from the distribution $\mathcal{Q}_j$. Specifically, each entry of $W_j$ equals the corresponding entry of $W_j'$ divided by $\|W_d' \cdots W_1'\|^{1/d}$. In this section, we will analyze the spectrum of the matrix $W_{d-1:2}' := W_{d-1}' \cdot \cdots \cdot W_2'$ using results from Hanin and Paouris [46] to show that this implies that with high probability, $W$ is close to a rank one matrix.

Note that because we normalize the final product, we can assume without loss of generality that $\mathcal{Q}(\cdot)$ is the standard normal distribution $\mathcal{N}(\cdot; 0, 1)$ rather than $\mathcal{N}(\cdot; 0, \nu)$. Any scaling factor from $\nu$ would be eliminated by the normalization. Thus, each entry of $W_j'$ is independently drawn from $\mathcal{N}(0, 1/m_j)$, where $m_j$ is the number of columns in $W_j$ as defined in Definition 3.

For any $d > 3$ and $t \in [k]$, we will denote the random variable that is the $t$th singular value of $W_{d-1:2}'$ by $s_{d-2,t}$, and the related quantity of the Lyapunov exponents by $\lambda_{d-2,t}$, defined as follows:

$$\lambda_{d-2,t} = \frac{1}{d-2} \log \left( s_{d-2,t} \right).$$

The following Theorem provides concentration bounds on the deviation of the Lyapunov exponents of $W_{d-1:2}'$.

**Theorem 8.** *There exist universal constants $\{\mu_{k,t}\}_{t=1}^k, c_1, c_2$ and $c_3$ such that for all $1 \leq p \leq r \leq k$, and any $s$ for which*

$$s \geq \frac{c_3 r}{(d-2)k} \log \left( \frac{ek}{r} \right),$$

*it holds that*

$$\mathcal{P} \left( \left| \frac{1}{k} \sum_{t=p}^r (\lambda_{d-2,t} - \mu_{k,t}) \right| \geq s \right) \leq c_1 \exp \left( -c_2 k(d-2)s \min \{1, \psi_{k,r}(s)\} \right),$$

*where $\psi_{k,r}(s)$ is the function*

$$\psi_{k,r}(s) = \begin{cases} k \min \left\{ 1, \frac{ks}{r} \right\}, & r \leq \frac{k}{2} \\ k \min \left\{ \eta_{k,r}, \frac{s}{\log(1/\eta_{k,r})} \right\}, & \frac{k}{2} < r \leq k \end{cases}$$

*for*

$$\eta_{k,r} := \frac{k-r+1}{k} \in \left[ \frac{1}{k}, \frac{k-1}{k} \right].$$

*Proof.* See Theorem 1.1 in Hanin and Paouris [46]. $\qquad \square$

We now show that the above deviation estimate implies that with probability converging exponentially to 1, there is a constant gap between the largest and second largest Lyapunov exponents.

**Lemma 7.** *There exist constants $c_4, c_5, c_6 > 0$ independent of $d$ such that for all $d \geq c_4$*

$$\mathcal{P} \left( \lambda_{d-2,2} \leq \lambda_{d-2,1} - c_5 \right) \geq 1 - \exp \left( -c_6(d-2) \right).$$

*Proof.* Obviously $\mu_{k,1} - \mu_{k,2} > 0$. We define $c_5 := \frac{\mu_{k,1} - \mu_{k,2}}{3}$. Plugging in $p = r = 1$ into Theorem 8 we get that

$$\mathcal{P} \left( |\lambda_{d-2,1} - \mu_{k,1}| \geq s \right) \leq c_1 \exp \left( -c_2 k(d-2)s \min \{1, \psi_{k,1}(s)\} \right)$$

for all $s \geq \frac{c_3}{k(d-2)} \log (ek)$. We take $d$ large enough such that $c_5 \geq \frac{c_3}{k(d-2)} \log (ek)$ and take $s = c_5$. Plugging in the definition of $\psi_{k,1}$, one may obtain that

$$c_2 k(d-2)s \min \{1, \psi_{k,1}(s)\} = \Omega(d),$$

hence

$$\mathcal{P} \left( |\lambda_{d-2,1} - \mu_{k,1}| \geq c_5 \right) \leq c_1 \exp \left( -\Omega(d) \right).$$

The same argument can be applied for $p = r = 2$ to conclude that

$$\mathcal{P} \left( |\lambda_{d-2,2} - \mu_{k,2}| \geq c_5 \right) \leq c_1 \exp \left( -\Omega(d) \right).$$

Combining these two results by union bound and the triangle inequality yields the theorem. $\qquad \square$

The following Lemma implies that the spectrum of $W'_{d-1:2}$ is rapidly decaying, and in particular that it can be well approximated by a rank one matrix.

**Lemma 8.** *Let $E$ be the best rank one approximation to $W'_{d-1:2}$. It holds with probability at least $1 - \exp\left(-c_6(d-2)\right)$ that*

$$\frac{\|W'_{d-1:2} - E\|_F}{\|E\|_F} \leq \sqrt{(k-1)} \exp\left(-c_5(d-2)\right),$$

*where $c_5$ and $c_6$ are the same constants as in Lemma 7.*

*Proof.* By the definition of Lyapunov exponents and Lemma 7 we have that with probability $\geq 1 - \exp\left(-c_6(d-2)\right)$, the following inequality holds for all $t \geq 2$:

$$\frac{s_{d-2,1}}{s_{d-2,t}} \geq \frac{s_{d-2,1}}{s_{d-2,2}} = \exp\left((d-2)(\lambda_{d-2,1} - \lambda_{d-2,2})\right) \geq \exp\left(c_5(d-2)\right),$$

hence we obtain that

$$\frac{\|W'_{d-1:2} - E\|_F}{\|E\|_F} = \frac{\sqrt{\sum_{t=2}^{k}(s_{d-2,t})^2}}{s_{d-2,1}} \leq \frac{\sqrt{(k-1)(s_{d-2,2})^2}}{s_{d-2,1}} \leq \sqrt{(k-1)} \exp\left(c_5(d-2)\right)$$

as required. $\qquad\square$

We now show that not only $W'_{d-1:2}$, but also the end-to-end matrix $W' = W'_d W'_{d-1:2} W'_1$ is approximately rank one.

**Lemma 9.** *There exist constants $c_{11}, c_{12}, c_{13} > 0$ independent of $d$ such that with probability at least*

$$1 - 2\frac{c_{12}}{d} - 2\frac{c_{11}}{d^{(k^2)}} - \exp\left(-c_{13}(d-2)\right),$$

*the product of unnormalized matrices $W' := W'_d \cdot \ldots \cdot W'_1$ can be written as*

$$W' = O + R$$

*where $O$ is a rank one matrix and*

$$\frac{\|R\|_F}{\|O\|_F} \leq d^6 \sqrt{(k-1)} \exp\left(-c_5(d-2)\right)$$

*for the constant $c_5$ described in Lemma 8.*

*Proof.* We start from the decomposition obtained in Lemma 8, namely the decomposition

$$W'_{d-1:2} = E + (W'_{d-1:2} - E),$$

where $E$ has rank one and

$$\frac{\|W'_{d-1:2} - E\|_F}{\|E\|_F} \leq \sqrt{(k-1)} \exp\left(-c_5(d-2)\right),$$

which holds with probability $\geq 1 - \exp\left(-c_6(d-2)\right)$. Plugging into $W'$ we obtain that

$$W' = W'_d W'_{d-1:2} W'_1 = W'_d E W'_1 + W'_d \left(W'_{d-1:2} - E\right) W'_1.$$

Note that $\text{rank}(W'_d E W'_1) = 1$ whenever $W'_d E W'_1 \neq 0$, which holds with probability 1. Now we can set $O = W'_d E W'_1$ and $R = W'_d(W'_{d-1:2} - E)W'_1$. It therefore suffices to upper bound the ratio

$$\frac{\|W'_d(W'_{d-1:2} - E)W'_1\|_F}{\|W'_d E W'_1\|_F}.$$

We will separately give an upper bound on

$$\|W'_d(W'_{d-1:2} - E)W'_1\|_F$$

and a lower bound on

$$\|W'_d E W'_1\|$$

that hold simultaneously with high probability. First, note that by Lemmas 24 and 25, for a sufficiently large $d$, with probability $\geq 1 - 2\exp\left(-c_{10}d\right)$ it holds that

$$\|W_d'(W_{d-1:2}' - E)W_1'\|_F \leq \|W_d'\|_F\|W_1'\|_F\|W_{d-1:2}' - E\|_F$$
$$\leq d^2\|W_{d-1:2}' - E\|_F.$$

For the lower bound on $\|W_d'EW_1'\|_F$, consider the SVD decompositions of $W_d'$ and $W_1'$ given by

$$W_1' = \sum_{t=1}^{r_1} \sigma_t^1 u_t^1 \left(v_t^1\right)^\top$$

and

$$W_d' = \sum_{t=1}^{r_d} \sigma_t^d u_t^d \left(v_t^d\right)^\top.$$

Likewise, as a rank one matrix, $E$ can be written as

$$E = \|E\|_F u_E v_E^\top.$$

Invoking Lemma 26 with $i = j = 1$ we obtain that

$$\|W_d'EW_1'\|_F \geq \|E\|_F \sigma_1^d \sigma_1^1 \left\langle v_1^d, u_E\right\rangle \left\langle v_E, u_1^1\right\rangle.$$

It suffices to lower bound the absolute values of each of the terms in the product above. Note that by Lemma 27 all terms are independent. To lower bound $\sigma_1^1$ and $\sigma_1^d$ we apply Lemma 28 which yields that

$$\mathcal{P}\left(\sigma_1^1 \geq \frac{1}{d}\right) \geq 1 - \frac{c_{11}}{d^{(k^2)}}$$

and

$$\mathcal{P}\left(\sigma_1^d \geq \frac{1}{d}\right) \geq 1 - \frac{c_{11}}{d^{(k^2)}}$$

where $c_{11}$ is the constant from Lemma 28. To lower bound $\left|\left\langle v_1^d, u_E\right\rangle\right|$ and $\left|\left\langle v_E, u_1^1\right\rangle\right|$ we invoke Lemma 29 which yields that

$$\mathcal{P}\left(\left|\left\langle v_1^d, u_E\right\rangle\right| \geq \frac{1}{d}\right) \geq 1 - \frac{c_{12}}{d}$$

and

$$\mathcal{P}\left(\left|\left\langle v_E, u_1^1\right\rangle\right| \geq \frac{1}{d}\right) \geq 1 - \frac{c_{12}}{d}$$

where $c_{12}$ is the constant from Lemma 29. Hence by the union bound, with probability at least $1 - 2\frac{c_{12}}{d} - 2\frac{c_{11}}{d^{(k^2)}}$ it holds that

$$\|W_d'EW_1'\|_F \geq \frac{1}{d^4}\|E\|_F.$$

Applying the union bound once more, we obtain that with probability at least

$$1 - 2\frac{c_{12}}{d} - 2\frac{c_{11}}{d^{(k^2)}} - \exp\left(-c_6(d - 2)\right) - 2\exp\left(-c_{10}d\right)$$

the following holds:

$$\frac{\|W_d'(W_{d-1:2}' - E)W_1'\|_F}{\|W_d'EW_1'\|_F} \leq d^6\sqrt{(k-1)}\exp\left(-c_5(d-2)\right).$$

The proof follows by choosing $c_{13}$ such that

$$\exp\left(-c_{13}(d-2)\right) \geq \exp\left(-c_6(d-2)\right) + 2\exp\left(-c_{10}d\right).$$

$\square$

Now that we've shown that $W'$ can be approximated by a rank one matrix with high probability as $d$ tends to infinity, we are ready to show this for the normalized matrix $W = W'/\|W'\|_F$ as well.

**Lemma 10.** *Let*

$$C(\gamma) := \{W : \ W = X + Y, \operatorname{rank}(X) = 1, \|Y\|_F < \gamma\} \ .$$

*Let $C_{d,\gamma}$ be the event that $\frac{W'}{\|W'\|_F} \in C(\gamma)$. Then for any $\gamma > 0$, if*

$$\gamma \geq \frac{1}{|d^{-6}(k-1)^{-0.5} \exp\left(c_5(d-2)\right) - 1|}$$

*then*

$$\mathcal{P}\left(C_{d,\gamma}\right) \geq 1 - 2\frac{c_{12}}{d} - 2\frac{c_{11}}{d^{(k^2)}} - \exp\left(-c_{13}(d-2)\right),$$

*where $c_{11}, c_{12}$, and $c_{13}$ are the same constants as in Lemma 9.*

*Proof.* By Lemma 9, with probability $\geq 1 - 2\frac{c_{12}}{d} - 2\frac{c_{11}}{d^{(k^2)}} - \exp\left(-c_{13}(d-2)\right)$ it holds that the unnormalized matrix $W'$ can be written as

$$W' = O + R\,,$$

where $O$ has rank one and

$$\frac{\|R\|_F}{\|O\|_F} \leq d^6 \sqrt{(k-1)} \exp\left(-c_5(d-2)\right)$$

from which it follows that

$$\frac{\|R\|_F}{\|W'\|_F} \leq \frac{\|R\|_F}{|\|O\|_F - \|R\|_F|} \leq \frac{1}{|d^{-6}(k-1)^{-0.5}\exp\left(c_5(d-2)\right) - 1|}\,.$$

Consider the normalized matrix $W'/\|W'\|_F$ and its best rank one approximation denoted as $X$. Then, it holds that

$$\left\|\frac{W'}{\|W'\|_F} - X\right\|_F \leq \frac{\|W' - O\|_F}{\|W'\|_F} = \frac{\|R\|_F}{\|W'\|_F} \leq \frac{1}{|d^{-6}(k-1)^{-0.5}\exp\left(c_5(d-2)\right) - 1|} \leq \gamma.$$

as required. $\qquad\square$

## B.2 If $W$ is Close to Rank One Then Low Training Loss Ensures Low Generalization Loss

In this appendix, we show that if the RIP holds and a learned matrix $W$ is close enough to a rank one matrix, achieving low training loss ensures low generalization loss. This is formally stated in the Lemma below.

**Lemma 11.** *Suppose that the measurement matrices $(A_i)_{i=1}^n$ satisfy the RIP of order $1$ (see Definition 1) with a constant $\delta \in (0,1)$ and $\mathcal{A}$ is defined as in Definition 1. Suppose that there exists some constant $b > 0$ such that for any matrix $M \in \mathbb{R}^{m,m'}$ it holds that*

$$\|\mathcal{A}(M)\|_F \leq b\|M\|_F\,.$$

*Let $M \in \mathbb{R}^{m,m'}$ be a matrix such that*

$$\|\mathcal{A}(M)\|_F^2 \leq \epsilon\,,$$

*and suppose that*

$$M = E + R\,,$$

*where $\operatorname{rank}(E) \leq 1$. If*

$$\|R\|_F \leq \frac{\sqrt{1-\delta}(\sqrt{2}-1)\sqrt{\epsilon}}{1 + b\sqrt{1-\delta}}\,,$$

*then*

$$\|M\|_F^2 \leq \frac{2\epsilon}{1-\delta}\,.$$

*Proof.* By the definition of $\mathcal{A}$ it holds that

$$\mathcal{A}(E) = \mathcal{A}(M) - \mathcal{A}(R),$$

therefore, using the triangle inequality we obtain that

$$\|\mathcal{A}(E)\|_F \leq \|\mathcal{A}(M)\|_F + \|\mathcal{A}(R)\|_F \leq \sqrt{\epsilon} + b\|R\|_F.$$

By the RIP, the above results in

$$\|E\|_F \leq (1 - \delta)^{-\frac{1}{2}}(\sqrt{\epsilon} + b\|R\|_F).$$

Finally, we obtain by the triangle inequality that

$$\|M\|_F \leq (1 - \delta)^{-\frac{1}{2}}(\sqrt{\epsilon} + b\|R\|_F) + \|R\|.$$

The proof follows by plugging the assumption on $\|R\|_F$ and rearranging. $\qquad\square$

## B.3 Lower Bounds on the Probability of Low Training Loss

In this Appendix, we show that the probability of attaining low training loss is bounded from below as $d$ tends to infinity. The argument is formally stated in the next Lemma.

**Lemma 12.** *Suppose that $W^*$ has rank one and that $\|W^*\|_F = 1$. Then for any $\epsilon > 0$ it holds that*

$$\mathcal{P}\left(\mathcal{L}_{\text{train}}(W) < \epsilon\right) \geq \Omega(1)$$

*as $d \to \infty$.*

*Proof.* By the law of total probability we have that

$$\mathcal{P}\left(\mathcal{L}_{\text{train}}(W) < \epsilon\right) \geq \mathcal{P}\left(\mathcal{L}_{\text{train}}(W) < \epsilon \Big| C_{d,\gamma}\right) \cdot \mathcal{P}\left(C_{d,\gamma}\right),$$

where $C_{d,\gamma}$ is as defined in Lemma 10. Additionally, By Lemma 10 it holds that

$$\lim_{d \to \infty} \mathcal{P}\left(C_{d,\gamma}\right) = 1.$$

It therefore suffices to show that

$$\mathcal{P}\left(\mathcal{L}_{\text{train}}(W) < \epsilon \Big| C_{d,\gamma}\right) \geq \Omega(1).$$

as $d \to \infty$. Observe that

$$\mathcal{P}\left(\mathcal{L}_{\text{train}}(W) < \epsilon \Big| C_{d,\gamma}\right) \geq \mathcal{P}\left(\|W - W^*\|_F^2 < \frac{\epsilon}{b} \Big| C_{d,\gamma}\right),$$

where $b$ is the Lipschitz constant of $\mathcal{A}$ as defined in Lemma 11. Thus it suffices to lower bound the latter probability. By the symmetry of the Gaussian distribution (see Lemma 27) and the symmetry of the event $C_{d,\gamma}$ we have that for any $c > 0$ the following holds for any rank one matrix $E$ with $\|E\|_F = 1$:

$$\mathcal{P}\left(\|W - W^*\|_F^2 < \frac{\epsilon}{b} \Big| C_{d,\gamma}\right) = \mathcal{P}\left(\|W - E\|_F^2 < \frac{\epsilon}{b} \Big| C_{d,\gamma}\right).$$

Now we set $\gamma = \epsilon/2b$ and consider the $M := M\left(\frac{\epsilon}{2b}, d\right)$ matrices $E_1, \ldots, E_M$ from Lemma 30. By the triangle inequality and the union bound we have that

$$
\begin{aligned}
1 &= \mathcal{P}\left(C_{d,\gamma} \Big| C_{d,\gamma}\right) \\
&= \mathcal{P}\left(\bigcup_{1 \leq i \leq M} \left\{W \; ; \; \|W - E_i\|_F^2 < \frac{\epsilon}{b}\right\} \Big| C_{d,\gamma}\right) \\
&\leq \sum_{1 \leq i \leq M} \mathcal{P}\left(\left\{W \; ; \; \|W - E_i\|_F^2 < \frac{\epsilon}{b}\right\} \Big| C_{d,\gamma}\right).
\end{aligned}
$$

Now again by symmetry we have

$$\sum_{1 \leq i \leq M} \mathcal{P}\left(\left\{W \; ; \; \|W - E_i\|_F^2 < \frac{\epsilon}{b}\right\} \Big| C_{d,\gamma}\right) = M \cdot \mathcal{P}\left(\left\{W \; : \; \|W - W^*\|_F^2 < \frac{\epsilon}{b}\right\} \Big| C_{d,\gamma}\right).$$

Therefore, we conclude that

$$\mathcal{P}\left(\left\{W \; : \; \|W - W^*\|_F^2 < \frac{\epsilon}{b}\right\} \Big| C_{d,\gamma}\right) \geq \frac{1}{M}$$

as required. $\qquad\square$

## B.4 Proof of Sought-After Result

We are now ready to prove Theorem 2. Let us define $C_{d,\gamma}$ as in Lemma 10. For convenience we will denote by $G_{d,c}$ the event that $\mathcal{L}_{\text{gen}}(W) < c\epsilon$, and by $L_d$ the event that $\mathcal{L}_{\text{train}}(W) < \epsilon$. By the law of total probability it holds that

$$\mathcal{P}\left(G_{d,c}\big|L_d\right) \geq \frac{\mathcal{P}\left(G_{d,c} \cap L_d \cap C_{d,\gamma}\right)}{\mathcal{P}\left(L_d\right)}.$$

By Lemma 11, if we set

$$\gamma = \frac{\sqrt{1-\delta}(\sqrt{2}-1)\sqrt{\epsilon}}{1+b\sqrt{1-\delta}}$$

where $b$ is the Lipschitz constant of $\mathcal{A}$, and

$$c = \frac{2}{1-\delta},$$

then we obtain that

$$G_{d,c} \cap L_d \cap C_{d,\gamma} \subseteq L_d \cap C_{d,\gamma}.$$

Hence,

$$\mathcal{P}\left(G_{d,c}\big|L_d\right) \geq \frac{\mathcal{P}\left(L_d \cap C_{d,\gamma}\right)}{\mathcal{P}\left(L_d\right)} = 1 - \frac{\mathcal{P}\left(L_d \cap C_{d,\gamma}^C\right)}{\mathcal{P}\left(L_d\right)} \geq 1 - \frac{\mathcal{P}\left(C_{d,\gamma}^C\right)}{\mathcal{P}\left(L_d\right)}.$$

By Lemma 10, for large enough $d$ it holds that

$$\mathcal{P}\left(C_{d,\gamma}^C\right) < 2\frac{c_{12}}{d} + 2\frac{c_{11}}{d^{(k^2)}} + \exp\left(-c_6(d-2)\right) = O(1/d).$$

By Lemma 12, $\mathcal{P}\left(L_d\right) = \Omega(1)$ and so

$$\lim_{d\to\infty} \mathcal{P}\left(G_{d,c}\big|L_d\right) = 1 - O\left(1/d\right)$$

completing the proof. $\qquad\square$

## C Increasing Width with Unspecified $\epsilon_{\text{train}}$

Theorem 1 requires the G&C training loss threshold $\epsilon_{\text{train}}$ to be specified. Thus, the theorem does not rule out the possibility that for any width, a sufficiently small $\epsilon_{\text{train}}$ will lead G&C to attain good generalization. In this appendix, we state and prove a result—Theorem 9 below—that allows for an unspecified $\epsilon_{\text{train}}$. Theorem 9 imposes assumptions beyond those of Theorem 1: *(i)* the depth of the matrix factorization is two; *(iii)* the prior distribution is generated by a zero-centered Gaussian; *(iii)* the activation is linear; and *(iv)* the factorization is square, *i.e.*, $m = m'$ (though this latter assumption can easily be lifted). Moreover, Theorem 9 considers a case in which the prior-induced probability distribution of the factorized matrix $W$ (Equation (4)) is shifted such that its mean is the ground truth matrix $W^*$. The theorem establishes that even with this shift—which makes good generalization easier to attain—the posterior probability of low generalization loss conditioned on low training loss converges to the prior probability of low generalization loss.

**Theorem 9.** *Let $m, k \in \mathbb{N}$ and let $\epsilon_{\text{gen}} > 0$. Let $W_1$ and $W_2$ be random matrices of dimensions $m, k$ and $k, m$, respectively. Assume that the entries of both $W_1$ and $W_2$ are drawn independently from $\mathcal{N}(0,1)$. Consider the normalized product which is then centered around the ground truth matrix $W^*$, i.e.:*

$$W := \frac{1}{\sqrt{mk}}W_1 W_2 + W^*.$$

*Then, it holds that:*

$$\lim_{k\to\infty} \sup_{\epsilon_{\text{train}}>0} \mathcal{P}\left(\mathcal{L}_{\text{gen}}(W) < \epsilon_{\text{gen}} \mid \mathcal{L}_{\text{train}}(W) < \epsilon_{\text{train}}\right) - \mathcal{P}\left(\mathcal{L}_{\text{gen}}(W) < \epsilon_{\text{gen}}\right) = 0.$$

*Proof.* The proof is delivered by Appendices C.1 to C.4 below. Appendix C.1 establishes that locally uniform convergence of densities implies convergence of probabilities. Appendix C.2 calculates the characteristic function of the factorized matrix $W$. Appendix C.3 establishes that the conditional probabilities in Theorem 9 can be approximated by conditional probabilities of bounded sets. Finally, Appendix C.4 combines the above to prove the sought-after result. $\qquad\square$

## C.1 Locally Uniform Convergence of Densities Implies Convergence of Probabilities

Convergence of probability measures in convex distance, as in Appendix A, is not sufficient for proving a results which deal directly with the case of unspecified $\epsilon_{\text{train}}$. This is so because the denominator and the numerator of the conditional probability can be arbitrarily small, and the convex distance gives an additive approximation guarantee. To prove such a result we will first have to prove that a stronger notion of convergence which we introduce in this Appendix holds in our case.

**Theorem 10.** *Suppose that $G_n$ and $G$ have continuous densities $g_n$ and $g$ (with respect to Lebesgue measure) on $\mathbb{R}^m$. Suppose that $G_n \xrightarrow{\text{dist.}} G$, the densities $g_n$ are uniformly bounded, i.e., there exists $M : \mathbb{R}^m \to \mathbb{R}$ such that*

$$\forall \mathbf{x} \in \mathbb{R}^m, \quad \sup_{n \in \mathbb{N}} |g_n(\mathbf{x})| \le M(\mathbf{x}) < \infty \,,$$

*and $g_n$ are also equicontinuous, i.e., for each $\epsilon > 0$ there exists $\delta(\epsilon)$ such that if $\|\mathbf{x} - \mathbf{y}\|_2 < \delta(\epsilon)$ then*

$$\forall n \in \mathbb{N}, \quad |g_n(\mathbf{x}) - g_n(\mathbf{y})| < \epsilon \,.$$

*Then*

$$\lim_{n \to \infty} \sup_{\mathbf{x} \in \mathbb{R}^m} |g_n(\mathbf{x}) - g(\mathbf{x})| \to 0 \,.$$

*Proof.* This is a slight adaptation of Lemma 1 in Boos [15]. $\qquad\square$

**Lemma 13.** *Let $\{f_n\}_{n \in \mathbb{N}}$ be a family of probability densities on $\mathbb{R}^m$ with respective characteristic functions $\varphi_n(\mathbf{t})$ (Definition 11). Suppose that the $L^1$-norms of the characteristic functions weighted by $\|\mathbf{t}\|_2$ are uniformly bounded, i.e.,*

$$\forall n \in \mathbb{N}, \quad \int_{\mathbb{R}^m} \|\mathbf{t}\|_2 \cdot |\varphi_n(\mathbf{t})| d\mathbf{t} \le M \,,$$

*where $M > 0$ is a constant independent of $n$. Then the family $f_n$ is uniformly bounded and equicontinuous.*

*Proof.* Note that the above bound implies that there also exists some $\hat{M} > 0$ independent of $n$ such that

$$\forall n \in \mathbb{N}, \quad \int_{\mathbb{R}^m} |\varphi_n(\mathbf{t})| d\mathbf{t} \le \hat{M} \,.$$

We employ the decomposition

$$\int_{\mathbb{R}^m} |\varphi_n(\mathbf{t})| d\mathbf{t} = \int_{\|\mathbf{t}\|_2 \le 1} |\varphi_n(\mathbf{t})| d\mathbf{t} + \int_{\|\mathbf{t}\|_2 \ge 1} |\varphi_n(\mathbf{t})| d\mathbf{t} \,.$$

The first summand is uniformly bounded for all $n$ because the integrand is at most $1$ (any characteristic function satisfies $|\phi(\mathbf{t})| \le \mathbb{E}(|\exp(i \langle \mathbf{t}, X \rangle)|) = 1$) and the domain has finite volume, whereas the second summand is upper bounded by

$$\int_{\|\mathbf{t}\|_2 \ge 1} \|\mathbf{t}\|_2 \cdot |\varphi_n(\mathbf{t})| d\mathbf{t} \le \int_{\mathbb{R}^m} \|\mathbf{t}\|_2 \cdot |\varphi_n(\mathbf{t})| d\mathbf{t} \le M.$$

It now follows by Lemma 32 that for any $\mathbf{x} \in \mathbb{R}^m$

$$\sup_{n \in \mathbb{N}} |f_n(\mathbf{x})| \le \sup_{n \in \mathbb{N}, \mathbf{y} \in \mathbb{R}^m} |f_n(\mathbf{y})| \le \frac{1}{(2\pi)^m} \hat{M} \,.$$

It remains to verify that $\{f_n\}_{n \in \mathbb{N}}$ are equicontinuous. To see this, note that by Lemma 33 :

$$|f_n(\mathbf{x}) - f_n(\mathbf{y})| \le \frac{\|\mathbf{x} - \mathbf{y}\|_2}{(2\pi)^m} \int_{\mathbb{R}^m} \|\mathbf{t}\|_2 \cdot |\varphi_n(\mathbf{t})| d\mathbf{t} \le \frac{\|\mathbf{x} - \mathbf{y}\|_2}{(2\pi)^m} M.$$

This bound is uniform in $n$ and depends only on the distance $\|\mathbf{x} - \mathbf{y}\|_2$. For any given $\epsilon > 0$, choose $\delta(\epsilon) = \frac{(2\pi)^m \epsilon}{M}$. Then for all $\|\mathbf{x} - \mathbf{y}\|_2 < \delta(\epsilon)$, we have:

$$\forall n \in \mathbb{N}, \quad |f_n(\mathbf{x}) - f_n(\mathbf{y})| < \epsilon \,.$$

Thus, the family $\{f_n\}_{n \in \mathbb{N}}$ is equicontinuous. $\qquad\square$

**Lemma 14.** *Let $\{f_n\}_{n\in\mathbb{N}}$ be a sequence of probability density functions on $\mathbb{R}^m$ that converges uniformly to a limit density $f$. Suppose that $f$ is positive and smooth on $\mathbb{R}^m$. For any bounded set $K \subset \mathbb{R}^m$ such that $K \subseteq B(0, R)$, the ratio of probabilities assigned by $f_n$ and $f$ to $K$ converges to 1, i.e.,*

$$\lim_{n\to\infty} \frac{\int_K f_n(\mathbf{x})d\mathbf{x}}{\int_K f(\mathbf{x})d\mathbf{x}} \to 1\,.$$

*furthermore, this convergence is uniform over all subsets contained in $B(0, R)$.*

*Proof.* The positivity and smoothness of $f$ imply that $f$ is bounded below on bounded sets—indeed, $K$ is contained in $B(0, R)$ on which $f$ is bounded from below. That is, there exists a constant $c > 0$ such that for all $\mathbf{x} \in B(0, R)$ it holds that

$$f(\mathbf{x}) \geq c\,.$$

Since $f_n \to f$ uniformly on $\mathbb{R}^m$, for any $\epsilon > 0$ there exists $N \in \mathbb{N}$ such that for all $n \geq N$ and all $\mathbf{x} \in K$ it holds that

$$|f_n(\mathbf{x}) - f(\mathbf{x})| < c\epsilon \leq \epsilon f(\mathbf{x})\,.$$

Thus, we can bound $f_n(\mathbf{x})$ as:

$$(1 - \epsilon)f(\mathbf{x}) \leq f_n(\mathbf{x}) \leq (1 + \epsilon)f(\mathbf{x})\,.$$

Integrating this inequality over $K$, we obtain

$$(1 - \epsilon)\int_K f(\mathbf{x})d\mathbf{x} \leq \int_K f_n(\mathbf{x})d\mathbf{x} \leq (1 + \epsilon)\int_K f(\mathbf{x})d\mathbf{x}\,.$$

Dividing through by $\int_K f(\mathbf{x})d\mathbf{x}$ (which is strictly positive since $f > 0$ and $K$ is compact), we obtain that

$$1 - \epsilon \leq \frac{\int_K f_n(\mathbf{x})d\mathbf{x}}{\int_K f(\mathbf{x})d\mathbf{x}} \leq 1 + \epsilon\,.$$

Taking the limit as $n \to \infty$, we conclude:

$$\frac{\int_K f_n(\mathbf{x})d\mathbf{x}}{\int_K f(\mathbf{x})d\mathbf{x}} \to 1\,.$$

Furthermore, the above convergence does not depend on $K$ itself but only on $B(0, R)$, hence the convergence is uniform over all subsets contained in $B(0, R)$. $\qquad\square$

### C.2 Calculation of Characteristic Function

To apply the results of the previous subsection, we need to calculate the characteristic function so that we can bound its integral as in Lemma 16. We begin with the following formula.

**Lemma 15.** *Given the random variable $W = \frac{1}{\sqrt{mk}}W_1 W_2$, where $W_1, W_2^\top \in \mathbb{R}^{m,k}$ are matrices with independent standard Gaussian entries, the characteristic function $\hat{f}_k(T) = \mathbb{E}[e^{i\langle T, W\rangle}]$ for $T \in \mathbb{R}^{m,m}$ is given by:*

$$\hat{f}_k(T) = \left( \frac{1}{\sqrt{\det\left( I_m + \frac{TT^\top}{km} \right)}} \right)^k\,.$$

*Proof.* Since for any random variable $X$ and constant $c \in \mathbb{R}$ we have

$$\hat{f}_{cX}(T) = \hat{f}_X(cT)\,,$$

it suffices to compute the characteristic function without the $\frac{1}{\sqrt{mk}}$ factor, which we denote by $\hat{f}$. We have that

$$
\begin{aligned}
\langle T, W \rangle &= \sum_{i=1}^{m} \sum_{j=1}^{m} T_{ij} W_{ij} \\
&= \sum_{i=1}^{m} \sum_{j=1}^{m} T_{ij} \sum_{p=1}^{k} W_{ip}^{(1)} W_{pj}^{(2)} \\
&= \sum_{i=1}^{m} \sum_{j=1}^{m} \sum_{p=1}^{k} T_{ij} W_{ip}^{(1)} W_{pj}^{(2)} \\
&= \sum_{p=1}^{k} \sum_{j=1}^{m} W_{pj}^{(2)} \sum_{i=1}^{m} T_{ij} W_{ip}^{(1)} .
\end{aligned}
$$

Note that the variables

$$
\sum_{j=1}^{m} W_{pj}^{(2)}, \quad \sum_{i=1}^{m} T_{ij} W_{ip}^{(1)}
$$

are independent for distinct values of $p$, hence

$$
\hat{f}_k(T) = \prod_{p=1}^{k} \mathbb{E}_{W_{ip}^{(1)}, W_{pj}^{(2)}} \left[ \exp \left( i \sum_{j=1}^{m} W_{pj}^{(2)} \sum_{i=1}^{m} T_{ij} W_{ip}^{(1)} \right) \right] .
$$

To evaluate the above expression we first fix $W_{pj}^{(2)}$, *i.e.*, we consider the expectation

$$
\mathbb{E}_{W_{ip}^{(1)}} \left[ \exp \left( i \sum_{j=1}^{m} W_{pj}^{(2)} \sum_{i=1}^{m} T_{ij} W_{ip}^{(1)} \right) \right] .
$$

Let $Z \in \mathbb{R}^m$ such that $(Z)_j := \sum_{i=1}^{m} T_{ij} W_{ip}^{(1)}$. Note that $Z$ is a Gaussian random variable (being a linear combination of Gaussian random variables) with mean zero and covariance

$$
(\Sigma_Z)_{pl} = \langle T_p, T_l \rangle ,
$$

where $T_p, T_l \in \mathbb{R}^m$ are the $p$th and $l$th rows of the matrix $T$, respectively. Therefore by the formula for the characteristic function of a Gaussian variable (see Lemma 34) we obtain

$$
\mathbb{E}_{W_{ip}^{(1)}} \left[ \exp \left( i \sum_{j=1}^{m} W_{pj}^{(2)} \sum_{i=1}^{m} T_{ij} W_{ip}^{(1)} \right) \right] = \exp \left( -0.5 \left\langle W_p^{(2)}, \Sigma_Z W_p^{(2)} \right\rangle \right) ,
$$

where $W_p^{(2)} \in \mathbb{R}^m$ is the vector whose $j$th entry is $W_{pj}^{(2)}$. It remains now to evaluate this expectation with respect to $W_p^{(2)}$ as well. Thus our task reduces to evaluating the expectation, with respect to a standard Gaussian, of a function of the form

$$
\exp \left( -0.5 \left\langle W_p^{(2)}, \Sigma_Z W_p^{(2)} \right\rangle \right)
$$

where $\Sigma$ is PSD. We now apply Lemma 35 to obtain the formula:

$$
\mathbb{E}_{W_p^{(2)}} \left[ \exp \left( -0.5 \left\langle W_p^{(2)}, \Sigma_Z W_p^{(2)} \right\rangle \right) \right] = \frac{1}{\sqrt{\det(I_m + \Sigma)}} = \frac{1}{\sqrt{\det(I_m + TT^\top)}}
$$

where the second equality uses the definition of $\Sigma$. It follows that

$$
\hat{f}_k(T) = \hat{f} \left( \frac{T}{\sqrt{k}} \right) = \left( \frac{1}{\sqrt{\det \left( I_m + \frac{TT^\top}{km} \right)}} \right)^k
$$

as required. $\qquad \square$

We are now ready to show that the integrals of the characteristic functions $\hat{f}_k$ are bounded, even if multiplied by $\|T\|_F$, which will allow us to derive the equicontinuity of the corresponding densities.

**Lemma 16.** *For $\hat{f}_k$ defined in Lemma 15 it holds that*

$$\sup_{k \in \mathbb{N}} \int_{\mathbb{R}^{m,m}} \|T\|_F \cdot |\hat{f}_k(T)| dT < \infty.$$

*Proof.* Recall that by Lemma 15 above it holds that

$$\int_{\mathbb{R}^{m,m}} \|T\|_F |\hat{f}_k(T)| dT = \int_{\mathbb{R}^{m,m}} \|T\|_F \left( \frac{1}{\sqrt{\det(I_m + \frac{TT^\top}{km})}} \right)^k dT.$$

Using the change of variables to singular values and the Vandermonde determinant (Corollary 3), we may write the above as

$$\int_{\mathbb{R}^{m,m}} \|T\|_F \left( \frac{1}{\sqrt{\det(I_m + \frac{TT^\top}{km})}} \right)^k dT$$

$$= C_m \int_{\mathbb{R}^m_+} \left( \sum_{p=1}^m \sigma_p^2 \right)^{1/2} \cdot \left( \frac{1}{\prod_{i=1}^m \sqrt{1 + \frac{\sigma_i^2}{km}}} \right)^k \prod_{1 \leq i < j \leq m} (\sigma_i^2 - \sigma_j^2) d\boldsymbol{\sigma}$$

where $C_m$ is a constant depending on the dimension $m$. Using the elementary bound

$$\prod_{1 \leq i < j \leq m} (\sigma_i^2 - \sigma_j^2) \leq \prod_{i=1}^m \sigma_i^{2(m-i)}$$

we obtain the following upper inequality

$$\int_{\mathbb{R}^m_+} \left( \sum_{p=1}^m \sigma_p^2 \right)^{1/2} \cdot \left( \frac{1}{\prod_{i=1}^m \sqrt{1 + \frac{\sigma_i^2}{km}}} \right)^k \prod_{1 \leq i < j \leq m} (\sigma_i^2 - \sigma_j^2) d\boldsymbol{\sigma}$$

$$\leq C_m \int_{\mathbb{R}^m_+} \left( \sum_{p=1}^m \sigma_p^2 \right)^{1/2} \cdot \left( \frac{1}{\prod_{i=1}^m \sqrt{1 + \frac{\sigma_i^2}{km}}} \right)^k \prod_{i=1}^m \sigma_i^{2(m-i)} d\boldsymbol{\sigma}.$$

Now we apply the inequality $\sqrt{a + b} \leq \sqrt{a} + \sqrt{b}$ to the Frobenius norm term:

$$\left( \sum_{p=1}^m \sigma_p^2 \right)^{1/2} \leq \sum_{p=1}^m \sqrt{\sigma_p^2}.$$

This separates the sum into individual terms:

$$\sum_{p=1}^m \int_{\mathbb{R}^m_+} \sigma_p \cdot \left( \frac{1}{\prod_{i=1}^m \sqrt{1 + \frac{\sigma_i^2}{km}}} \right)^k \prod_{i=1}^m \sigma_i^{2(m-i)} d\boldsymbol{\sigma}.$$

Using Fubini's Theorem, the integral separates into a sum over $m$ products of individual integrals:

$$\sum_{p=1}^m \prod_{\substack{i=1 \\ i \neq p}}^m \int_0^\infty \left( \frac{1}{\sqrt{1 + \frac{\sigma_i^2}{km}}} \right)^k \sigma_i^{2(m-i)} d\sigma_i \cdot \int_0^\infty \left( \frac{1}{\sqrt{1 + \frac{\sigma_i^2}{km}}} \right)^k \sigma_p^{2(m-p)+1} d\sigma_p.$$

This decomposition allows the integral to be expressed as a sum of $m$ factorized integrals, each involving a single variable.

We now perform for each $1 \leq i \leq m$ the change of variable $x_i = \frac{\sigma_i}{\sqrt{km}}$, which gives $dx_i = \frac{d\sigma_i}{\sqrt{km}}$ and overall

$$M_m \sum_{\substack{p=1}}^m k^{\frac{m}{2}} k^{\sum_{i=1}^m (m-i)} k^{\frac{1}{2}} \prod_{\substack{i=1 \\ i \neq p}}^m \int_0^\infty \left( \frac{1}{\sqrt{1+x_i^2}} \right)^k x_i^{2(m-i)} dx_i \cdot \int_0^\infty \left( \frac{1}{\sqrt{1+x_p^2}} \right)^k x_p^{2(m-p)+1} dx_p$$

$$= M_m \sum_{p=1}^m k^{\frac{m^2}{2}+\frac{1}{2}} \prod_{\substack{i=1 \\ i \neq p}}^m \int_0^\infty \left( \frac{1}{\sqrt{1+x_i^2}} \right)^k x_i^{2(m-i)} dx_i \cdot \int_0^\infty \left( \frac{1}{\sqrt{1+x_p^2}} \right)^k x_p^{2(m-p)+1} dx_p \,,$$

where we have again absorbed the multiplicative dependence on $m$ (which remains constant throughout our analysis) into a constant $M_m$.

We now perform another change of variables $x_i^2 = y_i$, which gives $2x_i dx_i = dy_i$. Overall for $i \neq p$ we get a factor of

$$\frac{1}{2} \int_0^\infty \frac{1}{(1+y_i)^{\frac{k}{2}}} y_i^{m-i-\frac{1}{2}} dy_i$$

and for $i = p$ a factor of

$$\frac{1}{2} \int_0^\infty \frac{1}{(1+y_p)^{\frac{k}{2}}} y_i^{m-p} dy_i \,.$$

It remains to examine the asymptotics of these expressions for large $k$. To do this we note that by the definition of the Beta function (Definition 12), we have that

$$\int_0^\infty \frac{1}{(1+y_i)^{\frac{k}{2}}} y_i^{m-i-\frac{1}{2}} dy_i = B\left( m - i + \frac{1}{2}, \frac{k}{2} - \left( m - i + \frac{1}{2} \right) \right)$$

and by Lemma 37 we have

$$B\left( m - i + \frac{1}{2}, \frac{k}{2} - \left( m - i + \frac{1}{2} \right) \right) = \frac{\Gamma\left( m - i + \frac{1}{2} \right) \Gamma\left( \frac{k}{2} - \left( m - i + \frac{1}{2} \right) \right)}{\Gamma\left( \frac{k}{2} \right)} \,,$$

where $\Gamma(\cdot)$ is the Gamma function (Definition 10). By Lemma 38 the above is of order $k^{-(m-i+\frac{1}{2})}$. The same calculation gives for $i = p$ a term of order $k^{-(m-p+1)}$. Summing these terms we get that the product is of order

$$k^{-\left( \frac{m}{2} + \frac{m(m-1)}{2} + \frac{1}{2} \right)} = k^{-\left( \frac{m^2}{2} + \frac{1}{2} \right)} \,.$$

Overall, the terms dependent on $k$ cancel and each of the $m$ integrals remain bounded as $k \to \infty$, as required. $\qquad \square$

We summarize the above discussion by the following Lemma.

**Lemma 17.** *Let $W_1, (W_2)^\top \in \mathbb{R}^{m,k}$ be matrices with entries drawn independently from $\mathcal{N}(0,1)$, and let $W = \frac{1}{\sqrt{mk}} W_1 W_2$. For any bounded set $K \subseteq B(0,R) \subseteq \mathbb{R}^{m,m}$ we have*

$$\lim_{k \to \infty} \frac{\mathcal{P}(W \in K)}{\mathcal{P}(W_{\mathrm{iid}} \in K)} \to 1,$$

*where $W_{\mathrm{iid}} \in \mathbb{R}^{m,m}$ is a matrix with entries drawn independently from $N(0, \frac{1}{m})$, and furthermore this convergence is uniform over all subsets $K \subseteq B(0,R)$.*

*Proof.* By Theorem 11, $W$ converges as $k \to \infty$ to $W_{\mathrm{iid}}$ in total variation distance, and hence also in distribution. Combining this with Lemmas 13 and 16 implies that the conditions of Theorem 10 are satisfied. Clearly the limiting density, being a product of Gaussian densities, is smooth and positive. Hence the conclusion is a consequence of Lemma 14. $\qquad \square$

## C.3 Approximation by Conditional Probabilities of Bounded Sets

We would ultimately like to bound conditional probabilities involving the events $\{\mathcal{L}_{\text{gen}}(W) < \epsilon_{\text{gen}}\}$ and $\{\mathcal{L}_{\text{train}}(W) < \epsilon_{\text{train}}\}$. Unfortunately, Lemma 14 applies only to bounded subsets of $R^{m,m}$, and the events above are unbounded. We will circumvent this difficulty by intersecting $\{\mathcal{L}_{\text{gen}}(W) < \epsilon_{\text{gen}}\}$ with $B(0, R)$, for sufficiently large $R$, and arguing that the approximations thus obtained are sufficiently precise. Specifically, we have the following Lemma.

**Lemma 18.** *Let $B_R(V)$ be the event that a random matrix $V$ is contained in an open ball of radius $R$ around the origin*

$$B_R(V) := \{V \in B(0, R)\}$$

*It holds that*

$$\lim_{R \to \infty} \sup_{k \in \mathbb{N}, \epsilon > 0} |\mathcal{P}\left(B_R(W) \mid \mathcal{L}_{\text{train}}(W) < \epsilon_{\text{train}}\right) - 1| = 0.$$

*Proof.* To see this, we first note that by the law of total probability

$$\mathcal{P}\left(B_R(W) \mid \mathcal{L}_{\text{train}}(W) < \epsilon_{\text{train}}\right)$$
$$= \int_{\mathbb{R}^{k,m}} \mathcal{P}\left(B_R(W) \mid \mathcal{L}_{\text{train}}(W) < \epsilon_{\text{train}}, W_2\right) f\left(W_2 \mid \mathcal{L}_{\text{train}}(W) < \epsilon_{\text{train}}\right) dW_2,$$

where $f(W_2 \mid \mathcal{L}_{\text{train}}(W) < \epsilon_{\text{train}})$ is the conditional density of $W_2$ given $\mathcal{L}_{\text{train}}(W) < \epsilon_{\text{train}}$. Now note that given $W_2$ we have that $W = \frac{1}{\sqrt{mk}} W_1 W_2$ is a zero-centered Gaussian random variable (with a covariance matrix which depends on $W_2$). Furthermore the set $\{W_1 : \mathcal{L}_{\text{train}}(W) < \epsilon_{\text{train}}\}$ is convex and since $W^* = 0$ it is also symmetric. Furthermore, the set $B(0, R)$ is convex and symmetric for any $R$. We can therefore apply the Gaussian Correlation inequality (Lemma 39) to conclude that

$$\mathcal{P}(B_R(W) \mid \mathcal{L}_{\text{train}}(W) < \epsilon_{\text{train}}, W_2) \geq \mathcal{P}(B_R(W) \mid W_2)$$

and therefore

$$\mathcal{P}\left(B_R(W) \mid \mathcal{L}_{\text{train}}(W) < \epsilon_{\text{train}}\right) \geq \int_{\mathbb{R}^{k,m}} \mathcal{P}\left(B_R(W) \mid W_2\right) f\left(W_2 \mid \mathcal{L}_{\text{train}}(W) < \epsilon_{\text{train}}\right) dW_2. \quad (8)$$

Let $W_{2,:j}$ be the $j$th column of $W_2$. Consider the following set:

$$\hat{R}(c) := \left\{W_2 \ : \ \forall i \in [m], \ \|W_{2,:j}\|_2 \leq c\sqrt{k}\right\}.$$

This set is convex and symmetric, so we can again apply the law of total probability by conditioning on $W_1$ to get

$$\mathcal{P}\left(W_2 \in \hat{R}(c) \mid \mathcal{L}_{\text{train}}(W) < \epsilon_{\text{train}}\right)$$
$$= \int_{\mathbb{R}^{m,k}} \mathcal{P}\left(W_2 \in \hat{R}(c) \mid \mathcal{L}_{\text{train}}(W) < \epsilon_{\text{train}}, W_1\right) f\left(W_1 \mid \mathcal{L}_{\text{train}}(W) < \epsilon_{\text{train}}\right) dW_1$$

Given $W_1$, the set $\{\mathcal{L}_{\text{train}}(W) < \epsilon_{\text{train}}\}$ is again convex and symmetric, and $W = \frac{1}{\sqrt{mk}} W_1 W_2$ is a zero-centered Gaussian, so again by the Gaussian Correlation inequality (Lemma 39) we have

$$\mathcal{P}\left(W_2 \in \hat{R}(c) \mid \mathcal{L}_{\text{train}}(W) < \epsilon_{\text{train}}\right)$$
$$\geq \int_{\mathbb{R}^{m,k}} \mathcal{P}\left(W_2 \in \hat{R}(c) \mid W_1\right) f\left(W_1 \mid \mathcal{L}_{\text{train}}(W) < \epsilon_{\text{train}}\right) dW_1$$
$$= \mathcal{P}\left(W_2 \in \hat{R}(c)\right) \int_{\mathbb{R}^{m,k}} f\left(W_1 \mid \mathcal{L}_{\text{train}}(W) < \epsilon_{\text{train}}\right) dW_1$$
$$= \mathcal{P}\left(W_2 \in \hat{R}(c)\right),$$

where we have used the fact that the event $W_2 \in \hat{R}(c)$ is independent of $W_1$. Overall we therefore obtain that

$$\mathcal{P}\left(W_2 \in \hat{R}(c) \mid \mathcal{L}_{\text{train}}(W) < \epsilon_{\text{train}}\right) \geq \mathcal{P}\left(W_2 \in \hat{R}(c)\right).$$

Now note that by Lemma 40, we can choose $c > 0$ independent of $k$ such that $\mathcal{P}\left(W_2 \in \hat{R}(c)\right)$ is arbitrarily close to 1. We have by Equation (8) that

$$\mathcal{P}\left(B_R(W) \mid \mathcal{L}_{\text{train}}(W) < \epsilon_{\text{train}}\right) \geq \int_{\hat{R}(c)} \mathcal{P}\left(B_R(W) \mid W_2\right) f\left(W_2 \mid \mathcal{L}_{\text{train}}(W) < \epsilon_{\text{train}}\right) dW_2 \, .$$

Now we claim that for $R(\eta)$ sufficiently large, independent of both $\epsilon_{\text{train}}$ and $k$, the integrand can be made to satisfy

$$\mathcal{P}\left(B_{R(\eta)}(W) \mid W_2\right) \geq 1 - \eta$$

for any $W_2 \in \hat{R}(c)$. To do this, note that each entry of $W$ is of the form

$$\frac{1}{\sqrt{mk}} \langle W_{1,i:}, W_{2,:j} \rangle$$

for some $i, j \in [m]$, where $W_{1,i:}$ is the $i$th row of $W_1$. Each row of $W_1$ consists of $k$ independent standard Gaussians, and by assumption $W_{2,:j}$ is a vector of norm $\leq c\sqrt{k}$. Thus we have that the product is a Gaussian variable with zero mean and variance $\|W_{2,:j}\|^2 \leq c^2 k$. Thus after dividing by $\frac{1}{\sqrt{mk}}$ we get a zero-centered Gaussian variable whose variance is independent of $k$. It now follows by a union bound that we can select $R := R(c, m)$ independent of $k$ and $\epsilon_{\text{train}}$ such that the matrix $W$ will lie in $B(0, R)$ with probability larger than $1 - \eta$ whenever $W_2 \in \hat{R}(c)$. Overall we get that

$$\mathcal{P}\left(B_R(W) \mid \mathcal{L}_{\text{train}}(W) < \epsilon_{\text{train}}\right) \geq (1-\eta)\mathcal{P}\left(W_2 \in \hat{R}(c) \mid \mathcal{L}_{\text{train}}(W) < \epsilon_{\text{train}}\right)$$

$$\geq (1-\eta)\mathcal{P}\left(W_2 \in \hat{R}(c)\right),$$

which can be made arbitrarily close to 1 (by Lemma 40), as required. $\qquad \square$

We would also like to show that the approximation is precise with respect to the measure of the random matrix $W_{\text{iid}}$ which $W$ converges to as $k \to \infty$:

**Lemma 19.** *Let $W_{\text{iid}}$ be a centered Gaussian matrix (Definition 8). It holds that*

$$\lim_{R \to \infty} \frac{\mathcal{P}\left(\mathcal{L}_{\text{train}}(W_{\text{iid}}) < \epsilon_{\text{train}} \cap \mathcal{L}_{\text{gen}}(W_{\text{iid}}) < \epsilon_{\text{gen}} \cap B_R(W_{\text{iid}})\right)}{\mathcal{P}\left(\mathcal{L}_{\text{train}}(W_{\text{iid}}) < \epsilon_{\text{train}} \cap B_R(W_{\text{iid}})\right)} = \mathcal{P}\left(\mathcal{L}_{\text{gen}}(W_{\text{iid}}) < \epsilon_{\text{gen}}\right).$$

*Furthermore, this convergence is uniform with respect to $\epsilon_{\text{train}}, \epsilon_{\text{gen}}$.*

*Proof.* Since $B_R$, $\{\mathcal{L}_{\text{train}}(W) < \epsilon_{\text{train}}\}$ and $\{\mathcal{L}_{\text{gen}}(W) < \epsilon_{\text{gen}}\}$ are convex and symmetric, we can apply the Gaussian Correlation inequality (Lemma 39) to the numerator to obtain that for all $R$

$$\frac{\mathcal{P}\left(\mathcal{L}_{\text{train}}(W_{\text{iid}}) < \epsilon_{\text{train}} \cap \mathcal{L}_{\text{gen}}(W_{\text{iid}}) < \epsilon_{\text{gen}} \cap B_R(W_{\text{iid}})\right)}{\mathcal{P}\left(\mathcal{L}_{\text{train}}(W_{\text{iid}}) < \epsilon_{\text{train}} \cap B_R(W_{\text{iid}})\right)}$$

$$\geq \frac{\mathcal{P}\left(\mathcal{L}_{\text{train}}(W_{\text{iid}}) < \epsilon_{\text{train}} \cap B_R(W_{\text{iid}})\right) \mathcal{P}\left(\mathcal{L}_{\text{gen}}(W_{\text{iid}}) < \epsilon_{\text{gen}}\right)}{\mathcal{P}\left(\mathcal{L}_{\text{train}}(W_{\text{iid}}) < \epsilon_{\text{train}} \cap B_R(W_{\text{iid}})\right)}$$

$$= \mathcal{P}\left(\mathcal{L}_{\text{gen}}(W_{\text{iid}}) < \epsilon_{\text{gen}}\right),$$

hence the same inequality holds in the limit. On the other hand, by applying the Gaussian Correlation inequality to the denominator we get

$$\frac{\mathcal{P}\left(\mathcal{L}_{\text{train}}(W_{\text{iid}}) < \epsilon_{\text{train}} \cap \mathcal{L}_{\text{gen}}(W_{\text{iid}}) < \epsilon_{\text{gen}} \cap B_R(W_{\text{iid}})\right)}{\mathcal{P}\left(\mathcal{L}_{\text{train}}(W_{\text{iid}}) < \epsilon_{\text{train}} \cap B_R(W_{\text{iid}})\right)}$$

$$\leq \frac{\mathcal{P}\left(\mathcal{L}_{\text{train}}(W_{\text{iid}}) < \epsilon_{\text{train}} \cap \mathcal{L}_{\text{gen}}(W_{\text{iid}}) < \epsilon_{\text{gen}} \cap B_R(W_{\text{iid}})\right)}{\mathcal{P}\left(\mathcal{L}_{\text{train}}(W_{\text{iid}}) < \epsilon_{\text{train}}\right) \mathcal{P}\left(B_R(W_{\text{iid}})\right)}$$

$$\leq \frac{\mathcal{P}\left(\mathcal{L}_{\text{train}}(W_{\text{iid}}) < \epsilon_{\text{train}} \cap \mathcal{L}_{\text{gen}}(W_{\text{iid}}) < \epsilon_{\text{gen}}\right)}{\mathcal{P}\left(\mathcal{L}_{\text{train}}(W_{\text{iid}}) < \epsilon_{\text{train}}\right) \mathcal{P}\left(B_R(W_{\text{iid}})\right)}$$

$$= \frac{\mathcal{P}\left(\mathcal{L}_{\text{train}}(W_{\text{iid}}) < \epsilon_{\text{train}}\right) \mathcal{P}\left(\mathcal{L}_{\text{gen}}(W_{\text{iid}}) < \epsilon_{\text{gen}}\right)}{\mathcal{P}\left(\mathcal{L}_{\text{train}}(W_{\text{iid}}) < \epsilon_{\text{train}}\right) \mathcal{P}\left(B_R(W_{\text{iid}})\right)}$$

$$= \frac{\mathcal{P}\left(\mathcal{L}_{\text{gen}}(W_{\text{iid}}) < \epsilon_{\text{gen}}\right)}{\mathcal{P}\left(B_R(W_{\text{iid}})\right)},$$

where the second inequality follows by basic probability properties, and the penultimate equality follows by the independence of $\mathcal{L}_{\text{train}}(W_{\text{iid}}) < \epsilon_{\text{train}}$ and $\mathcal{L}_{\text{gen}}(W_{\text{iid}}) < \epsilon_{\text{gen}}$. The ratio $\frac{\mathcal{P}(\mathcal{L}_{\text{gen}}(W_{\text{iid}}) < \epsilon_{\text{gen}})}{\mathcal{P}(B_R(W_{\text{iid}}))}$ tends to $\mathcal{P}\left(\mathcal{L}_{\text{gen}}(W_{\text{iid}}) < \epsilon_{\text{gen}}\right)$ as $R \to \infty$, hence the proof is complete. □

## C.4 Proof of Sought-After Result

We are now ready to prove Theorem 9. We assume WLOG that $W^* = 0$ as all claims are invariant to a mean shift. First note that we have

$$\lim_{k \to \infty} \mathcal{P}(\mathcal{L}_{\text{gen}}(W) < \epsilon_{\text{gen}}) = \mathcal{P}(\mathcal{L}_{\text{gen}}(W_{\text{iid}}) < \epsilon_{\text{gen}})$$

where $W_{\text{iid}}$ is a centered Gaussian matrix (Definition 8), hence it suffices to show that

$$\lim_{k \to \infty} \sup_{\epsilon_{\text{train}} > 0} \mathcal{P}\left(\mathcal{L}_{\text{gen}}(W) < \epsilon_{\text{gen}} \mid \mathcal{L}_{\text{train}}(W) < \epsilon_{\text{train}}\right) - \mathcal{P}\left(\mathcal{L}_{\text{gen}}(W_{\text{iid}}) < \epsilon_{\text{gen}}\right) = 0 .$$

First, let $\eta_1, \eta_2 > 0$. We can choose a radius $R := R(\eta_1, \eta_2) > 0$ such that both

$$\sup_{k \in \mathbb{N}, \epsilon_{\text{train}} > 0} |P(B_R(W) \mid \mathcal{L}_{\text{train}}(W) < \epsilon_{\text{train}}) - 1| < \eta_1$$

and

$$\left| \frac{\mathcal{P}\left(\mathcal{L}_{\text{train}}(W_{\text{iid}}) < \epsilon_{\text{train}} \cap \mathcal{L}_{\text{gen}}(W_{\text{iid}}) < \epsilon_{\text{gen}} \cap B_R(W_{\text{iid}})\right)}{\mathcal{P}\left(\mathcal{L}_{\text{train}}(W_{\text{iid}}) < \epsilon_{\text{train}} \cap B_R(W_{\text{iid}})\right)} - \mathcal{P}\left(\mathcal{L}_{\text{gen}}(W_{\text{iid}}) < \epsilon_{\text{gen}}\right) \right| < \eta_2 .$$

Note that such an $R$ exists by Lemma 18 and Lemma 19. Then, we rewrite the conditional probability using the law of total probability by conditioning on the events that $W$ is within $B(0, R)$ and its complement:

$$\mathcal{P}\left(\mathcal{L}_{\text{gen}}(W) < \epsilon_{\text{gen}} \mid \mathcal{L}_{\text{train}}(W) < \epsilon_{\text{train}}\right)$$
$$= \mathcal{P}\left(\mathcal{L}_{\text{gen}}(W) < \epsilon_{\text{gen}} \mid \mathcal{L}_{\text{train}}(W) < \epsilon_{\text{train}}, B_R(W)\right) \mathcal{P}\left(B_R(W) \mid \mathcal{L}_{\text{train}}(W) < \epsilon_{\text{train}}\right)$$
$$+ \mathcal{P}\left(\mathcal{L}_{\text{gen}}(W) < \epsilon_{\text{gen}} \mid \mathcal{L}_{\text{train}}(W) < \epsilon_{\text{train}}, B_R(W)^C\right) \mathcal{P}\left(B_R(W)^C \mid \mathcal{L}_{\text{train}}(W) < \epsilon_{\text{train}}\right) .$$

By the choice of $R$ and the triangle inequality we have

$$\left| \mathcal{P}\left(\mathcal{L}_{\text{gen}}(W) < \epsilon_{\text{gen}} \mid \mathcal{L}_{\text{train}}(W) < \epsilon_{\text{train}}\right) - \right.$$
$$\left. \mathcal{P}\left(\mathcal{L}_{\text{gen}}(W) < \epsilon_{\text{gen}} \mid \mathcal{L}_{\text{train}}(W) < \epsilon_{\text{train}}, B_R(W)\right) \right| < 2\eta_1 .$$

Next, note that

$$\mathcal{P}\left(\mathcal{L}_{\text{gen}}(W) < \epsilon_{\text{gen}} \mid \mathcal{L}_{\text{train}}(W) < \epsilon_{\text{train}}, B_R(W)\right)$$
$$= \frac{\mathcal{P}\left(\mathcal{L}_{\text{train}}(W_{\text{iid}}) < \epsilon_{\text{train}} \cap \mathcal{L}_{\text{gen}}(W_{\text{iid}}) < \epsilon_{\text{gen}} \cap B_R(W_{\text{iid}})\right)}{\mathcal{P}\left(\mathcal{L}_{\text{train}}(W_{\text{iid}}) < \epsilon_{\text{train}} \cap B_R(W_{\text{iid}})\right)} .$$

By Lemma 14, we can divide and multiply by the corresponding probabilities obtained with respect to the matrix $W_{\text{iid}}$ which we converge to as $k \to \infty$, and rewrite this ratio as

$$\frac{\mathcal{P}\left(\mathcal{L}_{\text{train}}(W) < \epsilon_{\text{train}} \cap \mathcal{L}_{\text{gen}}(W) < \epsilon_{\text{gen}} \cap B_R(W)\right)}{\mathcal{P}\left(\mathcal{L}_{\text{train}}(W) < \epsilon_{\text{train}} \cap B_R(W)\right)}$$
$$= \frac{\mathcal{P}\left(\mathcal{L}_{\text{train}}(W_{\text{iid}}) < \epsilon_{\text{train}} \cap \mathcal{L}_{\text{gen}}(W_{\text{iid}}) < \epsilon_{\text{gen}} \cap B_R(W_{\text{iid}})\right)}{\mathcal{P}\left(\mathcal{L}_{\text{train}}(W_{\text{iid}}) < \epsilon_{\text{train}} \cap B_R(W_{\text{iid}})\right)} \cdot \alpha(k, R)$$

where $\alpha(k, R) \to 1$ as $k \to \infty$ (uniformly in $\epsilon_{\text{train}}, \epsilon_{\text{gen}}$). Again by the choice of $R$ we have that

$$\left| \frac{\mathcal{P}\left(\mathcal{L}_{\text{train}}(W) < \epsilon_{\text{train}} \cap \mathcal{L}_{\text{gen}}(W) < \epsilon_{\text{gen}} \cap B_R(W)\right)}{\mathcal{P}\left(\mathcal{L}_{\text{train}}(W) < \epsilon_{\text{train}} \cap B_R(W)\right)} - \mathcal{P}\left(\mathcal{L}_{\text{gen}}(W_{\text{iid}}) < \epsilon_{\text{gen}}\right) \right| < \eta_2 .$$

Overall we obtain that for any $\epsilon_{\text{train}} > 0$

$$\limsup_{k \to \infty} \mathcal{P}\left(\mathcal{L}_{\text{gen}}(W) < \epsilon_{\text{gen}} \mid \mathcal{L}_{\text{train}}(W) < \epsilon_{\text{train}}\right)$$
$$\leq 2\eta_1 + \lim_{k \to \infty} \mathcal{P}\left(\mathcal{L}_{\text{gen}}(W) < \epsilon_{\text{gen}} \mid \mathcal{L}_{\text{train}}(W) < \epsilon_{\text{train}}, B_R(W)\right)$$
$$\leq 2\eta_1 + \eta_2 + \mathcal{P}\left(\mathcal{L}_{\text{gen}}(W_{\text{iid}}) < \epsilon_{\text{gen}}\right) .$$

Since the above holds for all $\eta_1, \eta_2 > 0$ we obtain that

$$\limsup_{k \to \infty} \mathcal{P}\left(\mathcal{L}_{\text{gen}}(W) < \epsilon_{\text{gen}} \mid \mathcal{L}_{\text{train}}(W) < \epsilon_{\text{train}}\right) \leq \mathcal{P}\left(\mathcal{L}_{\text{gen}}(W_{\text{iid}}) < \epsilon_{\text{gen}}\right).$$

A symmetric argument applied to $\liminf_{k \to \infty} \mathcal{P}\left(\mathcal{L}_{\text{gen}}(W) < \epsilon_{\text{gen}} \mid \mathcal{L}_{\text{train}}(W) < \epsilon_{\text{train}}\right)$ implies that

$$\liminf_{k \to \infty} \mathcal{P}\left(\mathcal{L}_{\text{gen}}(W) < \epsilon_{\text{gen}} \mid \mathcal{L}_{\text{train}}(W) < \epsilon_{\text{train}}\right) \geq \mathcal{P}\left(\mathcal{L}_{\text{gen}}(W_{\text{iid}}) < \epsilon_{\text{gen}}\right)$$

and hence

$$\lim_{k \to \infty} \mathcal{P}\left(\mathcal{L}_{\text{gen}}(W) < \epsilon_{\text{gen}} \mid \mathcal{L}_{\text{train}}(W) < \epsilon_{\text{train}}\right) = \mathcal{P}\left(\mathcal{L}_{\text{gen}}(W_{\text{iid}}) < \epsilon_{\text{gen}}\right).$$

Since the above holds uniformly in $\epsilon > 0$ we conclude that

$$\lim_{k \to \infty} \sup_{\epsilon_{\text{train}} > 0} \mathcal{P}\left(\mathcal{L}_{\text{gen}}(W) < \epsilon_{\text{gen}} \mid \mathcal{L}_{\text{train}}(W) < \epsilon_{\text{train}}\right) - \mathcal{P}\left(\mathcal{L}_{\text{gen}}(W) < \epsilon_{\text{gen}}\right) = 0$$

as required. $\qquad\square$

## D   Increasing Depth with Ground Truth Matrices of Higher Rank

Theorem 2 applies to cases where the ground truth matrix $W^*$ has rank equal to one. In this appendix we discuss an extension of Theorem 2 to ground truth matrices of higher rank. A complete formalization of this extension is left for future work.

Suppose the ground truth matrix $W^*$ has arbitrary rank $r$ and is non-degenerate, in the sense that its $r$ non-zero singular values have similar orders of magnitude. Suppose the matrix factorization is square, *i.e.*, its dimensions $m$ and $m'$ and its width $k$ are all equal. Finally, suppose temporarily that the probability distribution over weight settings $\mathcal{P}(\cdot)$ does not include normalization, *i.e.*, that $\mathcal{P}(\cdot)$ is generated by $\mathcal{Q}(\cdot)$ without normalization (Definition 3).

Hanin and Paouris [46] proved that, as the depth $d$ of the matrix factorization tends to infinity, drawing its weight setting from $\mathcal{P}(\cdot)$ leads the *Lyapunov exponents* of the factorized matrix $W$ (Equation (4)) to converge in distribution to independent Gaussians with means $0 > \mu_1 > \mu_2 > \cdots > \mu_m$ and variances $O(d^{-1})$, where the Lyapunov exponents of $W$ are respectively defined as $\lambda_i := \log(\sigma_i)/d$ for $i \in [m]$, with $\sigma_1 \geq \cdots \geq \sigma_m \geq 0$ standing for the singular values of $W$. Assume that $d$ is large enough for the distribution of $\lambda_1, \ldots, \lambda_m$ to be well approximated by its limit. In particular, assume that $\boldsymbol{\lambda} \sim \mathcal{N}(\cdot\,; \boldsymbol{\mu}, d^{-1}I)$, where $\boldsymbol{\lambda} := (\lambda_1, \ldots, \lambda_m)$, $\boldsymbol{\mu} := (\mu_1, \ldots, \mu_m)$ and $I \in \mathbb{R}^{m,m}$ is an identity matrix.

Let $j \in [m]$. We now estimate the probability that the $j$ leading singular values of $W$, *i.e.*, $\sigma_1, \ldots, \sigma_j$, have similar orders of magnitude. Per the definition of the Lyapunov exponents of $W$ (namely, $\lambda_i := \log(\sigma_i)/d$ for $i \in [m]$), this takes place when $\max_{i,i' \in [j]} |\lambda_i - \lambda_{i'}| \leq \Delta$ for some small $\Delta \in \mathbb{R}_{>0}$. By our assumption on the distribution of $\lambda_1, \ldots, \lambda_m$, the probability of the latter event is:

$$\mathcal{P}\left(\max_{i,i' \in [j]} |\lambda_i - \lambda_{i'}| \leq \Delta\right) = \int_{\mathcal{C}_{j,\Delta}} \left(\frac{1}{2\pi d^{-1}}\right)^{j/2} e^{-d\|\boldsymbol{\lambda}_{:j} - \boldsymbol{\mu}_{:j}\|^2} d\boldsymbol{\lambda}_{:j},$$

where $\boldsymbol{\lambda}_{:j} := (\lambda_1, \ldots, \lambda_j)$, $\boldsymbol{\mu}_{:j} := (\mu_1, \ldots, \mu_j)$ and:

$$\mathcal{C}_{j,\Delta} := \{\boldsymbol{\lambda}_{:j} \mid \max_{i,i' \in [j]} |\lambda_i - \lambda_{i'}| \leq \Delta\}. \tag{9}$$

For large $d$, the integral above is dominated by the maximum of the integrand. More precisely, we may apply Laplace's principle and obtain:

$$\int_{\mathcal{C}_{j,\Delta}} \left(\frac{1}{2\pi d^{-1}}\right)^{j/2} e^{-d\|\boldsymbol{\lambda}_{:j} - \boldsymbol{\mu}_{:j}\|^2} d\boldsymbol{\lambda}_{:j} \approx e^{-d \inf_{\boldsymbol{\lambda}_{:j} \in \mathcal{C}_{j,\Delta}} \|\boldsymbol{\lambda}_{:j} - \boldsymbol{\mu}_{:j}\|^2 + o(d)}.$$

We conclude that:

$$\mathcal{P}\left(\sigma_1, \ldots, \sigma_j \text{ have similar orders of magnitude}\right) \approx e^{-d \inf_{\boldsymbol{\lambda}_{:j} \in \mathcal{C}_{j,\Delta}} \|\boldsymbol{\lambda}_{:j} - \boldsymbol{\mu}_{:j}\|^2 + o(d)}. \tag{10}$$

Scaling $W$ by a positive factor does not change whether $\sigma_1, \ldots, \sigma_j$ have similar orders of magnitude. Therefore, Equation (10) carries over to the case where $\mathcal{P}(\cdot)$ includes normalization. We hereafter consider this case.

We would like to show that the probability of high generalization loss (Equation (3)) conditioned on low training loss (Equation (2)), namely

$$\mathcal{P}\big(\mathcal{L}_{\text{gen}}(W) \geq \epsilon_{\text{train}}c \,\big|\, \mathcal{L}_{\text{train}}(W) < \epsilon_{\text{train}}\big) = \frac{\mathcal{P}\big(\mathcal{L}_{\text{gen}}(W) \geq \epsilon_{\text{train}}c \,\wedge\, \mathcal{L}_{\text{train}}(W) < \epsilon_{\text{train}}\big)}{\mathcal{P}\big(\mathcal{L}_{\text{train}}(W) < \epsilon_{\text{train}}\big)}, \quad (11)$$

diminishes as the depth $d$ increases. The probability of low training loss—denominator in the right-hand side of Equation (11)—is lower bounded by the probability that $W$ is close (in Frobenius norm) to the ground truth matrix $W^*$. By isotropy and normalization, the latter probability should be proportional (identical up to multiplicative factors that do not depend on $d$) to the probability that $W$ has singular values akin to those of $W^*$, *i.e.*, to the probability that the $r$ leading singular values of $W$ have similar orders of magnitude while the following singular value is much smaller. We thus have:

$$\mathcal{P}\big(\mathcal{L}_{\text{train}}(W) < \epsilon_{\text{train}}\big) =$$
$$\Omega\left(\mathcal{P}\big(\sigma_1, \ldots, \sigma_r \text{ have similar orders of magnitude while } \sigma_{r+1} \text{ is much smaller}\big)\right).$$

Invoking Equation (10) with $j = r$ and with $j = r + 1$, we obtain:

$$\mathcal{P}\big(\mathcal{L}_{\text{train}}(W) < \epsilon_{\text{train}}\big) =$$
$$\Omega\left(e^{-d\inf_{\boldsymbol{\lambda}_{:r}\in\mathcal{C}_{r,\Delta}} \|\boldsymbol{\lambda}_{:r} - \boldsymbol{\mu}_{:r}\|^2 + o(d)} - e^{-d\inf_{\boldsymbol{\lambda}_{:r+1}\in\mathcal{C}_{r+1,\Delta}} \|\boldsymbol{\lambda}_{:r+1} - \boldsymbol{\mu}_{:r+1}\|^2 + o(d)}\right). \quad (12)$$

Moving to the numerator in the right-hand side of Equation (11), due to the measurement matrices satisfying an RIP (Definition 1), high generalization loss ($\mathcal{L}_{\text{gen}}(W) \geq \epsilon_{\text{train}}c$) in conjunction with low training loss ($\mathcal{L}_{\text{train}}(W) < \epsilon_{\text{train}}$) necessarily implies that the effective rank of $W$ is higher than $r$, meaning its $r + 1$ leading singular values have similar orders of magnitude. We thus have:

$$\mathcal{P}\big(\mathcal{L}_{\text{gen}}(W) \geq \epsilon_{\text{train}}c \,\wedge\, \mathcal{L}_{\text{train}}(W) < \epsilon_{\text{train}}\big) =$$
$$O\left(\mathcal{P}\big(\sigma_1, \ldots, \sigma_{r+1} \text{ have similar orders of magnitude}\big)\right) = \quad (13)$$
$$O\left(e^{-d\inf_{\boldsymbol{\lambda}_{:r+1}\in\mathcal{C}_{r+1,\Delta}} \|\boldsymbol{\lambda}_{:r+1} - \boldsymbol{\mu}_{:r+1}\|^2 + o(d)}\right),$$

where we invoked Equation (10) with $j = r + 1$.

Plugging Equations (12) and (13) into Equation (11) yields:

$$\mathcal{P}\big(\mathcal{L}_{\text{gen}}(W) \geq \epsilon_{\text{train}}c \,\big|\, \mathcal{L}_{\text{train}}(W) < \epsilon_{\text{train}}\big) =$$
$$O\left(\frac{e^{-d\inf_{\boldsymbol{\lambda}_{:r+1}\in\mathcal{C}_{r+1,\Delta}} \|\boldsymbol{\lambda}_{:r+1} - \boldsymbol{\mu}_{:r+1}\|^2 + o(d)}}{e^{-d\inf_{\boldsymbol{\lambda}_{:r}\in\mathcal{C}_{r,\Delta}} \|\boldsymbol{\lambda}_{:r} - \boldsymbol{\mu}_{:r}\|^2 + o(d)} - e^{-d\inf_{\boldsymbol{\lambda}_{:r+1}\in\mathcal{C}_{r+1,\Delta}} \|\boldsymbol{\lambda}_{:r+1} - \boldsymbol{\mu}_{:r+1}\|^2 + o(d)}}\right). \quad (14)$$

Recalling the definition of $\mathcal{C}_{r,\Delta}$ and $\mathcal{C}_{r+1,\Delta}$ (Equation (9)), as well as the fact that $\mu_1 > \cdots > \mu_{r+1}$, we have that for sufficiently small $\Delta$:

$$\inf_{\boldsymbol{\lambda}_{:r+1}\in\mathcal{C}_{r+1,\Delta}} \|\boldsymbol{\lambda}_{:r+1} - \boldsymbol{\mu}_{:r+1}\|^2 > \inf_{\boldsymbol{\lambda}_{:r}\in\mathcal{C}_{r,\Delta}} \|\boldsymbol{\lambda}_{:r} - \boldsymbol{\mu}_{:r}\|^2.$$

Together with Equation (14), this implies the desired result: the probability of high generalization loss conditioned on low training loss diminishes as the depth $d$ increases.

## E  Auxiliary Theorems, Lemmas and Definitions

In this appendix we provide additional theorems, lemmas and definitions used throughout our proofs.

**Definition 8.** Let $W \in \mathbb{R}^{m,m'}$ be a random matrix. We say that $W$ is a *centered Gaussian matrix* when the entries of $W$ are drawn independently from $\mathcal{N}(0, \nu)$ where $\nu \in \mathbb{R}_{>0}$ is some fixed variance.

**Lemma 20.** *Let $\mathcal{P}(\cdot)$ be some distribution on $\mathbb{R}$ that is symmetric, i.e., if $x \sim \mathcal{P}(\cdot)$ then $-x \sim \mathcal{P}(\cdot)$. Let $f : \mathbb{R} \to \mathbb{R}$ be some anti-symmetric function, i.e.,*

$$\forall x \in \mathbb{R}. \ f(x) = -f(-x).$$

*Then*

$$\mathop{\mathbb{E}}_{x \sim \mathcal{P}(\cdot)} [f(x)] = 0.$$

*Proof.* Since $\mathcal{P}(\cdot)$ is symmetric and $f$ is anti-symmetric, it holds that

$$\mathop{\mathbb{E}}_{x\sim\mathcal{P}(\cdot)}[f(x)] = \mathop{\mathbb{E}}_{-x\sim\mathcal{P}(\cdot)}[f(-x)] = - \mathop{\mathbb{E}}_{-x\sim\mathcal{P}(\cdot)}[f(x)] = - \mathop{\mathbb{E}}_{x\sim\mathcal{P}(\cdot)}[f(-x)] .$$

The claim follows by rearranging. $\qquad\square$

**Lemma 21.** *Let* $f : \mathbb{R} \to \mathbb{R}$ *be some function,* $A \in \mathbb{R}^{m,m'}$ *be some matrix and* $\alpha \in [m']$ *be some index. Denote by* $\mathbf{e}_\alpha \in \mathbb{R}^{m'}$ *the standard basis vector with 1 in its* $\alpha$ *entry and zeros elsewhere. Then*

$$f(A)\mathbf{e}_\alpha = f(A\mathbf{e}_\alpha)$$

*where* $f$ *applied to a matrix or a vector is a shorthand for* $f$ *applied to each entry.*

*Proof.* Observe that

$$f(A)\mathbf{e}_\alpha = \begin{pmatrix} f(A_{11}) & \cdots & f(A_{1m'}) \\ \vdots & \ddots & \vdots \\ f(A_{m1}) & \cdots & f(A_{mm'}) \end{pmatrix} \mathbf{e}_\alpha$$
$$= \begin{pmatrix} f(A_{1\alpha}) \\ \vdots \\ f(A_{m\alpha}) \end{pmatrix}$$
$$= f(A\mathbf{e}_\alpha)$$

as required. $\qquad\square$

**Definition 9.** *The* total variational distance (TV distance) *between two random variables* $X, Y$ *on the same space* $\Omega$ *is defined as*

$$TV(X,Y) = \sup_{A\subseteq\Omega} |\mathcal{P}(X \in A) - \mathcal{P}(Y \in A)|$$

**Lemma 22.** *For any two random variables* $X, Y$ *on the same space* $\Omega$ *and any* $c > 0$ *It holds that*

$$TV(cX, cY) = TV(X,Y)$$

*Proof.* By Definition 9 it holds that

$$TV(X,Y) = \sup_{A\subseteq\Omega} |\mathcal{P}(X \in A) - \mathcal{P}(Y \in A)|$$

For any $A \subseteq \Omega$ denote $c \cdot A := \{c \cdot x | x \in A\}$. Hence the above is equal to

$$\sup_{A\subseteq\Omega} |\mathcal{P}(cX \in c \cdot A) - \mathcal{P}(cY \in c \cdot A)| = TV(cX, cY)$$

as required. $\qquad\square$

**Theorem 11.** *Let* $\{W_j\}_{j\in[d]}$ *be a set of random matrices where for each* $j \in [d]$, $W_j \in \mathbb{R}^{m_{j+1},m_j}$ *where* $m_{d+1} = m$, $m_1 := m'$ *and* $m_j = k$ *for all* $j = 2, \ldots, d$. *Suppose that for each* $j \in [d]$, *the matrix* $W_j$ *is centered Gaussian (Definition 8) with variance* $\frac{1}{m_j}$. *Let* $W = \prod_{j=d}^{1} W_j$ *and let* $W^* \in \mathbb{R}^{m,m'}$ *be a centered Gaussian matrix with variance* $\frac{1}{m'}$. *Assume that* $k \geq m$. *Then*

$$TV(W, W^*) \leq C(d-1)\sqrt{\frac{m \cdot m'}{k}}$$

*for some universal constant* $C > 0$.

*Proof.* Theorem 1 in Li and Woodruff [71] states that for random matrices $U_j \in \mathbb{R}^{m_{j+1},m_j}, j \in [d]$ where $m_{d+1} = m, m_1 = m'$ and $m_j = k, j = 2, \ldots, d$ with entries drawn independently from $\mathcal{N}(0,1)$, and a random matrix $U \in \mathbb{R}^{m,m'}$ with entries drawn independently from $\mathcal{N}(0,1)$, it holds that

$$TV(\frac{1}{\sqrt{k}}U, \prod_{j=d}^{1}\frac{1}{\sqrt{k}}U_j) \leq C(d-1)\sqrt{\frac{m \cdot m'}{k}}$$

for some universal constant $C > 0$ such. Per Lemma 22, scaling $\frac{1}{\sqrt{k}}U$ and $\prod_{j=d}^{1} \frac{1}{\sqrt{k}}U_j$ by a factor of $\frac{\sqrt{k}}{\sqrt{m'}}$ preserves the TV distance between the two random variables. Hence,

$$TV(\frac{1}{\sqrt{m'}}U, \prod_{j=d}^{2} \frac{1}{\sqrt{k}}U_j \cdot \frac{1}{\sqrt{m'}}U_1) = TV(\frac{1}{\sqrt{k}}U, \prod_{j=d}^{1} \frac{1}{\sqrt{k}}U_j) \leq C(d-1)\sqrt{\frac{m \cdot m'}{k}}.$$

The proof concludes by noting that $W = \prod_{j=d}^{2} \frac{1}{\sqrt{k}}U_j \cdot \frac{1}{\sqrt{m'}}U_1$ and $W^* = \frac{1}{\sqrt{m'}}U.$ $\qquad\square$

**Lemma 23.** *Let $p : \mathbb{R}^d \to \mathbb{R}$ be some polynomial. The zero set of $p$,*

$$\{\mathbf{x} \in \mathbb{R}^d : p(\mathbf{x}) = 0\},$$

*is either $\mathbb{R}^d$ or has Lebesgue measure zero.*

*Proof.* See Caron and Traynor [20]. $\qquad\square$

**Lemma 24.** *For any two matrices $A \in \mathbb{R}^{m,n}$ and $B \in \mathbb{R}^{n,p}$ it holds that*

$$\|AB\|_F \leq \|A\|_F \|B\|_F$$

*Proof.* This is a clssical result that follows from the Cauchy-Schwarz inequality. $\qquad\square$

**Lemma 25.** *For any centered gaussian matrix $X \in \mathbb{R}^{p,q}$ (Definition 8) there exists a sufficiently large constant $N \in \mathbb{R}_{>0}$ and a constant $c_{10} \in \mathbb{R}_{>0}$ depdendent on $p, q$ such that with probability at least $1 - e^{-c_{10}N}$ it holds that*

$$\|X\|_F \leq N.$$

*Proof.* This follows from standard concentration inequalities for $\chi^2$ random variables. $\qquad\square$

**Lemma 26.** *Let $A \in \mathbb{R}^{m,n}$ and $C \in \mathbb{R}^{p,q}$ be matrices with singular value decompositions*

$$A = \sum_{i=1}^{r_A} \sigma_i^A \mathbf{u}_i^A \left(\mathbf{v}_i^A\right)^\top, \quad C = \sum_{j=1}^{r_C} \sigma_j^C \mathbf{u}_j^C \left(\mathbf{v}_j^C\right)^\top.$$

*We denote the the rank one summands of $A$ and $C$ as*

$$X_i = \sigma_i^A \mathbf{u}_i^A \left(\mathbf{v}_i^A\right)^\top, \quad Y_j = \sigma_j^C \mathbf{u}_j^C \left(\mathbf{v}_j^C\right)^\top.$$

*Let $B \in \mathbb{R}^{n,p}$ be a rank one matrix of the form*

$$B = \mathbf{u}_B \mathbf{v}_B^\top.$$

*Then for any $i \in [r_A], j \in [r_C]$ it holds that*

$$\|ABC\|_F^2 \geq \|X_i B Y_j\|_F^2.$$

*Proof.* Substituting the singular value decompositions of $A$ and $C$ into the product, we obtain

$$ABC = \left(\sum_{i=1}^{r_A} X_i\right) B \left(\sum_{j=1}^{r_C} Y_j\right).$$

Expanding the product yields

$$ABC = \sum_{i=1}^{r_A} \sum_{j=1}^{r_C} X_i B Y_j.$$

To show that the terms $X_i B Y_j$ are mutually orthogonal, we compute the Frobenius inner product between two distinct terms:

$$\langle X_i B Y_j, X_{i'} B Y_{j'} \rangle = \mathrm{Tr}\left((X_i B Y_j)^\top (X_{i'} B Y_{j'})\right).$$

Using the definitions
$$X_i = \sigma_i^A \mathbf{u}_i^A \left(\mathbf{v}_i^A\right)^\top, \quad Y_j = \sigma_j^C \mathbf{u}_j^C \left(\mathbf{v}_j^C\right)^\top,$$
the term $X_i B Y_j$ expands as
$$X_i B Y_j = \sigma_i^A \sigma_j^C \left((\mathbf{v}_i^A)^\top \mathbf{u}_B\right) \left(\mathbf{v}_B^\top \mathbf{u}_j^C\right) \mathbf{u}_i^A \left(\mathbf{v}_j^C\right)^\top.$$

Therefore,
$$(X_i B Y_j)^\top = \sigma_i^A \sigma_j^C \left(\left(\mathbf{v}_i^A\right)^\top \mathbf{u}_B\right) \left(\mathbf{v}_B^\top \mathbf{u}_j^C\right) \mathbf{v}_j^C \left(\mathbf{u}_i^A\right)^\top.$$

Likewise,
$$X_{i'} B Y_{j'} = \sigma_{i'}^A \sigma_{j'}^C \left(\left(\mathbf{v}_{i'}^A\right)^\top \mathbf{u}_B\right) \left(\mathbf{v}_B^\top \mathbf{u}_{j'}^C\right) \mathbf{u}_{i'}^A \left(\mathbf{v}_{j'}^C\right)^\top.$$

Substituting into the inner product and factoring out scalars we obtain that
$$\langle X_i B Y_j, X_{i'} B Y_{j'}\rangle =$$
$$\sigma_i^A \sigma_j^C \left(\left(\mathbf{v}_i^A\right)^\top \mathbf{u}_B\right) \left(\mathbf{v}_B^\top \mathbf{u}_j^C\right) \sigma_{i'}^A \sigma_{j'}^C \left(\left(\mathbf{v}_{i'}^A\right)^\top \mathbf{u}_B\right) \left(\mathbf{v}_B^\top \mathbf{u}_{j'}^C\right) \mathrm{Tr}\left(\mathbf{v}_j^C \left(\mathbf{u}_i^A\right)^\top \mathbf{u}_{i'}^A \left(\mathbf{v}_{j'}^C\right)^\top\right).$$

Using the cyclic property of trace,
$$\mathrm{Tr}\left(\mathbf{v}_j^C \left(\mathbf{u}_i^A\right)^\top \mathbf{u}_{i'}^A \left(\mathbf{v}_{j'}^C\right)^\top\right) = \left(\left(\mathbf{v}_{j'}^C\right)^\top \mathbf{v}_j^C\right) \left(\left(\mathbf{u}_i^A\right)^\top \mathbf{u}_{i'}^A\right).$$

Since the singular vectors $\mathbf{u}_i^A, \mathbf{v}_i^A, \mathbf{u}_j^C, \mathbf{v}_j^C$ are orthonormal,
$$\left(\mathbf{u}_i^A\right)^\top \mathbf{u}_{i'}^A = \delta_{ii'}, \quad \left(\mathbf{v}_j^C\right)^\top \mathbf{v}_{j'}^C = \delta_{jj'}.$$

Thus, the trace vanishes whenever $i \neq i'$ or $j \neq j'$. Applying the Pythagorean theorem for the Frobenius norm,
$$\|ABC\|_F^2 = \sum_{i=1}^{r_A} \sum_{j=1}^{r_C} \|X_i B Y_j\|_F^2.$$

Since every term in the sum is non-negative, for any $i \in [r_A], j \in [r_C]$ it holds that
$$\|ABC\|_F^2 \geq \|X_i B Y_j\|_F^2$$
as required. $\qquad\square$

**Lemma 27.** *Let $W \in \mathbb{R}^{m,n}$ be a centered Gaussian matrix (Definition 8) with variance one. Consider the singular value decomposition (SVD) of $W$:*
$$W = U\Sigma V^\top,$$
*where $U \in \mathbb{R}^{m,m}$ and $V \in \mathbb{R}^{n,n}$ are orthogonal matrices, and $\Sigma$ is a diagonal matrix of singular values. Then, the first left singular vector $\mathbf{u}_1$ (the first column of $U$) and the first right singular vector $\mathbf{v}_1$ (the first column of $V$) are uniformly distributed on the unit spheres $S^{m-1}$ and $S^{n-1}$, respectively.*

*Proof.* Since $W$ is an $m, n$ matrix with independent standard normal entries, its distribution is invariant under orthogonal transformations. That is, for any orthogonal matrices $Q \in O(m)$ and $P \in O(n)$, the distribution of $W$ satisfies
$$QWP \overset{d}{=} W.$$

This follows from the fact that a Gaussian matrix remains Gaussian after orthogonal transformations, and the standard normal distribution is rotationally invariant.

Consider the singular value decomposition
$$W = U\Sigma V^\top.$$

The left singular vectors of $W$ are the eigenvectors of $WW^\top$, and the right singular vectors are the eigenvectors of $W^\top W$. Since $W$ is rotationally invariant, so is the Gram matrix $WW^\top$, which determines the left singular vectors. Specifically, for any fixed orthogonal matrix $Q$,
$$QWW^\top Q^\top \overset{d}{=} WW^\top.$$

This implies that the eigenvectors of $WW^\top$, which form the columns of $U$, must be uniformly distributed on the unit sphere $S^{m-1}$, since no particular direction is preferred. Thus, $\mathbf{u}_1 \sim \mathrm{Unif}(S^{m-1})$.

Similarly, considering $W^\top W$, the right singular vectors (columns of $V$) are eigenvectors of $W^\top W$, and by the same rotational invariance argument,

$$PW^\top W P^\top \stackrel{d}{=} W^\top W$$

for any orthogonal matrix $P \in O(n)$. This implies that $\mathbf{v}_1 \sim \mathrm{Unif}(S^{n-1})$.

The singular values $\sigma_1, \ldots, \sigma_{\min(m,n)}$ of $W$ are independent of the singular vectors. This follows from standard results in random matrix theory, where the eigenvectors of a Wishart matrix (which are the singular vectors of $W$) are independent of its eigenvalues (which correspond to the squared singular values of $W$). Thus, $\mathbf{u}_1$ and $\mathbf{v}_1$ are independent from the singular values and remain uniformly distributed on their respective spheres. $\qquad\square$

**Definition 10.** The *Gamma function*, denoted by $\Gamma(z)$ for $z > 0$, is defined as:

$$\Gamma(z) = \int_0^\infty t^{z-1} e^{-t} dt \,.$$

**Lemma 28.** *Let $A \in \mathbb{R}^{m,n}$ be a centered Gaussian matrix (Definition 8) with variance one. Then there exists a constant $c_{11} \in \mathbb{R}_{>0}$ dependent on $m, n$ such that for any $x \in (0, 1)$ it holds that*

$$\mathcal{P}\left(\sigma_1(A) \le x\right) \le c_{11} x^{mn}$$

*where $\sigma_1(A)$ is the largest singular value of $A$.*

*Proof.* Note that $\|A\|_F^2$ is a Chi-squared random variable with $mn$ degrees of freedom, *i.e.*, it holds that $\|A\|_F^2 \sim \chi_{mn}^2$. The density of for this distribution is given by

$$f(x; mn) = \begin{cases} \dfrac{x^{\frac{mn}{2}-1} e^{-\frac{x}{2}}}{2^{\frac{mn}{2}} \Gamma(\frac{mn}{2})}, & x > 0 \\ 0, & x = 0 \end{cases},$$

where $\Gamma(\cdot)$ is the Gamma function (Definition 10). Hence, we obtain that

$$\mathcal{P}\left(\|A\|_F^2 \le x\right) = \int_0^x f(s; mn) ds = O(x^{\frac{mn}{2}}) \,.$$

Now note that for any matrix $M \in \mathbb{R}^{m,n}$ it holds that

$$\sigma_1(M)^2 \ge \frac{\|M\|_F^2}{\min\{m, n\}} \,,$$

thus

$$\mathcal{P}\left(\sigma_1(A) \le x\right) = \mathcal{P}\left(\sigma_1(A)^2 \le x^2\right) \le \mathcal{P}\left(\|A\|_F^2 \le \min\{m, n\} x^2\right) = O(x^{mn})$$

as required. $\qquad\square$

**Lemma 29.** *Let $\mathbf{v} \in \mathbb{R}^n$ be some fixed unit vector, and let $\mathbf{u}$ be a random vector uniformly distributed on the unit sphere $S^{n-1}$ in $\mathbb{R}^n$. Define $Z = \langle \mathbf{u}, \mathbf{v} \rangle$ as their inner product. Then there exists a constant $c_{12} \in \mathbb{R}_{>0}$ dependent on $n$ such that for any $x \in [-1, 1]$*

$$\mathcal{P}(|Z| \le x) \le c_{12} |x| \,.$$

*Proof.* First note that by symmetry, we can assume WLOG that $\mathbf{v}$ is also uniformly distributed on the unit sphere. By Theorem 1 in Cho [24], the probability density function of $Z$ for any $z \in [-1, 1]$ is given by:

$$f_Z(z) = \frac{\Gamma\left(\frac{n}{2}\right)}{\sqrt{\pi}\, \Gamma\left(\frac{n-1}{2}\right)} (1 - z^2)^{\frac{n-3}{2}} \,,$$

where $\Gamma(\cdot)$ is the Gamma function (Definition 10). In particular, $Z$ has a bounded density supported on $[-1, 1]$. It follows that for any $x \in [-1, 1]$

$$\mathcal{P}\left(|Z| \le x\right) = \int_{-x}^x f_Z(z) ds = O(|x|)$$

as required. $\qquad\square$

**Lemma 30.** *For any $m, n \in \mathbb{N}$ and $\epsilon \in \mathbb{R}_{>0}$, there exists a collection of rank 1 matrices $\{E_i \in \mathbb{R}^{m,n}\}_{i \in [M]}$ where $M$ is depdendent on $m, n$ and $\epsilon$, such that for any rank 1 matrix $E \in \mathbb{R}^{m,n}$ with $\|E\|_F = 1$ there exists some index $i \in [M]$ for which*

$$\|E - E_i\|_F < \epsilon.$$

*Proof.* Standard, see Vershynin [119]. $\qquad\square$

**Definition 11.** Let $X$ be a random vector taking values in $\mathbb{R}^m$. The *characteristic function* of $X$ is the function $\varphi_X : \mathbb{R}^m \to \mathbb{C}$ defined for any $\mathbf{t} \in \mathbb{R}^m$ by:

$$\varphi_X(\mathbf{t}) = \mathbb{E}\left[e^{i\langle \mathbf{t}, X\rangle}\right],$$

where $\langle \mathbf{t}, X\rangle$ denotes the standard inner product in $\mathbb{R}^m$.

**Lemma 31.** *Let $X$ be a random vector taking values in $\mathbb{R}^m$ and let $\phi_X(\mathbf{t})$ be its characteristic function (Definition 11). If $X$ has a probability density function $f_X(\mathbf{x})$, then it can be recovered using the following inversion formula*

$$f_X(\mathbf{x}) = \frac{1}{(2\pi)^m} \int_{\mathbb{R}^m} e^{-i\langle \mathbf{t}, \mathbf{x}\rangle} \phi_X(\mathbf{t}) d\mathbf{t}.$$

*Proof.* Standard, see Zitkovic [140]. $\qquad\square$

**Lemma 32.** *Let $f : \mathbb{R}^m \to [0, \infty)$ be a probability density function with characteristic function $\varphi(\mathbf{t})$. The supremum of $f$, denoted by $\|f\|_\infty := \sup_{\mathbf{x} \in \mathbb{R}^m} |f(\mathbf{x})|$, is bounded by the $L^1$-norm of its characteristic function. Specifically,*

$$\|f\|_\infty \leq \frac{1}{(2\pi)^m} \int_{\mathbb{R}^m} |\varphi(\mathbf{t})| d\mathbf{t}.$$

*Proof.* By the Fourier inversion formula for a probability density function $f$ on $\mathbb{R}^m$ (Lemma 31):

$$f(\mathbf{x}) = \frac{1}{(2\pi)^m} \int_{\mathbb{R}^m} e^{-i\langle \mathbf{t}, \mathbf{x}\rangle} \varphi(\mathbf{t}) d\mathbf{t}.$$

Taking the absolute value, we get:

$$|f(\mathbf{x})| \leq \frac{1}{(2\pi)^m} \int_{\mathbb{R}^m} \left|e^{-i\langle \mathbf{t}, \mathbf{x}\rangle}\right| |\varphi(\mathbf{t})| d\mathbf{t}.$$

Since $\left|e^{-i\langle \mathbf{t}, \mathbf{x}\rangle}\right| = 1$ for all $\mathbf{t} \in \mathbb{R}^m$, the latter simplifies to:

$$|f(\mathbf{x})| \leq \frac{1}{(2\pi)^m} \int_{\mathbb{R}^m} |\varphi(\mathbf{t})| d\mathbf{t}.$$

Taking the supremum over all $\mathbf{x} \in \mathbb{R}^m$, we obtain:

$$\|f\|_\infty \leq \frac{1}{(2\pi)^m} \int_{\mathbb{R}^m} |\varphi(\mathbf{t})| d\mathbf{t}$$

as required. $\qquad\square$

**Lemma 33.** *Let $X$ be a random vector in $\mathbb{R}^m$ with density function $f_X(\mathbf{x})$ and characteristic function $\phi_X(\mathbf{t})$. For any two points $\mathbf{x}, \mathbf{y} \in \mathbb{R}^m$, the difference between their densities is bounded by:*

$$|f_X(\mathbf{x}) - f_X(\mathbf{y})| \leq \frac{\|\mathbf{x} - \mathbf{y}\|_2}{(2\pi)^m} \int_{\mathbb{R}^m} \|\mathbf{t}\|_2 |\phi_X(\mathbf{t})| d\mathbf{t}.$$

*Proof.* From the inversion formula for the density function (Lemma 31), we write:

$$f_X(\mathbf{x}) - f_X(\mathbf{y}) = \frac{1}{(2\pi)^m} \int_{\mathbb{R}^m} \left(e^{-i\langle \mathbf{t}, \mathbf{x}\rangle} - e^{-i\langle \mathbf{t}, \mathbf{y}\rangle}\right) \phi_X(\mathbf{t}) d\mathbf{t}.$$

By factoring out the exponentials:

$$e^{-i\langle \mathbf{t}, \mathbf{x} \rangle} - e^{-i\langle \mathbf{t}, \mathbf{y} \rangle} = e^{-i\langle \mathbf{t}, \mathbf{y} \rangle} \left( 1 - e^{i\langle \mathbf{t}, \mathbf{y} - \mathbf{x} \rangle} \right).$$

Taking absolute values and using the bound:

$$\left| 1 - e^{i\langle \mathbf{t}, \mathbf{y} - \mathbf{x} \rangle} \right| \leq |\langle \mathbf{t}, \mathbf{y} - \mathbf{x} \rangle|,$$

resulting in

$$|f_X(\mathbf{x}) - f_X(\mathbf{y})| \leq \frac{1}{(2\pi)^m} \int_{\mathbb{R}^m} |\langle \mathbf{t}, \mathbf{y} - \mathbf{x} \rangle| |\phi_X(\mathbf{t})| d\mathbf{t}.$$

Finally, applying the Cauchy-Schwarz inequality we obtain that

$$|\langle \mathbf{t}, \mathbf{y} - \mathbf{x} \rangle| \leq \|\mathbf{t}\|_2 \|\mathbf{x} - \mathbf{y}\|_2,$$

which implies that

$$|f_X(\mathbf{x}) - f_X(\mathbf{y})| \leq \frac{\|\mathbf{x} - \mathbf{y}\|_2}{(2\pi)^m} \int_{\mathbb{R}^m} \|\mathbf{t}\|_2 |\phi_X(\mathbf{t})| d\mathbf{t}$$

as required. $\square$

**Lemma 34.** *Let $X \sim \mathcal{N}(0, \Sigma)$ be a zero-centered Gaussian random vector in $\mathbb{R}^m$ with covariance matrix $\Sigma \in \mathbb{R}^{m,m}$. The characteristic function of $X$ is given for any $\mathbf{t} \in \mathbb{R}^m$ by:*

$$\varphi_X(\mathbf{t}) = \exp\left( -\frac{1}{2} \langle \mathbf{t}, \Sigma \mathbf{t} \rangle \right).$$

*Proof.* Standard, see Vershynin [120]. $\square$

**Lemma 35.** *Let $X \sim \mathcal{N}(0, I_m)$ be a standard Gaussian random vector in $\mathbb{R}^m$, and let $A \in \mathbb{R}^{m,m}$ be a positive semi-definite (PSD) matrix. It holds that*

$$\mathbb{E}\left[ e^{-X^\top A X} \right] = \frac{1}{\sqrt{\det(I_m + 2A)}}.$$

*Proof.* The expectation is given by:

$$\mathbb{E}\left[ e^{-X^\top A X} \right] = \int_{\mathbb{R}^m} e^{-\mathbf{x}^\top A \mathbf{x}} \frac{1}{(2\pi)^{m/2}} e^{-\frac{1}{2}\mathbf{x}^\top \mathbf{x}} d\mathbf{x}.$$

Combine the exponential terms:

$$e^{-\mathbf{x}^\top A \mathbf{x}} e^{-\frac{1}{2}\mathbf{x}^\top \mathbf{x}} = e^{-\frac{1}{2}\mathbf{x}^\top (2A + I_m)\mathbf{x}}.$$

Thus,

$$\mathbb{E}\left[ e^{-X^\top A X} \right] = \frac{1}{(2\pi)^{m/2}} \int_{\mathbb{R}^m} e^{-\frac{1}{2}\mathbf{x}^\top (2A + I_m)\mathbf{x}} d\mathbf{x}.$$

This is the Gaussian integral over $\mathbb{R}^m$ for a quadratic form. Using the standard result for multivariate Gaussian integrals

$$\int_{\mathbb{R}^m} e^{-\frac{1}{2}\mathbf{x}^\top B \mathbf{x}} d\mathbf{x} = \frac{(2\pi)^{m/2}}{\sqrt{\det(B)}}$$

where $B$ is a PSD matrix. Observing that the matrix $B = I_m + 2A$ is PSD, we conclude that

$$\mathbb{E}\left[ e^{-X^\top A X} \right] = \frac{1}{\sqrt{\det(I_m + 2A)}}$$

as required. $\square$

**Lemma 36.** *Let $f : \mathbb{R}^{m,m} \to \mathbb{R}$ be a function that depends only on the singular values of a matrix $X \in \mathbb{R}^{m,m}$. Then, the integral of $f$ over the space $\mathbb{R}^{m,m}$ matrices can be expressed as an integral over the singular values as follows:*

$$\int_{\mathbb{R}^{m,m}} f(X)dX = C_m \int_{\sigma_1 \geq \sigma_2 \geq \cdots \geq \sigma_m \geq 0} f(\boldsymbol{\sigma}) \cdot \Delta(\boldsymbol{\sigma})^2 d\boldsymbol{\sigma}$$

*where:*

- $C_m$ *is a constant depending on the dimension $m$.*

- $\Delta(\boldsymbol{\sigma}) = \prod_{1 \leq i < j \leq m}(\sigma_i^2 - \sigma_j^2)$ *is the Vandermonde determinant of the squared singular values.*

- $d\boldsymbol{\sigma}$ *represents the differential volume element over the singular values.*

*Proof.* Consider the singular value decomposition (SVD) of $X$:

$$X = U\Sigma V^\top,$$

where $U, V \in O(m)$ are orthogonal matrices, and $\Sigma = \mathrm{diag}(\sigma_1, \sigma_2, \ldots, \sigma_m)$ is a diagonal matrix containing the singular values $\sigma_i$ of $X$.

The differential volume element $dX$ in the space of $m, m$ matrices can be decomposed into the product of volume elements corresponding to $U$, $\Sigma$, and $V$, along with the Jacobian determinant of the transformation:

$$dX = J(\Sigma)dU\,d\Sigma\,dV$$

where $J(\Sigma)$ is the Jacobian determinant associated with the change of variables from $X$ to $(U, \Sigma, V)$.

For a function $f$ that depends only on the singular values, the integral over the orthogonal matrices $U$ and $V$ contribute to the constant $C_m$, allowing us to focus on the integral over the singular values. The Jacobian determinant $J(\Sigma)$ for the transformation involving singular values in the space of $m, m$ matrices is given by:

$$J(\Sigma) = \Delta(\boldsymbol{\sigma})^2$$

where $\Delta(\boldsymbol{\sigma}) = \prod_{1 \leq i < j \leq m}(\sigma_i^2 - \sigma_j^2)$ (see Rennie [99]).

Therefore, the integral over the space of $m, m$ matrices can be rewritten as:

$$\int_{\mathbb{R}^{m,m}} f(X)dX = C_m \int_{\sigma_1 \geq \sigma_2 \geq \cdots \geq \sigma_m \geq 0} f(\boldsymbol{\sigma}) \cdot \Delta(\boldsymbol{\sigma})^2 d\boldsymbol{\sigma}$$

where $d\boldsymbol{\sigma}$ is the measure on the singular values. $\qquad\square$

**Corollary 3.** *Consider the function $f(X) = \|X\|_F \left( \dfrac{1}{\sqrt{\det\left(I + \frac{X^\top X}{km}\right)}} \right)^k$, where $X \in \mathbb{R}^{m,m}$. Using the change of variables to singular values (Lemma 36):*

$$\int_{\mathbb{R}^{m,m}} \|X\|_F \left( \frac{1}{\sqrt{\det\left(I + \frac{X^\top X}{km}\right)}} \right)^k dX$$

$$= C_m \int_{\sigma_1 \geq \sigma_2 \geq \cdots \geq \sigma_m \geq 0} \left( \sum_{i=1}^m \sigma_i^2 \right)^{1/2} \cdot \left( \frac{1}{\sqrt{\prod_{i=1}^m 1 + \frac{\sigma_i^2}{km}}} \right)^k \Delta(\boldsymbol{\sigma})^2 d\boldsymbol{\sigma},$$

*where:*

- $\|X\|_F = \left( \sum_{i=1}^m \sigma_i^2 \right)^{1/2}$ *is the Frobenius norm of $X$.*

- $\sqrt{\det\left(I + \frac{X^\top X}{km}\right)} = \sqrt{\prod_{i=1}^{m}\left(1 + \frac{\sigma_i^2}{km}\right)}.$

- $\Delta(\sigma) = \prod_{1 \leq i < j \leq m}\left(\sigma_i^2 - \sigma_j^2\right)$ *is the Vandermonde determinant of the squared singular values.*

*This reduces the integral to one over the singular values $\sigma_1, \sigma_2, \ldots, \sigma_m$.*

**Definition 12.** The *Beta function*, denoted by $B(x, y)$ for $x, y > 0$, is defined as:

$$B(x, y) = \int_0^\infty \frac{t^{x-1}}{(1+t)^{x+y}}\,dt\,.$$

**Lemma 37.** *For any $x, y > 0$ it holds that*

$$B(x, y) = \frac{\Gamma(x)\Gamma(y)}{\Gamma(x+y)}\,.$$

*Proof.* Standard, see Davis [31]. □

**Lemma 38.** *For large $z$, the ratio of the Gamma function evaluated at $z$ and $z + c$ for any constant $c$ satisfies:*

$$\frac{\Gamma(z+c)}{\Gamma(z)} \sim z^c \quad \text{as } z \to \infty\,.$$

*Proof.* Using Stirling's approximation for the Gamma function:

$$\Gamma(z) \sim \sqrt{2\pi}z^{z-1/2}e^{-z}, \quad \text{as } z \to \infty\,,$$

we compute the ratio:

$$\frac{\Gamma(z+c)}{\Gamma(z)} \sim \frac{\sqrt{2\pi}(z+c)^{z+c-1/2}e^{-(z+c)}}{\sqrt{2\pi}z^{z-1/2}e^{-z}}\,.$$

Simplify the terms:

$$\frac{\Gamma(z+c)}{\Gamma(z)} \sim \frac{(z+c)^{z+c-1/2}e^{-c}}{z^{z-1/2}}\,.$$

Taking the dominant term for large $z$, we approximate $z + c \sim z$, so:

$$\frac{\Gamma(z+c)}{\Gamma(z)} \sim z^c$$

as required. □

**Lemma 39.** *Let $\Phi$ denote a Gaussian measure on $\mathbb{R}^n$ with mean zero and covariance matrix $\Sigma$. For any two closed, symmetric, convex subsets $A, B \subset \mathbb{R}^n$, the Gaussian Correlation Inequality (GCI) states:*

$$\Phi(A \cap B) \geq \Phi(A)\Phi(B)\,.$$

*Proof.* See Latała and Matlak [67]. □

**Lemma 40.** *For all $\delta > 0$, there exists a constant $c(\delta)$ such that with probability at least $1 - \delta$, the norm of an $L$-dimensional standard Gaussian vector $X \sim \mathcal{N}(0, I_L)$ satisfies:*

$$\|X\| \leq c(\delta)\sqrt{L}\,.$$

*Proof.* See Vershynin [118]. □

## F   Theorem 3.3 From Soltanolkotabi et al. [106]

Proposition 1 restates Theorem 3.3 from Soltanolkotabi et al. [106] using $O$- and $\tilde{O}$-notations. For completeness, Proposition 4 below restates the theorem without these notations.

**Proposition 4** (restatement of Theorem 3.3 from [106], without $O$- and $\tilde{O}$-notations). *There exist universal constants $c_1, \ldots, c_{10} \in \mathbb{R}_{>0}$ with which the following holds. Suppose the activation $\sigma(\cdot)$ is linear (i.e., $\sigma(\alpha) = \alpha$ for all $\alpha = \mathbb{R}$), and the depth $d$ equals two. Let $\kappa \in \mathbb{R}_{>0}$ be the condition number of $W^*$. Let $\mathcal{Q}(\cdot)$ be a zero-centered Gaussian probability distribution, i.e., $\mathcal{Q}(\cdot) = \mathcal{N}(\cdot\,; 0, \nu)$, with variance*

$$0 < \nu \leq \frac{c_1 \|W^*\|_F \sqrt{m'}}{k^{9.5} \left(\max\{m+m', k\}\right)^4} \left( \frac{\sqrt{k} - \sqrt{r} - 1}{c_2 \kappa^2 \sqrt{\max\{m+m', k\}}} \right)^{c_3 \kappa}$$

*(recall that $r$ is the rank of the ground truth matrix $W^*$, whose dimensions are $m$ and $m'$). Let $\mathcal{P}(\cdot)$ be the probability distribution over weight settings that is generated by $\mathcal{Q}(\cdot)$ (Definition 3). Assume the measurement matrices $(A_i)_{i=1}^n$ satisfy an RIP (Definition 1) of order $2r + 1$ with a constant $\delta \in (0, \min\{1, c_4/(\kappa^3 \sqrt{r})\})$. Consider minimization of the training loss $\mathcal{L}_{\text{train}}(\cdot)$ via gradient descent (Equation (7)) with initialization drawn from $\mathcal{P}(\cdot)$ and step size*

$$0 < \eta \leq \frac{c_5}{\kappa^5 \|W^*\|_F} \cdot \frac{1}{\ln\left( \frac{4\sqrt{2}(km')^{1/4}\|W^*\|_F}{\sqrt{\nu}(\sqrt{k} - \sqrt{r} - 1)} \right)}.$$

*Then, there exists some $\tau \in \mathbb{N}$ which satisfies*

$$\tau \leq \frac{c_6 \ln\left( \frac{4\sqrt{2}(km')^{1/4}\|W^*\|_F}{\sqrt{\nu}(\sqrt{k} - \sqrt{r} - 1)} \right)}{\eta \sigma_{\min}(W^*)},$$

*such that for any width $k$ of the matrix factorization, after $\tau$ iterations of gradient descent, with probability at least $1 - c_7 \exp(-c_8 k) + c_9^{k-r+1}$ over its initialization, the generalization loss $\mathcal{L}_{\text{gen}}(\cdot)$ is no more than $c_{10} \|W^*\|_F^{7/10} \nu^{3/10}/(km')^{3/20}$.*

## G   Further Experiments

Section 5 corroborates our theory by empirically demonstrating that in matrix factorization (Section 3.3), the generalization attained by G&C (Section 3.5) deteriorates as width increases and improves as depth increases, whereas gradient descent (Section 3.4) attains good generalization throughout. This appendix reports further experiments.

Figures 3 and 4 respectively extend Figures 1 and 2 to account for gradient descent with momentum. Figures 5 and 6 respectively extend Figures 1 and 2 to account for a ground truth matrix of rank two.[8] Figures 7 and 8 respectively extend Figures 1 and 2 to account for G&C with a Kaiming Uniform prior distribution. Figures 9 and 10 respectively extend Figures 1 and 2 to a special case where the measurement matrices are indicator matrices (meaning each holds one in a single entry and zeros elsewhere), leading to what is known as *low rank matrix completion*—a problem that has been studied extensively [144, 142, 146, 141, 148, 143]. Figure 11 extends Figure 2 to account for G&C with a prior distribution that does not include normalization (Definition 3). Figure 12 establishes that our large width results are robust to the exact choice of $\epsilon_{\text{train}}$. Finally, Figure 13 shows that the trends displayed in Figure 2—namely improved performance of G&C—continue for larger values of depth.

## H   Experimental Details

In this appendix, we provide experimental details omitted from Section 5 and Appendix G. Code for reproducing all demonstrations can be found in `https://github.com/YoniSlutzky98/nn-gd-gen-mf`. All experiments were implemented using PyTorch [145] and carried out on a single Nvidia RTX A6000 GPU.

**Ground truth matrix.** In all experiments the ground truth matrices were generated via the following procedure. First, two matrices $U \in \mathbb{R}^{m,r}$ and $V \in \mathbb{R}^{r,m'}$ were generated by independently drawing

---

[8]We note that in Figures 1, 3, and 5, a double descent like phenomenon , namely a (small) spike in generalization loss which is then reversed as model size is further increased, is observed for gradient descent (but not for G&C) with the Leaky ReLU activation. We leave investigating this phenomenon for future work.

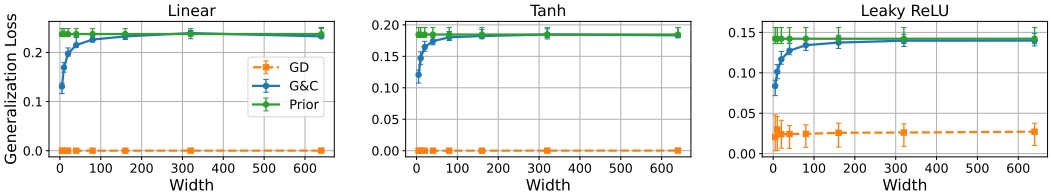

Figure 3: In line with our theory (Section 4.2), as the width of a matrix factorization increases, the generalization attained by G&C deteriorates, to the point of being no better than chance, *i.e.*, no better than the generalization attained by randomly drawing a single weight setting from the prior distribution while disregarding the training data. This figure adheres to the caption of Figure 1, except that we employ gradient descent with a momentum coefficient of 0.9 [91]. For further details see Appendix H.

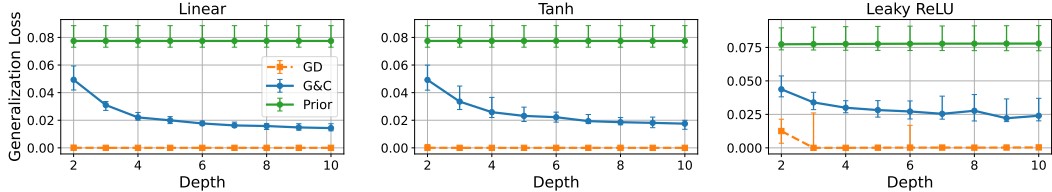

Figure 4: In line with our theory (Section 4.3), as the depth of a matrix factorization increases, the generalization attained by G&C improves, drawing closer to that of gradient descent. This figure adheres to the caption of Figure 2, except that we employ gradient descent with a momentum coefficient of 0.9 [91]. For further details see Appendix H.

each of their entries from the standard Gaussian distribution $\mathcal{N}(\cdot; 0, 1)$. Here $r$ stands for the desired ground truth matrix rank. Then, the ground truth matrix was set as

$$W^* = \frac{b}{\|UV\|_F} \cdot UV \,,$$

where $b$ stands for the desired ground truth matrix norm. This procedure ensured that $W^*$ had rank $r$ and norm $b$. In the experiments reported in Figures 5 and 6, the ground truth matrices were of rank two and norm one. In the rest of the experiments, the ground truth matrices were of rank one and norm one.

**Measurement matrices.** In all experiments, the training measurement matrices were generated by independently drawing each of their entries from the standard Gaussian distribution $\mathcal{N}(\cdot; 0, 1)$, and then normalizing each matrix to have norm one. For each set of training measurements, the corresponding orthonormal basis $\mathcal{B}$ was generated by performing the Gram-Schmidt process and taking the components which were not spanned by the original set of measurements. In the experiments reported in Figures 5 and 6 the amount of training measurements was 22. In the rest of the experiments the amount of training measurements was 15.

**G&C optimization.** A G&C sample consisted of a drawing of the weight matrices $W_1, \ldots, W_d$ and computation of the factorization $W$ (Equation (4)). If the training loss (Equation (5)) of the given factorization is lower than $\epsilon_{\text{train}}$ then the sample is considered successful. Table 1 reports the value of $\epsilon_{\text{train}}$ used in each experiment.

For each trial—that is, for each random draw of the ground truth and measurement matrices—the G&C algorithm was executed by drawing `num_samples` samples and averaging the generalization losses of all successful samples. Table 2 reports the value of `num_samples` used in each experiment.

To efficiently execute the G&C algorithm, the following batched implementation was used. Given a sample batch size `bs`, for each layer $j \in [d]$, the layer's weight matrices $W_j \in \mathbb{R}^{m_{j+1}, m_j}$ were: *(i)* drawn in parallel as a tensor of dimensions $(\text{bs}, m_{j+1}, m_j)$; *(ii)* multiplied in parallel with the factorizations produced in the previous layer via the `bmm(·)` function of PyTorch; and; *(iii)* applied the activation function elementwise. The weights of the $j$th layer were drawn with independent entries from either $\mathcal{N}(\cdot; 0, 1)$ or $\mathcal{U}(\cdot; -1, 1)$ and then scaled by $\sqrt{m_j}$. This procedure was performed until a

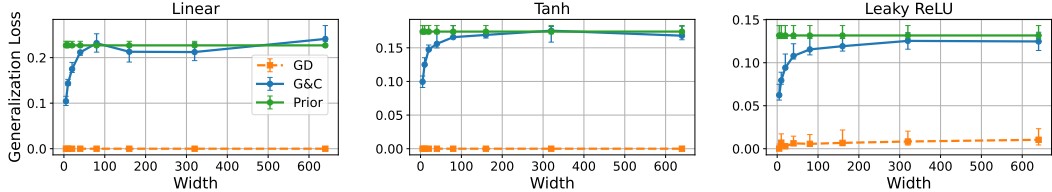

Figure 5: In line with our theory (Section 4.2), as the width of a matrix factorization increases, the generalization attained by G&C deteriorates, to the point of being no better than chance, *i.e.*, no better than the generalization attained by randomly drawing a single weight setting from the prior distribution while disregarding the training data. In contrast, gradient descent attains good generalization across all widths. This figure adheres to the caption of Figure 1, except that the ground truth matrix had rank two and the training data size was $n = 22$. For further details see Appendix H.

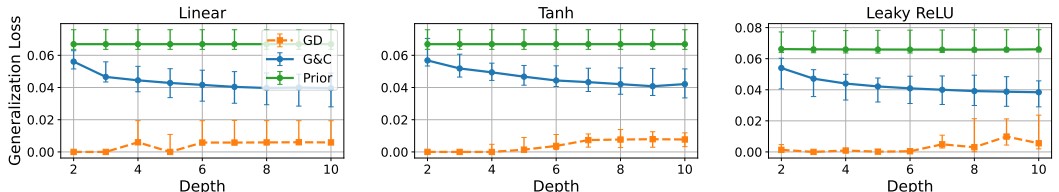

Figure 6: In line with our theory (Section 4.3), as the depth of a matrix factorization increases, the generalization attained by G&C improves, drawing closer to that of gradient descent. This figure adheres to the caption of Figure 2, except that the ground truth matrix had rank two and the training data size was $n = 22$. We note that with larger depths, the generalization attained by gradient descent is not as good as it is with smaller depths.[9] For further details see Appendix H.

total of `num_samples` samples are accumulated. In experiments where the final factorizations were normalized, a softening constant of size $10^{-6}$ was added to the denominator.

**GD optimization.** In all of the experiments we trained gradient descent using the empirical sum of squared errors as a loss function and optimized over full batches.

All weights matrices were initialized as follows. First, for each layer $j \in [d]$, the layer's weight matrix $W_j \in \mathbb{R}^{m_{j+1}, m_j}$ was drawn with independent entries from $\mathcal{U}\left(\cdot; -1/\sqrt{m_j}, 1/\sqrt{m_j}\right)$ (this is the default PyTorch initialization). Next, in order to facilitate a near-zero initialization, all weight matrices were further scaled by a scalar `init_scale`. `init_scale` was set to $10^{-3}$ in all the experiments of Figures 1, 3, 5, 7, 9, and 12. Table 3 reports the values of `init_scale` used in the experiments of Figures 2, 4, 6, 8, 10, 11, and 13.

In order to facilitate more efficient experimentation, we optimized using gradient descent with an adaptive learning rate scheme, where at each iteration a base learning rate is divided by the square root of an exponential moving average of squared gradient norms (see appendix D.2 in Razin et al. [97] for more details). We used a weighted average coefficient of $\alpha = 0.99$ and a softening constant of $10^{-6}$. Note that only the learning rate (step size) is affected by this scheme, not the direction of movement. Comparisons between the adaptive scheme and optimization with a fixed learning rate showed no significant difference in terms of the dynamics, while run times of the former were considerably shorter. The base learning rate $\eta$ was set to $10^{-4}$ in all the experiments of Figures 1, 3, 5, 7, 9, and 12. Table 4 specifies the base learning rates used in the experiments of Figures 2, 4, 6, 8, 10, 11, and 13. Table 5 specifies the number of epochs used in each experiment.

---

[9] We examined weight settings found by gradient descent, and observed that with larger depths, the factorized matrix $W$ (Equation (4)) had an effective rank [147] lower than that of the ground truth matrix $W^*$. This aligns with the conventional wisdom by which adding layers to a matrix factorization leads gradient descent to have stronger implicit bias towards low rank [6, 25]. The fact that it was possible to fit the training data with an effective rank lower than that of the ground truth matrix, is an artifact of the training data size being limited in order to ensure reasonable runtime by G&C.

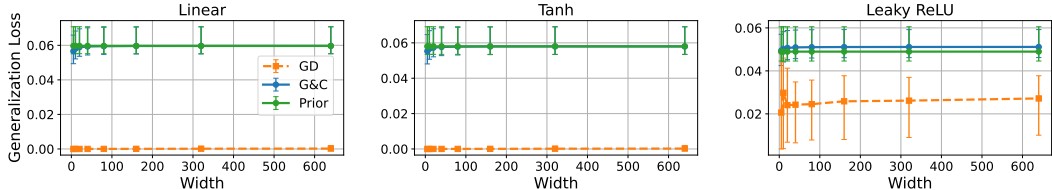

Figure 7: In line with our theory (Section 4.2), as the width of a matrix factorization increases, the generalization attained by G&C deteriorates, to the point of being no better than chance, *i.e.*, no better than the generalization attained by randomly drawing a single weight setting from the prior distribution while disregarding the training data. This figure adheres to the caption of Figure 1, except that the prior distribution of G&C was Kaiming Uniform. For further details see Appendix H.

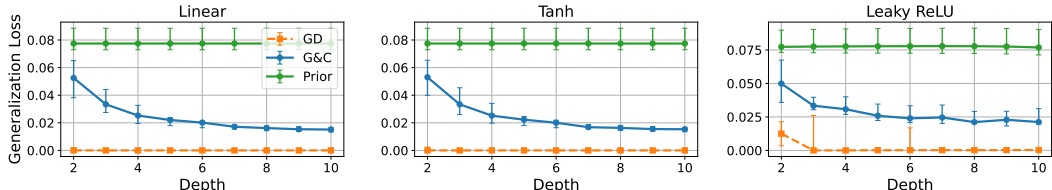

Figure 8: In line with our theory (Section 4.3), as the depth of a matrix factorization increases, the generalization attained by G&C improves, drawing closer to that of gradient descent. This figure adheres to the caption of Figure 2, except that the prior distribution of G&C was Kaiming Uniform. For further details see Appendix H.

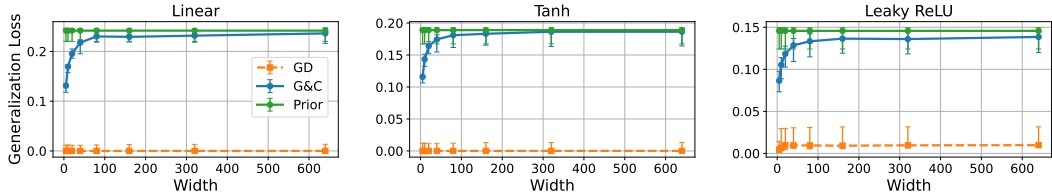

Figure 9: In line with our theory (Section 4.2), as the width of a matrix factorization increases, the generalization attained by G&C deteriorates, to the point of being no better than chance, *i.e.*, no better than the generalization attained by randomly drawing a single weight setting from the prior distribution while disregarding the training data. This figure adheres to the caption of Figure 1, except that measurement matrices were indicator matrices (meaning each held one in a single entry and zeros elsewhere), leading to a low rank matrix completion problem. For further details see Appendix H.

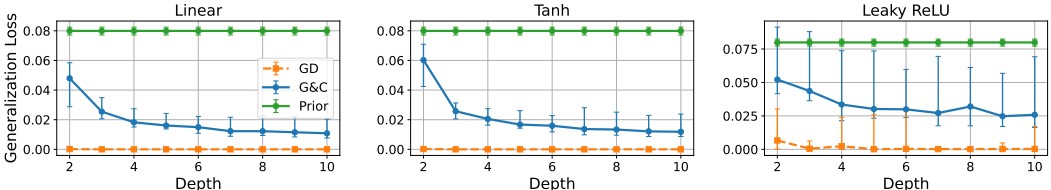

Figure 10: In line with our theory (Section 4.3), as the depth of a matrix factorization increases, the generalization attained by G&C improves, drawing closer to that of gradient descent. This figure adheres to the caption of Figure 2, except that measurement matrices were indicator matrices (meaning each held one in a single entry and zeros elsewhere), leading to a low rank matrix completion problem. For further details see Appendix H.

Table 1: Training error $\epsilon_{\text{train}}$ used in the experiments of Figures 1 to 13.

| Setting | $\epsilon_{\text{train}}$ |
|---|---|
| Figures 1, 3, 5, 7, and 9 | 0.02 |
| Figures 2, 4, 8, 10, and 13 | 0.0035 |
| Figures 6 and 11 | 0.01 |
| Figure 12 | 0.025, 0.03 |

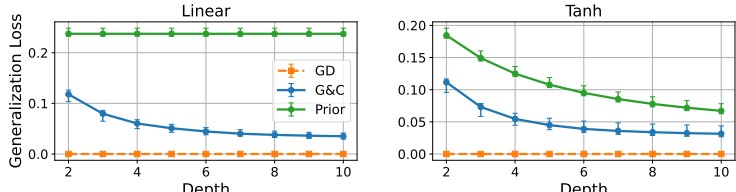

Figure 11: In line with our theory (Section 4.3), as the depth of a matrix factorization increases, the generalization attained by G&C improves, drawing closer to that of gradient descent. This figure adheres to the caption of Figure 2, except for the following differences: *(i)* the prior distribution of G&C did not include normalization; and *(ii)* experiments with Leaky ReLU activation were omitted (because without normalization the number of draws it required from G&C was computationally prohibitive). We note that with tanh activation, the generalization attained by drawing a single weight setting from the prior distribution (while disregarding the training data) improves as depth increases. This results from the factorized matrix W (Equation (4)) approaching the zero matrix, whose generalization loss is greater than that attainable by G&C. For further details see Appendix H.

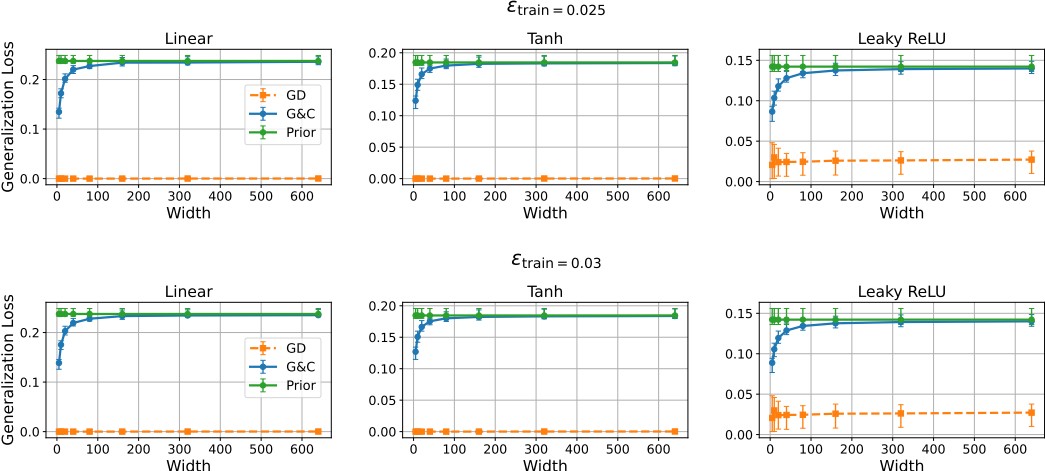

Figure 12: In line with our theory (Section 4.2), as the width of a matrix factorization increases, the generalization attained by G&C deteriorates, to the point of being no better than chance, *i.e.*, no better than the generalization attained by randomly drawing a single weight setting from the prior distribution while disregarding the training data. The above figures demonstrate that the results in Figure 1 are robust to the choice of $\epsilon_{train}$. Here we show $\epsilon_{train} = 0.025$ and $\epsilon_{train} = 0.03$ to complement $\epsilon_{train} = 0.02$ shown in the main text. We see that the qualitative behavior is the same.

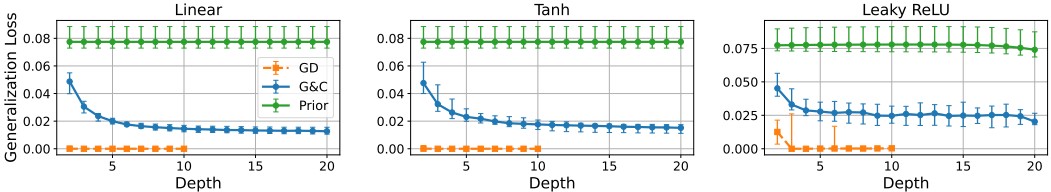

Figure 13: In line with our theory (Section 4.3), as the depth of a matrix factorization increases, the generalization attained by G&C improves, drawing closer to that of gradient descent. This figure adheres to the caption of Figure 2, except that we plot results for G&C up to depth 20 as opposed to 10. Results for gradient descent were omitted for larger depths because running it became computationally prohibitive. Note that our theory does not imply that as the depth tends to infinity the generalization loss of G&C will tend to zero, but rather that the probability that it is smaller than some constant times $\epsilon_{train}$ will tend to 1. For further details see Appendix H

Table 2: Number of G&C samples used in the experiments of Figures 1 to 13.

| Setting | num_samples |
|---|---|
| Figures 1, 3, 5, 7, 9, and 12 | $10^8$ |
| Figures 2, 4, 6, 8, 10, and 11 | $10^9$ |
| Figure 13 | $5 \times 10^9$ |

Table 3: Gradient descent initialization scale used in the experiments of Figures 2, 4, 6, 8, 10, 11, and 13 (increasing depth). The first three columns (left) specify the experiment (setting, activation function and associated depths $d$), and the last column specifies value of init_scale used.

| Setting | Activation | $d$ | init_scale |
|---|---|---|---|
| Figures 2, 4, 6, 8, 10, 11, and 13 | Linear, Tanh, Leaky ReLU | $2, 3, 4$ | 0.001 |
| Figures 2, 4, 6, 8, 10, 11, and 13 | Linear, Tanh | $5, 6, 7, 8$ | 0.1 |
| Figures 2, 4, 6, 8, 10, 11, and 13 | Linear, Tanh | $9, 10$ | 0.2 |
| Figures 2, 4, 6, 8, 10, 11, and 13 | Leaky ReLU | $5$ | 0.03 |
| Figures 2, 4, 6, 8, 10, 11, and 13 | Leaky ReLU | $6, 7$ | 0.1 |
| Figures 2, 4, 6, 8, 10, 11, and 13 | Leaky ReLU | $8, 9$ | 0.2 |
| Figures 2, 4, 6, 8, 10, 11, and 13 | Leaky ReLU | $10$ | 0.8 |

Table 4: Gradient descent base learning rate used in the experiments of Figures 2, 4, 6, 8, 10, 11, and 13 (increasing depth). The first three columns (left) specify the experiment (setting, activation function and associated depths $d$), and the last column specifies the base learning rate $\eta$ used.

| Setting | Activation | $d$ | $\eta$ |
|---|---|---|---|
| Figures 2, 4, 6, 8, 10, 11, and 13 | Linear, Tanh | $2, \dots, 10$ | 0.01 |
| Figures 2, 4, 6, 8, 10, 11, and 13 | Leaky ReLU | $2, 3, 4$ | 0.01 |
| Figures 2, 4, 6, 8, 10, 11, and 13 | Leaky ReLU | $5, \dots, 10$ | 0.1 |

Table 5: Number of gradient descent epochs used in the experiments of Figures 1 to 13. The first two columns (left) specify the experiment (setting and activation functions), and the last column specifies the number of epochs used.

| Setting | Activation | Number of Epochs |
|---|---|---|
| Figures 1, 3, 5, 7, 9, and 12 | Linear, Tanh, Leaky ReLU | 100000 |
| Figures 2, 4, 8, 10, 11, and 13 | Linear, Tanh | 20000 |
| Figure 6 | Linear, Tanh | 50000 |
| Figures 2, 4, 6, 8, 10, 11, and 13 | Leaky ReLU | 50000 |

