# OpenReview forum: "Do Neural Networks Need Gradient Descent to Generalize? A Theoretical Study"
_NeurIPS.cc/2025/Conference — NeurIPS 2025 poster_

### Official Review · Reviewer_2Hfc · 2025-06-21

**Clarity:** 3
**Significance:** 3
**Originality:** 3
**Rating:** 4
**Confidence:** 5

**Summary:**

This paper studies the generalization properties of over-parameterized neural networks by considering scenario of low-rank matrix sensing. The central contribution of this paper derives generalization bounds for a guess-and-check optimization algorithm. In particular, when the weights of a fully connected FFN is drawn from an iid Gaussian distribution:

- In the limit as the width of a fully-connected FFN $\to \infty$, the out-of-sample loss of guess and check is as bad as random guessing.

- Under additional assumptions such as *restricted isometry property*, in the limit as the depth of the network $\to \infty$, the out-of-sample loss approaches 1.

These two results demonstrate that whether guess-and-check can match gradient descent in generalization performance is instance dependent.

**Questions:**

see above

**Ethical Concerns:**

["NO or VERY MINOR ethics concerns only"]

**Final Justification:**

This paper is technically sound and offers some interesting perspectives. While the authors' rebuttal is quite detailed, I still have some lingering doubts about the deeper implications of this work. Therefore, I will maintain my original score, but will raise my confidence to show my support for this paper.

**Limitations:**

see above

**Paper Formatting Concerns:**

No major formatting issues.

**Quality:**

3

**Strengths And Weaknesses:**

I do not have too many comments on this work because the main takeaway of this paper is clear and the overall technical setup is not particularly complicated. It is actually refreshing to read this paper given the recent trend of content creeps in ML theory papers.

In terms of the main message of the paper, I think the testbed should be called low-rank matrix sensing rather than matrix factorization, since the later does not make much sense when the activation in eq (4) is non-linear.

The paper as whole is not difficult to understand and the author did a good job at identifying a concise theoretical framework that captures the intended messages. Personally, I feel that the first result on infinite width (Theorem 1) is not too surprising, and the second result on infinite depth (Theorem 2) does require some fairly strong assumptions. But both of these results answered very relevant questions and therefore they should be shared with the wider theory community.

For the technical contents, I briefly read over the proofs in Appendix A and B, though I did not check the details line-by-line. The math seems to be sound overall. The theorem statements in the main body are clear and easy to understand, but their counterparts in the Appendix (e.g. Theorem 6) are somewhat bloated.

Things that could be improved:
1. Despite the conciseness of the overall message, I found the paper to be a bit more difficult to read than it should be. In particular, Definition 3 should be split into 2. And in many instances, equations should be on their own lines and equation block alignments are often not good.

2. Definition 1 should be moved to Section 4.3. And the author should add a bit more discussing the RIP property because it may be unfamiliar to a lot of readers.

3. I feel the experiments in Section 5 is unnecessary.

4. The proof sketches for both Theorem 1 and 2 are kind of rushed and do not seem to capture all of the main technical steps of the actual proofs.

5. The assumptions for Theorem 2 is really, really strong. Imposing the ground truth matrix to be rank 1 really simplifies the problem by a lot. I recommend the authors to extend this result to more general settings.

6. The proof of Theorem 1 seems to heavily rely on the well-known equivalence between infinite width neural networks and Gaussian processes. This does somewhat diminish the technical novelties of the paper.

Overall, I think the results of this paper is sufficiently interesting for publication. But it is not particularly sophisticated in the technical contribution and some might argue that the guess-and-check algorithm does not warrant research in the first place. Also, there is very little from this paper that could lead to wider implications for both future theoretical and empirical studies. So I think it is fair to recommend a **borderline accept**.

---

> ### Author Rebuttal · Authors · 2025-07-30
>
> Thank you for highlighting that our results answer very relevant questions and should be shared with the wider theory community. Thank you also for noting the soundness of our theory and the clarity of the text. Below we respond to the points you raised. If you find our responses satisfactory, we would greatly appreciate it if you would consider raising your score.
>
> >I feel the experiments in Section 5 is unnecessary.
>
> While we agree that experiments are not strictly necessary for a theoretical paper such as ours, we chose to include them in order to demonstrate that the conclusions from our theory extend beyond the technical assumptions made.
>
> >The assumptions for Theorem 2 is really, really strong. Imposing the ground truth matrix to be rank 1 really simplifies the problem by a lot. I recommend the authors to extend this result to more general settings.
>
> Theorem 2 indeed assumes that the ground truth matrix has rank one. While the theorem’s proof currently requires this assumption, its intuition is far more general: as the depth of the network increases, the product of weight matrices (drawn from a Gaussian distribution) tends toward low rank, thus fitting training data will be achieved with lowest rank possible.  We corroborate this intuition empirically in the supplementary material (Figures 5 and 6), demonstrating the conclusions of Theorem 2 with ground truth ranks beyond one.  Extending the formal proof of Theorem 2 beyond ground truth rank one is a valuable direction for future research.
>
> >The proof of Theorem 1 seems to heavily rely on the well-known equivalence between infinite width neural networks and Gaussian processes. This does somewhat diminish the technical novelties of the paper
>
> The proof of Theorem 1 indeed builds upon existing machinery, in particular connections between neural networks and Gaussian Processes. We stress however that the theorem is not an immediate outcome of existing machinery (see complete proof in Appendix A).  Moreover, its extension to the case where $\epsilon_{\text{train}}$ is unspecified (Appendix C) requires highly non-trivial arguments, namely, a strong form of random variable convergence, as well as Gaussian Correlation inequalities.
>
> >some might argue that the guess-and-check algorithm does not warrant research in the first place
>
> As an algorithm, the relevance of G&C is indeed limited. However, from a theoretical standpoint, studying G&C is essentially equivalent to studying the volume hypothesis, and thus has profound implications on our understanding of neural networks (see our introduction).
>
> >little from this paper that could lead to wider implications
>
> The validity of our conclusions for real-world non-linear neural networks was empirically demonstrated by Peleg and Hein ([1]), which found that the generalization of G&C is inferior to that of GD with wide networks, but comparable to it with deep networks.  From a theoretical perspective, as we note in the text, we hope that our theory will serve as a stepping stone towards analysis of more realistic settings. One promising avenue in this direction would be the analysis of tensor factorization (tensor sensing with neural network overparameterization), which is a model much closer to real-world neural networks than matrix factorization (see, e.g., [2, 3]).  Tensor factorization has various similarities to matrix factorization, thus we believe elements of our theory may carry over to this richer model. We will mention this in the conclusion section of the camera-ready version. Thank you!
>
> >Despite the conciseness of the overall message, I found the paper to be a bit more difficult to read than it should be. In particular, Definition 3 should be split into 2. And in many instances, equations should be on their own lines and equation block alignments are often not good.
>
> >Definition 1 should be moved to Section 4.3. And the author should add a bit more discussing the RIP property because it may be unfamiliar to a lot of readers.
>
> >The proof sketches for both Theorem 1 and 2 are kind of rushed and do not seem to capture all of the main technical steps of the actual proofs.
>
> Thank you for these suggestions; they will certainly help us improve the readability and clarity of the paper toward the camera-ready version. In particular, we will carefully revisit the presentation of definitions and equations, consider additional clarifications on the RIP, and enhance the exposition of the proof sketches.
>
> ---
> **References**
>
> [1] Peleg and Hein. “Bias of Stochastic Gradient Descent or the Architecture: Disentangling the Effects of Overparameterization of Neural Networks”
>
> [2] Razin et al. “Implicit Regularization in Tensor Factorization.”
>
> [3] Razin et al. “Implicit Regularization in Hierarchical Tensor Factorization and Deep Convolutional Neural Networks.“

---

> > ### Comment · Reviewer_2Hfc · 2025-08-02
> >
> > I thank the author for their detailed response.
> >
> > > Thank you for these suggestions; they will certainly help us improve the readability and clarity of the paper toward the camera-ready version. In particular, we will carefully revisit the presentation of definitions and equations, consider additional clarifications on the RIP, and enhance the exposition of the proof sketches.
> >
> > Would you mind giving me a draft of the improved proof sketches? While I think this paper is overall solid, I cannot raise my rating unless I see some more concrete changes.

---

> > > ### Author Response · Authors · 2025-08-04
> > > **Expanded Proof Sketches - Part 1**
> > >
> > > Thank you for your engagement, and for your willingness to consider raising your score! Below are drafts of expanded proof sketches. Please let us know if there are any additional details you would like the proof sketches to include.
> > >
> > > -----
> > >
> > > **Theorem 1:**
> > >
> > > The proof begins by establishing an equivalence between a matrix factorization and a feedforward fully connected neural network: each column of a factorized matrix $W:=W_{d}\cdots W_{1}$ can be seen as the output of a feedforward fully connected neural network when its input is a standard basis vector. This equivalence facilitates utilization of the following theoretical results from [1] and [2]:
> > >
> > > (i) when the width $k$ of a feedforward fully connected neural network grows, drawing its weight settings from any $\mathcal{P} ( \cdot )$ generated by an arbitrary regular probability distribution $\mathcal{Q} ( \cdot )$, leads the output of each of the network’s layers to converge in distribution to a Gaussian process, whose covariance structure can be computed recursively using the covariance structures of previous layers; and
> > >
> > > (ii) when $\mathcal{Q} ( \cdot )$ is a zero-centered Gaussian distribution, the convex distance (a common distance metric for multivariate distributions) between the output of a network’s layer and the Gaussian process to which it converges, is $O ( 1 / \sqrt{k} )$.
> > >
> > > For treating the case where $\mathcal{Q} ( \cdot )$ is an arbitrary regular probability distribution and the width $k$ tends to infinity, the proof utilizes result (i) above. Namely, it utilizes the recursive computation of covariance structures to show that the columns of the factorized matrix $W$ (i.e., that the vectors obtained by applying the feedforward fully connected neural network to standard basis vectors) converge to Gaussian vectors with statistically independent entries, and furthermore, since the activation $\sigma ( \cdot )$ is anti-symmetric, these Gaussian vectors are statistically independent of one another.  Overall, it holds that $W$ converges in distribution to a random matrix $W_{\text{iid}}$ whose entries are independently drawn from a zero-centered Gaussian distribution. Since the measurement matrices $( A_{i} )\_{i=1}^{n}$ are orthogonal to the basis $\mathcal{B}$ that defines the generalization loss, the events $\mathcal{L}\_{\text{train}} ( W\_{\text{iid}}) < \epsilon\_{\text{train}} $ and $\mathcal{L}\_{\text{gen}}  ( W\_{\text{iid}} ) < \epsilon\_{\text{gen}} $ are statistically independent. Therefore, conditioning on the event that the training loss is lower than $\epsilon_{\text{train}}$ does not change the probability of the event that the generalization loss is lower than $\epsilon_{\text{gen}}$.
> > >
> > > The proof concludes by treating the case where $\mathcal{Q} ( \cdot )$ is a zero-centered Gaussian distribution and the width $k$ is finite. There, result (ii) above is utilized to show that the probabilities of the events $\mathcal{L}\_{\text{train}} ( W ) < \epsilon\_{\text{train}}$ and $\mathcal{L}\_{\text{gen}} ( W ) < \epsilon\_{\text{gen}}$ converge to those of the events $\mathcal{L}\_{\text{train}} ( W\_{\text{iid}} ) < \epsilon\_{\text{train}}$ and $\mathcal{L}\_{\text{gen}} ( W\_{\text{iid}} ) < \epsilon\_{\text{gen}}$, respectively, at a sufficiently fast rate.

---

> ### Author Response · Authors · 2025-08-04
> **Expanded Proof Sketches - Part 2**
>
> **Theorem 2:**
>
> The proof begins by decomposing the factorized matrix $W$ into a product of three matrices: $W = W_d W_{d - 1 : 2} W_1$, where $W\_{d - 1 : 2} := W_{d - 1} \cdots W_2$. It then utilizes concentration bounds established by [3] for the *Lyapunov exponents* of $W_{d - 1 : 2}$ (product of $d - 2$ zero-centered Gaussian $k$-by-$k$ matrices), where the Lyapunov exponents, denoted $\lambda_1 , \ldots , \lambda_k$, are defined by $\lambda_i := \log ( \sigma_i ) / ( d - 2 )$, with $\sigma_1 \geq \cdots \geq \sigma_k$ standing for the singular values of $W_{d - 1 : 2}$. [3] gives non-asymptotic bounds on the deviation of $\lambda_1 , \ldots , \lambda_k$ from their infinite depth ($d \to \infty$) limits $\mu_1 , \ldots , \mu_k$, which satisfy $\mu_1 > \ldots > \mu_k$.  Because $\sigma_i = \exp\left(( d - 2 ) \lambda_i\right)$, and because $\mu_i \neq \mu_j$ for all $i \neq j$, the non-asymptotic bounds imply that when $d$ grows, with high probability, $W_{d - 1 : 2}$ has one singular value much larger than the rest. More precisely, for any $\gamma \in \mathbb{R}\_{> 0}$, the probability that $W_{d - 1 : 2}$ is within $\gamma$ (in Frobenius norm) of a rank one matrix is $1 - \exp ( - \Omega ( d ) )$.
>
> Next, the proof uses properties of standard multivariate Gaussian distributions (namely, their rotational invariance and concentration bounds on their norms), to argue that with high probability, multiplication of $W_{d - 1 : 2}$ by $W_d$ from the left and by $W_1$ from the right preserves the approximate rank one structure, and specifically, $W = W_d W_{d - 1 : 2} W_1$ is within $\gamma$ of a rank one matrix with probability $1 - O ( 1 / d )$. Employing compressed sensing arguments that rely on the RIP and the ground truth matrix being rank one (see, e.g., [4]), it is then proven that the probability of the events $\mathcal{L}\_{\text{train}} ( W ) < \epsilon\_{\text{train}}$ and $\mathcal{L}\_{\text{gen}}( W ) \geq \epsilon\_{\text{train}}c$ occurring simultaneously is $O ( 1 / d )$.
>
> Finally, due to the fact that the distribution of $W$ is rotationally invariant, and that $W$ is with high probability close to a rank one matrix, the probability of $\mathcal{L}\_{\text{train}}(W) < \epsilon\_{\text{train}}$ is independent of the depth $d$, i.e., it is $\Omega(1)$. By the definition of conditional probability, this implies that the probability of $\mathcal{L}\_{\text{gen}}( W ) \geq \epsilon\_{\text{train}}c$ conditioned on $\mathcal{L}\_{\text{train}}( W ) < \epsilon\_{\text{train}}$ is $O ( 1 / d )$. This is the sought-after result.
>
> ---
>
> **References**
>
> [1] Hanin. “Random Fully Connected Neural Networks as Perturbatively Solvable Hierarchies.”
>
> [2] Favaro et al. “Quantitative CLTs in Deep Neural Networks.”
>
> [3] Hanin and Paouris. “Non-asymptotic Results for Singular Values of Gaussian Matrix Products.”
>
> [4] Kutyniok. "Theory and applications of compressed sensing."

---

> ### Author Response · Authors · 2025-08-05
>
> Dear Reviewer,
>
> Kind reminder in light of the approaching deadline: we would greatly appreciate your response to our updated proof sketches.
> Thank you again for your engagement and willingness to consider raising your score.
>
> Best wishes,
>
> Authors

---

> > ### Comment · Reviewer_2Hfc · 2025-08-06
> >
> > Apologize for the late reply. I am currently caught up in multiple deadlines. I do not have any further question as of right now. I will let know about any updates by tomorrow.

---

> > > ### Author Response · Authors · 2025-08-06
> > >
> > > Thank you very much!  We're looking forward to your update.

---

> > > > ### Comment · Reviewer_2Hfc · 2025-08-08
> > > >
> > > > I thank the author for their detailed responses. The new proof sketches are significantly stronger than the original ones, but I don't quite see the technical novelties mentioned in the author's rebuttals. In particular, I am still not sure what are the main differences between Theorem 1 and the equivalence between neural networks vs. Gaussian process. Therefore, I will maintain my current score, but I will raise my confidence to my support for this paper's acceptance.

---

> > > > > ### Author Response · Authors · 2025-08-08
> > > > >
> > > > > Thank you for your engagement!
> > > > >
> > > > > In general, we believe that a theoretical work should be evaluated primarily based on rigor, significance and the novelty of its findings, more than on technical complexity. We are pleased that you recognize the rigor and significance of our study. As for the novelty of our findings, our paper is the first to examine the volume hypothesis in matrix factorization—a common testbed in the theory of neural networks—and the first to formally establish cases where G&C generalizes provably worse than GD, i.e., where the volume hypothesis does not hold.
> > > > >
> > > > > Notwithstanding the above, we stress that our results are not an immediate outcome of existing machinery, and that our proofs involve highly non-trivial technical arguments. For example, in the extension of Theorem 1 to the case where $\epsilon_{\text{train}}$ is unspecified (Appendix C), the proof involves a strong form of random variable convergence, as well as Gaussian Correlation inequalities.

---

### Official Review · Reviewer_yR2t · 2025-06-22

**Clarity:** 4
**Significance:** 2
**Originality:** 3
**Rating:** 5
**Confidence:** 4

**Summary:**

This submission studies the volume hypothesis -- the claim that "most" near-interpolating solutions have small generalization error -- in the context of low-rank matrix factorization. The authors specifically consider a model parameterized by a (deep) feed-forward neural network with i.i.d random initialization. The first main result, Theorem 1, is that as the width of the neural network increases, the probability that a randomly sampled near-interpolating solution has small generalization error tends to zero. On the other hand, the second main result, Theorem 2, is that as the depth increases, the probability that a near-interpolator has small generalization error tends to 1.

**Questions:**

- The initialization in Theorem 1 seems to be an "NTK" initialization (hidden layer variances are 1/width), and the proof relies on showing that the pre-activations are close to Gaussian. It is not particularly surprising that in the linear/kernel regime, interpolating solutions do not generalize -- the intuition from linear regression being that the parameters in the subspace of the datapoints and the orthogonal subspace are independent. Is this understanding correct? And can you comment on what would happen if a different initialization were used that permitted feature learning (say muP/mean-field where the hidden layer variances are 1/width^2)?
- Theorems 1 and 2 focus on what happens if either width or depth is taken to infinity, while the other is fixed. What if both are taken to infinity at the same time (say at a fixed ratio)? What would the generalization error look like?
- I would also appreciate if the authors could comment on my above concerns re the rank 1 assumption in Theorem 2.

**Ethical Concerns:**

["NO or VERY MINOR ethics concerns only"]

**Final Justification:**

The authors have explained how their proof techniques should generalize to rank > 1, which was my main initial concern, and thus I recommend acceptance for this paper.

**Limitations:**

Limitations are adequately addressed.

**Quality:**

3

**Strengths And Weaknesses:**

## Strengths:
- The problem studied here is well motivated, and matrix factorization is a tractable and interesting testbed to understand how randomly sampled near-interpolators generalize. I thus think the results are of interest to the community.
- To the best of my knowledge, the results presented here are novel.
- The submission is well-written and easy to understand. I skimmed the appendices, and the proofs appear to be correct.
- Theorems 1 and 2 are supported empirically.

## Weaknesses
- As the authors indeed admit, the theoretical results rely on a number of assumptions, such as the choice of activation function. In my opinion, the most restrictive assumption is that Theorem 2 requires the ground truth target to be rank 1. The proof of Theorem 2 relies on showing that in the large depth limit, the product $W_d \cdots W_1$ is close to a rank 1 matrix. As such, if the ground truth were actually rank 2, it does not seem like the same proof technique would still work.
- Implications beyond matrix factorization towards more standard neural network settings (such as training a fully connected network on a simple dataset) are not clear.
- (minor) In Figure 2, it is not obvious that as depth increases, the generalization error converges to zero. The authors state that running gradient descent at larger depths is computationally prohibitive. But it seems like it should be straightforward to run guess and check at larger depths?

---

> ### Author Rebuttal · Authors · 2025-07-30
>
> Thank you for highlighting the importance of studying the volume hypothesis in matrix factorization, the novelty of our results, the clarity of our writing, and the solidity of our theory and empirical support. We respond to your reservations and questions below. If you find our responses satisfactory, we would greatly appreciate it if you would consider raising your score.
>
> >In my opinion, the most restrictive assumption is that Theorem 2 requires the ground truth target to be rank 1. The proof of Theorem 2 relies on showing that in the large depth limit, the product $W_d…W_1$  is close to a rank 1 matrix. As such, if the ground truth were actually rank 2, it does not seem like the same proof technique would still work.
>
> Theorem 2 indeed assumes that the ground truth matrix has rank one. While the theorem’s proof currently requires this assumption, its intuition is far more general: as the depth of the network increases, the product of weight matrices (drawn from a Gaussian distribution) tends toward low rank, thus fitting training data will be achieved with lowest rank possible.  We corroborate this intuition empirically in the supplementary material (Figures 5 and 6), demonstrating the conclusions of Theorem 2 with ground truth ranks beyond one. Extending the formal proof of Theorem 2 beyond ground truth rank one is a valuable direction for future research.
>
> >Implications beyond matrix factorization towards more standard neural network settings (such as training a fully connected network on a simple dataset) are not clear.
>
> The validity of our conclusions for real-world non-linear neural networks was empirically demonstrated by Peleg and Hein ([1]), which found that the generalization of G&C is inferior to that of GD with wide networks, but comparable to it with deep networks. From a theoretical perspective, as we note in the text, we hope that our theory will serve as a stepping stone towards analysis of more realistic settings. One promising avenue in this direction would be the analysis of tensor factorization (tensor sensing with neural network overparameterization), which is a model much closer to real-world neural networks than matrix factorization (see, e.g., [2, 3]). Tensor factorization has various similarities to matrix factorization, thus we believe elements of our theory may carry over to this richer model. We will mention this in the conclusion section of the camera-ready version. Thank you!
>
> >(minor) In Figure 2, it is not obvious that as depth increases, the generalization error converges to zero. The authors state that running gradient descent at larger depths is computationally prohibitive. But it seems like it should be straightforward to run guess and check at larger depths?
>
> Thank you for highlighting a potential source of confusion. To clarify, our theory does not imply that the generalization loss of G&C tends to zero as depth increases. Rather, it implies that under G&C, if the training loss is smaller than $\epsilon_{\text{train}}$, then the probability of the generalization loss being less than some universal constant times $\epsilon_{\text{train}}$ tends to one as depth increases. Accordingly, achieving near-zero generalization loss could require a very small $\epsilon_{\text{train}}$, significantly lengthening the run-time of G&C. This point will be emphasized in the camera-ready version of the paper.
>
> Notwithstanding the above, following your question we ran G&C with larger depths, and observed that the generalization loss continues to decline. These results will be added to the camera ready version of the paper.
>
> >The initialization in Theorem 1 seems to be an "NTK" initialization (hidden layer variances are 1/width), and the proof relies on showing that the pre-activations are close to Gaussian. It is not particularly surprising that in the linear/kernel regime, interpolating solutions do not generalize -- the intuition from linear regression being that the parameters in the subspace of the datapoints and the orthogonal subspace are independent. Is this understanding correct? And can you comment on what would happen if a different initialization were used that permitted feature learning (say muP/mean-field where the hidden layer variances are 1/width^2)?
>
> Your intuition is generally correct, but its formalization requires a non-trivial step of showing that under the “NTK” initialization, the entries of the represented matrix $W$ (Equation (4)) are close to being independent. This is the essence behind the proof of Theorem 1.
>
> If the initialization was smaller, in particular “mean field,” then as width increases, the distribution of $W$ would concentrate around zero. It is possible to show that in this regime, when the activation is linear, the entries of $W$ would still be close to independent, thus with high probability, the generalization of G&C would be close to that of the zero matrix (i.e., G&C would still fail).
>
> > Theorems 1 and 2 focus on what happens if either width or depth is taken to infinity, while the other is fixed. What if both are taken to infinity at the same time (say at a fixed ratio)? What would the generalization error look like?
>
> Our theory does not indicate how G&C would compare to GD in a regime where network width and depth grow jointly. This is a very interesting direction for future work.
>
> > I would also appreciate if the authors could comment on my above concerns re the rank 1 assumption in Theorem 2.
>
> See above.
>
> ---
> **References**
>
> [1] Peleg and Hein. “Bias of Stochastic Gradient Descent or the Architecture: Disentangling the Effects of Overparameterization of Neural Networks”
>
> [2] Razin et al. “Implicit Regularization in Tensor Factorization.”
>
> [3] Razin et al. “Implicit Regularization in Hierarchical Tensor Factorization and Deep Convolutional Neural Networks.“

---

> > ### Comment · Reviewer_yR2t · 2025-08-05
> >
> > Thank you for your response, in particular for the references to GD vs G&C for more standard neural network settings, and the comment on how formalizing the NTK intuition is nontrivial.
> >
> > I am still not convinced that the current arguments will work if the ground truth is rank 2 or greater. In particular, the proof of Theorem 2 seems to argue that the top eigenvalue of $W_d\cdots W_1$ is much larger than the second largest eigenvalue. As such, if the ground truth is rank 2 with two equal eigenvalues, it seems very unlikely that $W_d \cdots W_1$ will exactly align with the ground truth. Is my understanding here correct?

---

> ### Author Response · Authors · 2025-08-07
>
> —-
>
> **Quantitative details**
>
>  Consider a square ground truth matrix $W^{*} \in \mathbb{R}^{k \times k}$ of rank two whose two leading singular values are equal (ground truth matrices of higher rank can be treated analogously).  Hanin et al. ([1]) show that the Lyapunov exponents of $W := W\_d \cdots W\_1$, defined as $ \lambda\_i = \frac{\log \sigma\_i}{d}$, $i = 1 , \ldots , k$, where $\sigma\_1 \ge \sigma\_2 \ge \cdots \ge \sigma\_k$ are the singular values of $W$, converge as the depth grows ($d \to \infty$) to independent Gaussians with means $\mu\_1 > \mu\_2 > \dots > \mu\_k$ and variances $O( 1 / d )$.
>
>  The probability of achieving low training loss is lower bounded by the probability that $W$ lies sufficiently close (in Frobenius norm) to the ground truth  (if $W$ and $W^{*}$ are sufficiently close in Frobenius norm, then $W$ will also obtain low training loss).
>
>  By isotropy, the latter probability should be proportional (identical up to multiplicative factors depending on the training loss and the ambient dimension) to the probability of sampling an arbitrary rank two matrix with two equal nonzero singular values.  Thus, the probability of approximating the ground truth matrix (in Frobenius norm) is governed by the event $\lambda\_1 \approx \lambda\_2$.  If one assumes that for large but finite $d$ the limit established by ([1]) provides a sufficiently good approximation (this is the primary claim that requires more work to formalize), this probability should scale like $e^{-d(\mu\_1-\mu\_2)^2}$.
>
> Failure to generalize can occur only if $W$ has rank $\ge 3$ (otherwise, generalization is guaranteed by the RIP).  If we again assume that the limit provides a good approximation for large **but finite $d$**, then by independence this event should have a probability of roughly $e^{-d\bigl[(\mu\_1-\mu\_2)^2 + (\mu\_1-\mu\_3)^2\bigr]}$.  Conditioning on low training loss, the probability of generalization is therefore lower bounded by
> $$
>   1 -
>   \frac{e^{-d\bigl[(\mu\_1-\mu\_2)^2 + (\mu\_1-\mu\_3)^2\bigr]}}
>        {e^{-d(\mu\_1-\mu\_2)^2}}
>   = 1 - e^{-d(\mu\_1-\mu\_3)^2}
>   \xrightarrow[d \to \infty]{}\ 1.
> $$
> Thus, as the depth $d$ increases, the conditional probability of generalization given low training loss tends to 1.
>
> —-
>
> **References**
>
> [1] Hanin and Paouris. “Non-asymptotic Results for Singular Values of Gaussian Matrix Products.”

---

> > ### Comment · Reviewer_yR2t · 2025-08-09
> >
> > Thank you to the authors for explaining in detail the argument for rank > 1. This resolves my initial concern, and so I am happy to increase my score.

---

### Official Review · Reviewer_yf5s · 2025-06-26

**Clarity:** 4
**Significance:** 2
**Originality:** 3
**Rating:** 5
**Confidence:** 3

**Summary:**

This paper studies the question if the use of gradient descent as an optimizer generalizes systematically better than a random sampling strategy, Guess and Check (G&C), at the exemplary problem of (low rank) matrix recovery. More specifically, it uses linear measurements of a (ground truth) low-rank matrix as training data, and measures generalization via the L2 norm of the projection of predicted and ground truth matrix on the subspace orthogonal to the measurements. The work proves that, in the limit of infinite width, G&C does not generalize well, despite the existence of a result proving that – in a specific case – gradient descent does generalize: The probability of G&C generalizing well becomes as bad as randomly drawing from the prior distribution independent of the training data. In contrast, in the limit of infinite depth d, the probability of G&C generalizing converges to 1 at a rate of 1/d. The authors claim that the former result makes this paper the first case where G&C has proven to be inferior to gradient descent. Overall, the paper concludes that the is no simple answer to the question if networks need gradient descent to generalize well.

**Questions:**

Please clarify my main question raised under "strength and weaknesses" above. I might very well just be mistaken on this, as I am not an expert on this topic. If not, please explain how you plan to change the interpretation of the result.

**Ethical Concerns:**

["NO or VERY MINOR ethics concerns only"]

**Final Justification:**

I'd like to thank the authors for the discussion. I like the overall result and contribution of this paper. To my mind, the results are, however, pitched in a little more generality than they contain. As the authors acknowledge in the discussion, creating a difference (on any artificially manufactured problem) between the distributions obtained by applying gradient descent and G&C is sufficient to claim that there exists a case where GD is systematically better than G&C (and vice versa). Manufacturing such a problem is likely not difficult. Thus, the authors' results should rather be interpreted in the specific (but widely studied) context of low-rank matrix factorization than as a trend in deep learning problems in general. Trusting the authors to make this clear in a possible camera-ready version of the paper, I tend towards accepting this work and have updated my score accordingly.

**Limitations:**

yes

**Paper Formatting Concerns:**

no concerns

**Quality:**

3

**Strengths And Weaknesses:**

The paper tackles a very interesting question and is very well-written. All technical results are described (in terms of their meaning, interpretation and intuition) before stating them formally, allowing readers to see the reasons for certain formal definitions. Similarly, the main ideas of the proofs have been stated in the papers, while all details are deferred to the appendix.

While the contribution and overall study are nice, I have some doubts about the conclusions. The following part is based on my (possibly wrong) understanding of the results and I’d highly appreciate a comment/correction in the rebuttal: The results seem to heavily depend on the desire to find low-rank matrices. The setup is not a supervised learning scenario. The training data could be identical for many different distributions of ground truth matrices, i.e., the same y_i could originate from a sparse ground truth matrix or a ground truth matrix with a small $\ell_p$ norm. As for identical training data the resulting matrices produced by gradient descent and G&C will not change, the generalization seems to depend on the (hidden) desired ground truth. If G&C and gradient descent (GD) yield probability distributions p( . | L_{train}<epsilon) that are sufficiently different, doesn’t that immediately mean there are settings where G&C generalizes better than GD and settings where GD generalizes better than GD (because the unknown ground truth distribution could coincide with either of the two without really changing the training data)? In this sense generalization does not seem to be a property to associate with an algorithm (alone).

If we, for instance, consider the least-squares problem $||Ax-y||^2$. Running gradient descent converges to a solution such that $x-x_0 \in \text{kern}(A)^\perp$, i.e., it commonly ends at a solution close to the minimal norm solution (for typical initializations). If $y = Ax_{true}$ and $x_{true}$ is sparse, then the solution of gradient descent will be quite different from $x_{true}$. But would we really blame this on gradient descent not generalizing well? Don’t we need to talk about the property of generalization for the combination of architecture, optimization algorithm and data distribution? If so, this would be something to emphasize in the paper.

---

> ### Author Rebuttal · Authors · 2025-07-30
>
> Thank you for your thoughtful review, and for highlighting that our paper tackles a very interesting question and is very well-written. We respond to your comments and questions below. If you find our response satisfactory, we would be grateful if you would consider raising your score.
>
> You are absolutely correct in that generalization is defined with respect to a ground truth distribution. Indeed, without any assumption about the ground truth, no guarantee can be derived for any learning algorithm. In the context of matrix factorization (and, more generally, matrix recovery problems), the canonical assumption is that the ground truth matrix has low rank (see, e.g., [1, 2, 3, 4, 5, 6]). This assumption is so common that it is often not mentioned explicitly (e.g., “matrix sensing” is used instead of “low rank matrix sensing”). The main motivation behind it is that matrices corresponding to real-world data distributions are often close to being low rank ([7, 8, 9]).  Moreover, when extended from matrices to tensors, the low rank assumption captures real-world data distributions of high dimensions as well ([10]).
>
> We will clarify the above in the camera-ready version of the paper. In particular, we will clarify that our claim of being the first to establish “cases” where the generalization of G&C is inferior to that of GD, refers to cases with a common assumption on the ground truth distribution. Thank you very much for raising this matter!
>
> ---
> **References**
>
> [1] Candes et al. “Exact Matrix Completion via Convex Optimization.”
>
> [2] Sun et al. “Guaranteed Matrix Completion via Non-convex Factorization.”
>
> [3] Tu et al. “Low-rank Solutions of Linear Matrix Equations via Procrustes Flow.”
>
> [4] Bhojanapalli et al. “Global Optimality of Local Search for Low Rank Matrix Recovery.”
>
> [5] Gunasker et al. “Implicit Regularization in Matrix Factorization.”
>
> [6] Razin et al. “Implicit Regularization in Tensor Factorization.”
>
> [7] Bell and Koren. “Lessons from the Netflix Prize Challenge.”
>
> [8] Ma. “Three-Dimensional Irregular Seismic Data Reconstruction via Low-Rank Matrix Completion.”
>
> [9] Kapur et al. “Gene Expression Prediction Using Low-Rank Matrix Completion.”
>
> [10] Razin et al. “Implicit Regularization in Hierarchical Tensor Factorization and Deep Convolutional Neural Networks.“

---

> > ### Comment · Reviewer_yf5s · 2025-08-05
> >
> > Thanks a lot for the explanation. Yet, I have a follow-up question. The hidden dependence on the ground truth distribution means that as soon as GD and G&C give rise to different solution distributions, one has immediately found both, a case where GD generalizes better than G&C and a case where G&C generalizes better than GD, as both could be the ground truth distribution. If your work is the first where G&C has been proven to be inferior to GD, does that mean that no work has previously shown that the solution distributions are different? Or is your result particular to the underlying assumption of the true matrices being low-rank? I.e. was it clear from prior work already that G&C generalizes better than GD if the prior distribution is different from being low-rank?

---

> > > ### Author Response · Authors · 2025-08-06
> > >
> > > Thank you for the follow-up!
> > >
> > > To our knowledge, our paper is the first to establish a theoretical comparison between the generalization of G&C and that of GD (in general, not only in the context of matrix problems).
> > >
> > > You are absolutely correct that in order to prove existence of **some** case where the generalization of G&C is inferior to that of GD, it suffices to prove that G&C and GD can produce different solutions when applied to the same training loss. While we are not aware of such results in the literature, deriving one is not necessarily difficult if the setting under consideration can be contrived. However, an important merit of our theory is that it does not consider a contrived setting, but rather one that is canonical and studied extensively across science and engineering: matrix sensing with low rank ground truth ([1, 2, 3, 4, 5, 6, 7, 8, 9, 10, 11, 12, 13]). We will highlight the above in the camera-ready version of the paper.
> > >
> > > Please let us know if any additional information may be helpful. Otherwise, if you find our response satisfactory, we would be grateful if you would consider raising your score. Thank you!
> > >
> > > Authors
> > >
> > > ---
> > >
> > > **References**
> > >
> > > [1] Recht et al. “Guaranteed Minimum-Rank Solutions of Linear Matrix Equations via Nuclear Norm Minimization.”
> > >
> > > [2] Candes et al. “Exact Matrix Completion via Convex Optimization.”
> > >
> > > [3] Markovsky. “Low Rank Approximation: Algorithms, Implementation, Applications.”
> > >
> > > [4] Foucart and Rauhut. “A Mathematical Introduction to Compressive Sensing.”
> > >
> > > [5] Sun et al. “Guaranteed Matrix Completion via Non-convex Factorization.”
> > >
> > > [6] Tu et al. “Low-rank Solutions of Linear Matrix Equations via Procrustes Flow.”
> > >
> > > [7] Bhojanapalli et al. “Global Optimality of Local Search for Low Rank Matrix Recovery.”
> > >
> > > [8] Bouwmans et al. “Handbook of Robust Low-Rank and Sparse Matrix Decomposition.”
> > >
> > > [9] Gunasker et al. “Implicit Regularization in Matrix Factorization.”
> > >
> > > [10] Razin et al. “Implicit Regularization in Tensor Factorization.”
> > >
> > > [11] Tong et al. “Accelerating Ill-Conditioned Low-Rank Matrix Estimation via Scaled Gradient Descent.”
> > >
> > > [12] Jin et al. “Understanding Incremental Learning of Gradient Descent: A Fine-grained Analysis of Matrix Sensing.”
> > >
> > > [13] Qin et al. “A General Algorithm for Solving Rank-one Matrix Sensing.”

---

### Official Review · Reviewer_a89s · 2025-06-28

**Clarity:** 3
**Significance:** 1
**Originality:** 2
**Rating:** 3
**Confidence:** 5

**Summary:**

The authors investigate the generalization behavior of the Guess & Check (G&C) algorithm in the context of matrix factorization problems, both with and without activation functions. The G&C algorithm operates by sampling parameters from a prior distribution without any gradient-based training, and selecting those that yield sufficiently low training loss. The paper presents two main theoretical findings. First, under certain assumptions, as the network width tends to infinity, the generalization performance of the G&C algorithm degrades to the level of random parameter selection. Second, under specific rank-1 assumption, increasing the network depth can lead to improved generalization performance of the G&C algorithm. The author uses this to illustrate whether gradient-based algorithms are necessary for the generalization of neural networks. The conclusion is that there is no simple answer and specific analysis is required for specific situations.

**Questions:**

1. On the one hand, the authors aim to generalize matrix factorization results to multi-layer neural networks with nonlinear activation functions, which goes beyond the classical matrix factorization setting.  However, why the analysis must be restricted specifically to matrix sensing settings? Could the authors clarify whether this limitation is inherent to their theoretical framework, or if it is possible to extend the results to more general supervised learning settings?

2. According to the restatement of Proposition 1, is it implied that the generalization loss tends to zero as the width $k \to \infty$? If so, I believe this conclusion requires a more careful justification or even a formal proof. It is not immediately clear how this follows from Theorem 3.3.

3. In Figure 1, what is the precise relationship between the plotted curves and the training error threshold $\epsilon_{\text{train}}$? How would the results change if a different value of $\epsilon_{\text{train}}$ were used? A sensitivity analysis would help clarify this.

4. Why is the prior baseline curve omitted in Figure 2? It was included in Figure 1, so it would be helpful to explain the rationale for excluding it here.

4. The experimental section lacks sufficient persuasive power. According to the authors’ theoretical claims:
   - As width increases, the difference between the generalization error of the G&C algorithm and that of random guessing should follow a straight line with slope $-1/2$ in a log-log plot.
   - As depth increases, the generalization error of the G&C algorithm should follow a straight line with slope $-1$ in a log-log plot.

   A more quantitative and explicit visualization of these trends in Figures 1 and 2—e.g., by plotting the log-log slope or fitting lines—would significantly enhance the clarity and credibility of the empirical findings.


Other Minor Comments:

1. Line 26: Reference [102] appears to be cited twice; please check for redundancy.

2. The notation for sets is somewhat confusing. In Line 142, parentheses are used to denote a set, while in Line 148, the same notation appears to indicate concatenation. It would be clearer and more conventional to use curly braces for sets to avoid ambiguity.

**Ethical Concerns:**

["NO or VERY MINOR ethics concerns only"]

**Final Justification:**

Resolved Issues:

Technical issues have been addressed, including clarification of experimental details.

Unresolved Issues:

The authors' response to my primary concerns remains inadequate. First, regarding the theoretical scope limitation to matrix sensing settings: while the authors claim other works also operate under matrix sensing assumptions, most of these studies address the more significant problem of implicit regularization with visible generalization potential. In contrast, this work's results heavily depend on the matrix sensing sampling constraint $n < m \times m'$, making generalization to standard deep learning scenarios with infinite sampling capacity questionable.

Second, concerning the authors' assertion that their theory merely provides an upper bound that may not align with experiments: there is insufficient discussion on bound tightness. We have observed numerous theoretical works offering elegant upper bound theorems (e.g., generalization bounds for deep learning models) that fail to enhance our genuine understanding of the underlying problems. Therefore, we should exercise caution when evaluating such seemingly elegant theorems derived under unrealistic assumptions.

Overall, I maintain that this work's contribution remains quite limited. I preserve my score of 3.

**Limitations:**

In addition to the theoretical assumptions and settings discussed by the authors, the limited empirical validation also constitutes a potential limitation of the paper.

**Quality:**

2

**Strengths And Weaknesses:**

**Strengths**

1. The volume hypothesis that the author is trying to solve seems to be a very interesting problem. Generalization, as the most fundamental ability of neural networks, is guaranteed by the optimization algorithm or by the architecture itself, which is a question worth studying.

2. The authors provide a reasonably comprehensive review of relevant literature and appropriately cite several classical and foundational results in the field.

3. The conclusions that the author wants to convey are relatively clear, and the overall writing logic of the paper is relatively clear.

4. I checked some theoretical details, and the author's theoretical results are relatively solid. The derivations build upon well-established statistical principles, and the arguments are generally rigorous.

**Weakness**

1. My primary concern lies in the theoretical contributions. The core results, particularly Theorems 1 and 2, appear to be largely derived from well-established prior work on the limiting behavior of neural networks as width or depth tends to infinity. For instance, it is a well-known result that infinitely wide neural networks converge to Gaussian processes. The theoretical derivations in the current paper seem to build directly upon such existing limit results to obtain implications for matrix sensing problems. In this context, the derivation of the limiting distribution itself is arguably more fundamental, and the additional insights provided in this work seem rather limited in terms of novelty. I think it would be a greater theoretical contribution if the authors could provide a generalization analysis of the G&C algorithm under finite width/depth (this setting in Chiang et al.).

2. Another major concern is the limited scope of the theoretical results. All analysis in the paper is conducted under the matrix sensing setting, where the number of observed samples $n < m \times m'$. Theorem 1 essentially states that in the infinite-width limit, under undersampling, the training and test data become statistically independent—i.e., no generalization is possible. However, this conclusion heavily relies on the special structure of matrix sensing, particularly the inner-product structure of the measurement operator. I am skeptical whether this result would hold in more general settings, such as learning a target function using a fully connected neural network (which allows unlimited number of samples). In such cases, the training and test are unlikely to be completely independent.

3. The authors cite Proposition 1 to argue that gradient descent (GD) generalizes well in the infinite-width limit. However, I believe this claim requires more careful justification and proof. Specifically, the cited result from [92] shows that for any fixed width  $k$, gradient descent can recover the target matrix with high probability as the initialization scale tends to zero. This setting corresponds to the nonlinear training dynamics regime, and crucially, it does **not** require $k \to \infty$. The authors seem to use it to illustrate the generalization guarantee of GD when the width tends to infinity. I think the conditions of these two theorems are completely different, so the two cannot be compared to show the generalization of GD is guaranteed, and the generalization of G \& C is weaker.

4. Although the authors cite a fairly comprehensive literature, there are some important and related works that are not mentioned. For example, [1, 2] discovered the frequency principle (or spectral bias) of the general neural network structure itself. Because of the smoothness of the activation function, the network itself will learn from low frequency to high frequency, and thus has good generalization for the actual data distribution (often low frequency). In addition, more specifically, for matrix factorization models, [3] systematically studied the implicit regularization effect of data connectivity on the low rank and low nuclear norm of the matrix decomposition problem, especially highlighting the impact of the intrinsic invariant manifold induced by the model structure on global dynamics. In addition, [4] studied how the implicit preference for depth promotes generalization.

[1] Frequency principle: Fourier analysis sheds light on deep neural networks. Communications in Computational Physics, 28(5):1746–1767, 2020.

[2] On the spectral bias of deep neural networks. International Conference on Machine Learning, 2019.

[3] Connectivity shapes implicit regularization in matrix factorization models for matrix completion. In The Thirty-eighth Annual Conference on Neural Information Processing Systems, volume 37, pages 45914–45955, 2024.

[4] The implicit bias of depth: How incremental learning drives generalization, in: International Conference on Learning Representations, 2019.

---

> ### Author Rebuttal · Authors · 2025-07-30
>
> Thank you for highlighting the importance of the problem we study, the rigor of our theoretical arguments, and the clarity of our writing. We respond to your reservations and questions below. In light of our responses, we would greatly appreciate it if you would consider increasing your score.
>
> >My primary concern lies in the theoretical contributions. The core results, particularly Theorems 1 and 2, appear to be largely derived from well-established prior work on the limiting behavior of neural networks as width or depth tends to infinity…
>
> In general, we believe that theoretical results should be evaluated primarily by their rigor, importance and novelty, rather than by the technical complexity of their derivation.  We are glad that you appreciated the rigor and importance of our study.  With regards to novelty, we emphasize that, as stated in the paper, our theory is not only the first to analyze the volume hypothesis in the context of matrix factorization (an important testbed in the theory of neural networks), but also the first to formally prove existence of cases where the generalization attainable by G&C is provably inferior to that of GD.
>
> Notwithstanding the above, from a technical perspective, our theory indeed builds upon existing machinery, e.g., connections between neural networks and Gaussian Processes, and characterizations of singular values in Gaussian matrix products.  We stress however that our results are not an immediate outcome of existing machinery, and some of our proofs involve highly non-trivial arguments. For example, Theorem 2 requires applications of concentration bounds on the Lyapunov exponents of a product of Gaussian square matrices, and the extension of Theorem 1 to the case where $\epsilon_{\text{train}}$ is unspecified (Appendix C) requires a stronger form of random variable convergence, as well as the use of Gaussian Correlation inequalities.
>
> >Another major concern is the limited scope of the theoretical results. All analysis in the paper is conducted under the matrix sensing setting, where the number of observed samples $n<m \times m’$...
>
> As noted by reviewers **tfAT** and **yR2T**, matrix factorization (matrix sensing with neural network overparameterization) is a well established testbed in the theory of neural networks.  Seminal papers in the field focused solely on matrix factorization, e.g., [1, 2, 3, 4, 5].  We believe our work should be held to a similar standard, meaning it should not be disqualified for being limited to matrix factorization (especially given how transparent the text is about this).
>
> With regards to the validity of our conclusions with real-world non-linear neural networks, note that Peleg and Hein ([6]) empirically evaluated such models and produced findings consistent with our theory, namely, findings by which the generalization of G&C is inferior to that of GD with wide networks, but comparable to it with deep networks.
>
> >The authors cite Proposition 1 to argue that gradient descent (GD) generalizes well in the infinite-width limit. However, I believe this claim requires more careful justification and proof. Specifically, the cited result from [92] shows that for any fixed width $k$, gradient descent can recover the target matrix with high probability as the initialization scale tends to zero. This setting corresponds to the nonlinear training dynamics regime, and crucially, it does not require $k\rightarrow\infty$...
>
> We fear there may be a misunderstanding. We do not cite Proposition 1 to argue that GD generalizes well in the infinite width limit. Rather, we combine Proposition 1 with the result in Theorem 1 **applicable to finite widths**, for proving (for the first time, to our knowledge) that there exist (finite width) cases where the generalization attainable by G&C is provably inferior to that of GD.
>
> >Although the authors cite a fairly comprehensive literature, there are some important and related works that are not mentioned. For example…
>
> Thank you for noting that we “cite a fairly comprehensive literature,” and for pointing out the additional references. We will cite them all in the camera-ready version of the paper.
>
> >On the one hand, the authors aim to generalize matrix factorization results to multi-layer neural networks with nonlinear activation functions, which goes beyond the classical matrix factorization setting. However, why the analysis must be restricted specifically to matrix sensing settings?...
>
> Our analysis indeed accounts for non-linear activations, going beyond the classical form of matrix factorization. It is however limited to the matrix sensing problem. As we note in the text, we hope that our theory will serve as a stepping stone towards analysis of more realistic settings. One promising avenue in this direction would be the analysis of tensor factorization (tensor sensing with neural network overparameterization), which is a model much closer to real-world neural networks (see, e.g., [7, 8]).  Tensor factorization has various similarities to matrix factorization, thus we believe elements of our theory may carry over to this richer model. We will mention this in the conclusion section of the camera-ready version.
>
> >According to the restatement of Proposition 1, is it implied that the generalization loss tends to zero as the width $k\rightarrow\infty$? If so, I believe this conclusion requires a more careful justification or even a formal proof. It is not immediately clear how this follows from Theorem 3.3.
>
> Again, we fear there may be a misunderstanding. Proposition 1 includes no new content beyond Theorem 3.3 from [9]. Note that in this latter theorem, the upper bound on the generalization loss is given in terms of $\alpha$, which itself satisfies an upper bound that tends to zero as $k$ grows. For comparing Proposition 1 with Theorem 3.3, you are welcome to consult Appendix E in our paper, which includes a version of Proposition 1 without O-notations.
>
> >In Figure 1, what is the precise relationship between the plotted curves and the training error threshold $\epsilon_{\text{train}}$? How would the results change if a different value of $\epsilon_{\text{train}}$ were used? A sensitivity analysis would help clarify this.
>
> As stated in Appendix G, the plots in Figure 1 were generated with $\epsilon_{\text{train}}=0.02$. Following your question we conducted additional experiments with $\epsilon_{\text{train}}=0.03$ and lower values. All of the results we obtained were qualitatively similar to those reported in Figure 1. We will add this information to the camera-ready version of the paper. Thank you for the question.
>
> >Why is the prior baseline curve omitted in Figure 2? It was included in Figure 1, so it would be helpful to explain the rationale for excluding it here.
>
> We believe it makes less sense to plot generalization of priors in Figure 2, as the purpose of the figure is to demonstrate that the generalization of G&C improves with depth, in contrast to Figure 1 whose purpose was to demonstrate that the generalization of G&C approaches chance (generalization of the prior) with width. Nonetheless, we will specify generalization of priors in Figure 2 (as expected, these are well above the generalization of G&C and GD).
>
> >The experimental section lacks sufficient persuasive power. According to the authors’ theoretical claims:
> >- As width increases, the difference between the generalization error of the G&C algorithm and that of random guessing should follow a straight line with slope -1/2  in a log-log plot.
> >- As depth increases, the generalization error of the G&C algorithm should follow a straight line with slope -1  in a log-log plot.
>
> >A more quantitative and explicit visualization of these trends in Figures 1 and 2—e.g., by plotting the log-log slope or fitting lines—would significantly enhance the clarity and credibility of the empirical findings.
>
> We believe there is a misunderstanding: our theory provides upper bounds (which we do not claim are tight), and these upper bounds pertain to probabilities (rather than generalization errors).
>
> >Line 26: Reference [102] appears to be cited twice; please check for redundancy.
>
> We will fix this.  Thank you.
>
> >The notation for sets is somewhat confusing. In Line 142, partentheses are used to denote a set, while in Line 148, the same notation appears to indicate concatenation. It would be clearer and more conventional to use curly braces for sets to avoid ambiguity.
>
> We use parentheses to denote tuples, and this is appropriate for both lines 142 and 148.
>
> ---
> **References**
>
> [1] Candes et al. “Exact Matrix Completion via Convex Optimization.”
>
> [2] Sun et al. “Guaranteed Matrix Completion via Non-convex Factorization.”
>
> [3] Tu et al. “Low-rank Solutions of Linear Matrix Equations via Procrustes Flow.”
>
> [4] Bhojanapalli et al. “Global Optimality of Local Search for Low Rank Matrix Recovery.”
>
> [5] Gunasker et al. “Implicit Regularization in Matrix Factorization.”
>
> [6] Peleg and Hein. “Bias of Stochastic Gradient Descent or the Architecture: Disentangling the Effects of Overparameterization of
> Neural Networks.”
>
> [7] Razin et al. “Implicit Regularization in Tensor Factorization.”
>
> [8] Razin et al. “Implicit Regularization in Hierarchical Tensor Factorization and Deep Convolutional Neural Networks.“
>
> [9] Soltanolkotabi et al. “Implicit Balancing and Regularization: Generalization and Convergence Guarantees for Overparameterized Asymmetric Matrix Sensing.”

---

> > ### Comment · Reviewer_a89s · 2025-08-07
> >
> > I thank the authors for their response. However, my concerns remain inadequately addressed. First, regarding the theoretical scope being limited to the matrix sensing setting: while the authors note that other works also operate under matrix sensing, most of these studies investigate the more significant problem of implicit regularization with visible extensions to broader contexts. I believe the results in this paper are heavily dependent on the matrix sensing sampling constraint $n < m \times m'$, making it difficult to generalize to general deep learning settings where infinite samples can be drawn. Second, concerning the authors' claim that theory only provides an upper bound that may not match experiments: there is insufficient discussion of whether this bound is tight. We have seen numerous theoretical works provide elegant upper bound theorems (e.g., various generalization bounds for deep learning models), but these contribute little to our genuine understanding of the underlying problems. Overall, I believe the contribution of this work remains quite limited, and I am inclined to maintain my original score.

---

> ### Author Response · Authors · 2025-08-07
>
> Thank you! We are glad to have clarified some of the previous misunderstandings. It seems that there remain additional ones. We address these below. If there is anything else that we can clarify please let us know. Otherwise, we would greatly appreciate it if you would consider raising your score.
>
> >most of these studies investigate the more significant problem of implicit regularization..
>
> We stress that our work investigates this exact problem. Namely, we study the question of whether the implicit regularization (i.e., the generalization in overparameterized settings) necessitates gradient descent, or, alternatively, takes place with any reasonable (non-adversarial) optimizer that fits the training data.
>
> >the results in this paper are heavily dependent on the matrix sensing sampling constraint $n < m \times m’$...
>
> All of the seminal papers which we have mentioned (and which you have acknowledged) [1, 2, 3, 4, 5] make the exact same assumption (i.e., they assume $n < m \times m’$, meaning that the number of measurements is smaller than the number of matrix entries). When this assumption is violated the question of generalization becomes vacuous (generalization is guaranteed just by fitting the training data).
>
> >numerous theoretical works provide elegant upper bound theorems …, but these contribute little to our genuine understanding of the underlying problems.
>
> We reiterate that our bounds apply to probabilities, not generalization losses.
>
> Moreover, papers as you mention include seminal works that significantly advanced the field (e.g., [6, 7, 8, 9, 10, 11, 12]). We believe our work should be held to a similar standard, meaning it should not be disqualified for providing bounds that are not necessarily tight.
>
> ---
>
> **References**
>
> [1] Candes et al. “Exact Matrix Completion via Convex Optimization.”
>
> [2] Sun et al. “Guaranteed Matrix Completion via Non-convex Factorization.”
>
> [3] Tu et al. “Low-rank Solutions of Linear Matrix Equations via Procrustes Flow.”
>
> [4] Bhojanapalli et al. “Global Optimality of Local Search for Low Rank Matrix Recovery.”
>
> [5] Gunasker et al. “Implicit Regularization in Matrix Factorization.”
>
> [6] Bartlett et al. “Spectrally-Normalized Margin Bounds for Neural Networks.”
>
> [7] Neyshabur et al. “A PAC-Bayesian Approach to Spectrally-Normalized Margin Bounds for Neural Networks.”
>
> [8] Arora et al. “Stronger Generalization Bounds for Deep Nets via a Compression Approach.”
>
> [9] Golowich et al. “Size-Independent Sample Complexity of Neural Networks.”
>
> [10] Li and Liang. “Learning Overparameterized Neural Networks via Stochastic Gradient Descent on Structured Data.
>
> [11] Cao and Gu. “Generalization Error Bounds of Gradient Descent for Learning Over-parameterized Deep ReLU
> Networks.”
>
> [12] Bartlett. “The Sample Complexity of Pattern Classiﬁcation with Neural Networks: The Size of the Weights is More
> Important than the Size of the Network.”

---

### Official Review · Reviewer_tfAT · 2025-06-30

**Clarity:** 4
**Significance:** 3
**Originality:** 4
**Rating:** 5
**Confidence:** 3

**Summary:**

This paper studies whether gradient descent is necessary for neural networks to generalise well, using matrix factorisation as a simplified and analytically tractable setting. The authors compare the generalisation performance of gradient descent and a non-optimised alternative known as Guess & Check (G&C). They analyse how generalisation under G&C behaves as network width and depth vary. Their theoretical results suggest that generalisation under G&C deteriorates with increasing width but improves with increasing depth, in contrast to gradient descent, which performs consistently well. These theoretical findings are complemented by empirical experiments within the matrix factorisation framework. The study aims to shed light on the conditions under which gradient descent may — or may not — be necessary for generalisation.

**Questions:**

**1)** In Figure 1c, a double descent behaviour [Belkin et al., 2018] appears to emerge around width $\approx 12$. Would the authors say that, in this setting, this behaviour stems from the algorithmic component of the learning process (i.e., the choice of optimiser), rather than from data or model architecture alone?

**2)** Do the authors have any intuition about the contrasting behaviour under increasing width and increasing depth? I wonder whether this contrast could be related—perhaps even indirectly—to the classical universal approximation theorem for networks of arbitrary width [Cybenko, 1989].

**3)** The numerical experiments are performed in very low dimensions. For instance, in Figure 1, we have $m=m' = 5$ and $n=15$. Could there be finite-size effects influencing the observed trends? This is particularly relevant since Theorem 1 considers the limit $k \to \infty$.

**4)** The paper seems to support the perspective that architecture, data, and algorithmic choices cannot be studied in isolation when analysing generalisation in overparameterised systems [Zdeborová, 2020, Nature Physics]. Do the authors agree with this interpretation, and if so, could they comment on how their findings might inform this broader view?

**Ethical Concerns:**

["NO or VERY MINOR ethics concerns only"]

**Final Justification:**

I thank the authors for their careful and detailed rebuttal addressing all questions and remarks. I maintain my recommendation for acceptance.

**Limitations:**

Yes.

**Paper Formatting Concerns:**

The paper follows the NeurIPS 2025 Paper Formatting.

**Quality:**

3

**Strengths And Weaknesses:**

**Quality**

The paper is solid and presents rigorous theoretical analyses under well-defined assumptions. The authors provide formal results characterising the generalisation performance of G&C as a function of network width and depth. The empirical results are consistent with the theoretical predictions (see question regarding the dimensionality). Naturally, as acknowledged by the authors, the results are derived in a very specific setting (matrix factorisation with specific activations and priors), and their relevance to more realistic networks remains to be established in this context. Since matrix factorisation is a traditional playground in theoretical analysis, its use is well motivated. That said, I would highlight more carefully the additional restrictions imposed by Theorem 2 compared to Theorem 1, and their connection to the most standard settings in matrix factorisation.


**Clarity**

The paper is generally well written and well structured. The motivation is clear, and the presentation of results is good.


**Significance**

The work addresses a timely and important question raised by recent publications: whether gradient-based optimisation is necessary for generalisation. While the matrix factorisation setting is simplified, it serves as a common theoretical proxy. The observed contrast between width and depth is interesting, though its underlying cause still needs to be better understood.


**Originality**

The paper offers new theoretical contributions regarding the limitations and capabilities of G&C, particularly the contrasting behaviour under increasing width and depth. This perspective contributes to the ongoing discussion around the volume hypothesis and the role of implicit bias in generalisation.

---

> ### Author Rebuttal · Authors · 2025-07-30
>
> Thank you for your thoughtful review, and for highlighting the quality, clarity, significance and originality of the paper. We address your comments and questions below.
>
> >In Figure 1c, a double descent behaviour [Belkin et al., 2018] appears to emerge around width ~12. Would the authors say that, in this setting, this behaviour stems from the algorithmic component of the learning process (i.e., the choice of optimiser), rather than from data or model architecture alone?
>
> Thank you for pointing this out! We indeed observe a double descent phenomenon, and only with GD. This suggests that the phenomenon may indeed relate to the optimizer more than the architecture or data. Investigation of this prospect is a very interesting direction for future work, which will be discussed in the camera-ready version of the paper.
>
> >Do the authors have any intuition about the contrasting behaviour under increasing width and increasing depth? I wonder whether this contrast could be related—perhaps even indirectly—to the classical universal approximation theorem for networks of arbitrary width [Cybenko, 1989]
>
> Our results can be interpreted as suggesting that in terms of generalization, neural network architectures benefit from depth more than they do from width. In analogy, various works have shown that in terms of approximation (expressiveness), neural network architectures benefit from depth more than they do from width. We will discuss this analogy in the camera-ready version of the paper. Thank you for bringing it up!
>
> >The numerical experiments are performed in very low dimensions. For instance, in Figure 1, we have $m=m’=5$  and $n=15$. Could there be finite-size effects influencing the observed trends? This is particularly relevant since Theorem 1 considers the limit $k\rightarrow\infty$.
>
> Due to the run-time of G&C being exponential in the number of training examples, we (like all prior works, see for example [1, 2]) are limited to small scale experiments. While this might influence observed trends, our empirical findings generally align with our theoretical predictions (which are not limited to small scale).
>
> Note that Theorem 1 applies to finite width settings, not only the infinite width case.
>
> >The paper seems to support the perspective that architecture, data, and algorithmic choices cannot be studied in isolation when analysing generalisation in overparameterised systems [Zdeborová, 2020, Nature Physics]. Do the authors agree with this interpretation, and if so, could they comment on how their findings might inform this broader view?
>
> Our paper actually points to a more nuanced claim: in some settings (those where the generalization of G&C is inferior to that of GD) generalization crucially depends on the architecture, data and optimizer; whereas in other settings (those where the generalization of G&C is on par with that of GD) generalization stems primarily from the architecture and data, with the optimizer playing a secondary role (it only needs to fit training data in a non-adversarial way). We hope that our study will serve as a stepping stone towards delineating between the two regimes, i.e., towards identifying when the choice of optimizer has significant implications on generalization.
>
> >That said, I would highlight more carefully the additional restrictions imposed by Theorem 2 compared to Theorem 1, and their connection to the most standard settings in matrix factorisation
>
> The additional restrictions imposed by Theorem 2 compared to Theorem 1 are explicitly discussed in our limitations section (lines 347 to 349). Following your comment, we will also highlight them in Section 4.
>
> ---
> **References**
>
> [1] Chiang et al. “Loss Landscapes are All You Need: Neural Network Generalization Can Be Explained Without the Implicit Bias of Gradient Descent.”
>
> [2] Peleg and Hein. “Bias of Stochastic Gradient Descent or the Architecture: Disentangling the Effects of Overparameterization of Neural Networks.”

---

> > ### Comment · Reviewer_tfAT · 2025-08-04
> >
> > I thank the authors for their careful and detailed rebuttal addressing all questions and remarks. I maintain my recommendation for acceptance.

---

> > > ### Author Response · Authors · 2025-08-04
> > >
> > > Thank you for your support! Your thoughtful feedback helped us improve the text.

---

### Note · Authors · 2025-08-16

Dear Committee,

We sincerely thank you for the time and constructive feedback provided during the review and discussion process.  Many of your suggestions have helped us improve the clarity and quality of the manuscript, and we are incorporating them into the camera-ready version as discussed.

Sincerely,

Authors

---

### Decision · Program_Chairs · 2025-09-17

**Decision:**

Accept (poster)

**Comment:**

This paper examines whether gradient descent is necessary for neural networks to generalize effectively, using matrix factorization as a testbed. The authors compare the generalization performance of gradient descent with a non-optimized alternative, Guess & Check (G&C), and analyze how generalization varies with network width and depth. Their theoretical results show that generalization under G&C deteriorates with increasing width but improves with increasing depth, in contrast to gradient descent, which remains consistently strong across settings.

The reviewers agreed that the paper introduces interesting ideas that are well-motivated and carefully validated, supported by both theoretical analysis and empirical evidence. Based on this, I recommend acceptance.